# No-Regret Learning Under Adversarial Resource Constraints: A Spending Plan Is All You Need!

**Francesco Emanuele Stradi**[*]
Politecnico di Milano
francescoemanuele.stradi@polimi.it

**Matteo Castiglioni**
Politecnico di Milano
matteo.castiglioni@polimi.it

**Alberto Marchesi**
Politecnico di Milano
alberto.marchesi@polimi.it

**Nicola Gatti**
Politecnico di Milano
nicola.gatti@polimi.it

**Christian Kroer**
Columbia University
ck2945@columbia.edu

## Abstract

We study online decision making problems under resource constraints, where both reward and cost functions are drawn from distributions that may change adversarially over time. We focus on two canonical settings: $(i)$ *online resource allocation* where rewards and costs are observed before action selection, and $(ii)$ *online learning with resource constraints* where they are observed after action selection, under *full feedback* or *bandit feedback*. It is well known that achieving sublinear regret in these settings is impossible when reward and cost distributions may change arbitrarily over time. To address this challenge, we analyze a framework in which the learner is guided by a *spending plan*—a sequence prescribing expected resource usage across rounds. We design general (primal-)dual methods that achieve sublinear regret with respect to baselines that follow the spending plan. Crucially, the performance of our algorithms improves when the spending plan ensures a well-balanced distribution of the budget across rounds. We additionally provide a robust variant of our methods to handle worst-case scenarios where the spending plan is highly imbalanced. To conclude, we study the regret of our algorithms when competing against benchmarks that deviate from the prescribed spending plan.

## 1 Introduction

In this paper we study online decision making problems with resource constraints. In this class of problems, a decision maker has $m$ resources and a decision set $\mathcal{X}$. Across a sequence of $T$ timesteps, the decision maker must repeatedly choose decisions $\boldsymbol{x}_t \in \mathcal{X}$, where each decision leads to some resource depletion specified by a cost function $c_t(\boldsymbol{x}_t) \in [0,1]^m$ and a reward $f_t(\boldsymbol{x}_t) \in [0,1]$. We study two canonical settings. $(i)$ *Online resource allocation* (ORA). In this setting, the decision maker observes the reward and cost functions $f_t, c_t$ before choosing $\boldsymbol{x}_t$. This setting captures problems such as budget management in second-price auctions [12] and network revenue management [57]. $(ii)$ *Online learning with resource constraints* (OLRC), where the decision maker chooses $\boldsymbol{x}_t$ first, and *then* receives some feedback on rewards and costs. Under *full feedback*, they observe the entire

---

[*]Corresponding author.

39th Conference on Neural Information Processing Systems (NeurIPS 2025).

functions $f_t, c_t$, while under *bandit feedback*, only the values $f_t(\boldsymbol{x}_t)$ and $c_t(\boldsymbol{x}_t)$. This setting captures problems such as budget management in first-price auctions and online pricing [7, 15, 19].

Across the two settings, there have been extensive studies on what types of regret guarantees are possible under different input models. First, in ORA one studies dynamic benchmarks that vary the decision over time (this is made possible due to observing $f_t, c_t$ before making a decision), whereas in OLRC the benchmark is the best single decision in hindsight. In both settings, it is known that it is possible to achieve $O(\sqrt{T})$ regret when the rewards and costs are drawn from a fixed distribution at each time step [14, 7]. In contrast, Balseiro and Gur [12] show that it is not possible to achieve no-regret guarantees when the inputs $f_t, c_t$ are chosen adversarially. Their result is for ORA, but it extends easily to OLRC. It is also possible to achieve "best-of-both worlds" guarantees where a single algorithm asymptotically achieves the optimal regret guarantee in the stochastic setting while simultaneously guaranteeing the optimal competitive ratio on adversarial input [14, 19].

In real-world settings such as when an advertiser performs budget management in internet advertising, the environment is not stationary, both due to the time-varying nature of internet traffic, but also due to the fact that other advertisers are simultaneously adjusting their bidding behavior. Yet the worst-case guarantees offered in the adversarial setting are not sufficiently strong for such real-world scenarios. To address this issue, platforms such as Meta and Google typically provide a predicted *spending plan* [38, 13]. A spending plan for a given advertiser specifies a recommended amount that the advertiser should aim to spend in each timestep. In other words, the spending plan takes the advertiser's daily (or weekly) budget, and allocates it non-uniformly across time based on the predicted behavior of the advertising ecosystem in each timestep. Typically, advertisers cede control of their budget management to an algorithm offered by the platform. Such an algorithm typically takes as input the spending plan and uses a control algorithm (known as a *pacing* algorithm in the ad auction industry) to attempt to match the predicted spending plan.

**Contributions**   Inspired by the use of spending plans in practice, we study the two types of online decision making with resource constraints—ORA and OLRC—in scenarios where the decision maker is additionally given a suggested spending plan, and the baseline that we compare to must similarly follow the spending plan. We focus on the case in which both reward and cost functions are sampled from distributions that adversarially change over time, thus generalizing the standard adversarial case. In ORA, we develop a dual algorithm that exploits the knowledge of the spending plan to attain *dynamic* regret of order $\widetilde{O}(\frac{1}{\rho_{\min}}\sqrt{T})$, where $T$ is the horizon and $\rho_{\min}$ is the minimum per-round budget expenditure of the spending plan, *i.e.*, the Slater parameter of the offline allocation problem. Next, we focus on the OLRC setting. For the *full feedback* case, we develop a primal-dual procedure that attains *static* regret of order $\widetilde{O}(\frac{1}{\rho_{\min}}\sqrt{T})$. Similarly, we show that the results attained with *full feedback* can be generalized to the *bandit feedback* setting. All the algorithms mentioned above employ black box regret minimizers as primal and dual algorithms in order to be as general as possible.[2] We then focus on the case in which $\rho_{\min}$ can be arbitrarily small and the results mentioned above become vacuous. For this case, we propose a general meta-procedure that modifies the input parameters of our (primal-)dual methods, thereby attaining sublinear regret in the worst-case scenario, which is when $\rho_{\min} \leq O(T^{-1/4})$. For ORA, we show that the meta-procedure can still obtain a $\widetilde{O}(T^{3/4})$ *dynamic* regret. For OLRC, we show that our meta-procedure guarantees $\widetilde{O}(T^{3/4})$ *static* regret. Finally, we show that the results mentioned above are robust to optimal solutions that follow the spending plan up to a sublinear error, thus analyzing sensitivity to suboptimal spending plans.

**Comparison with [34, 13]**   In the context of ORA, variations of the spending plan problem have been previously studied by Jiang et al. [34], Balseiro et al. [13]. Specifically, [34, 13] study the ORA problem in which rewards and costs are drawn from adversarially changing distributions. In this setting, they assume that a certain amount of samples from the time-varying distributions are given beforehand. Then, the authors show how to use samples to build a spending plan that guarantees no regret during the budget-constrained learning dynamic. On the one hand, [34, 13] do not assume that a spending plan is given as input. Instead, their works focus on how to design robust spending plans and algorithms using available prior data. On the other hand, their analysis relies on the assumption that there exists a $\kappa \in \mathbb{R}_{\geq 0}$ s.t. $f_t(\boldsymbol{x}) \leq \kappa c_t(\boldsymbol{x})[i]$ for all $\boldsymbol{x} \in \mathcal{X}, i \in [m], t \in [T]$, and their regret

---

[2]We remark that, in the ORA setting, there is no need for a primal regret minimizer.

bound scales as $\widetilde{O}(\kappa\sqrt{T})$. This assumption allows them to avoid one of the main challenges in our work, which is the dependence on the minimum per-round budget provided by the spending plan.

We refer to Appendix A for a more detailed discussion of related work.

## 2 Preliminaries

We study problems where a decision maker (the learner) is given $T$ rounds, $m$ resources, and a non-empty set of available strategies $\mathcal{X} \subseteq \mathbb{R}^n$, which may be non-convex, integral or even non-compact. At each round $t \in [T]$,[3] the learner selects a strategy $\boldsymbol{x}_t \in \mathcal{X}$, gains a reward $f_t(\boldsymbol{x}_t) \in [0,1]$, and pays a cost $c_t(\boldsymbol{x}_t)[i] \in [0,1]$ for each resource $i \in [m]$.[4] At each round $t \in [T]$, the reward function $f_t : \mathcal{X} \to [0,1]$ is sampled from a reward distribution $\mathcal{F}_t$, while the cost function $c_t : \mathcal{X} \to [0,1]^m$ is sampled from a cost distribution $\mathcal{C}_t$.[5] We do *not* make any assumption on how $\mathcal{F}_t$ and $\mathcal{C}_t$ are selected, namely, we allow them to be chosen adversarially, and thus change arbitrarily over the rounds. In the following, we will denote by $\bar{f}_t : \mathcal{X} \to [0,1]$ the expected value of $\mathcal{F}_t$ and by $\bar{c}_t : \mathcal{X} \to [0,1]^m$ the expected value of $\mathcal{C}_t$.

Each resource $i \in [m]$ is endowed with a budget $B_i > 0$ that the learner is allowed to spend over the $T$ rounds. Without loss of generality, we assume that $B_i = B$ for all $i \in [m]$.[6] As is standard in the literature, we focus on the regime where $B = \Omega(T)$ and we define the average budget per round as $\rho := B/T$. The interaction between the learner and the environment stops at any round $\tau \in [T]$ in

---
**Protocol 1** Learner-Environment Interaction
---
1: **for** $t \in [T]$ **do**
2:     $\mathcal{F}_t$ and $\mathcal{C}_t$ are selected adversarially
3:     Reward function $f_t \sim \mathcal{F}_t$ and cost function $c_t \sim \mathcal{C}_t$ are sampled
4:     Learner observes $f_t$ and $c_t$     ▷ ORA
5:     Learner chooses a strategy mixture $\boldsymbol{\xi}_t \in \Xi$
6:     Learner plays a strategy $\boldsymbol{x}_t \sim \boldsymbol{\xi}_t$
7:     Learner observes $f_t$ and $c_t$     ▷ OLRC *Full feed.*
8:     Learner observes $f_t(\boldsymbol{x}_t)$ and $c_t(\boldsymbol{x}_t)$     ▷ OLRC *Bandit feed.*
9:     Learner gets reward $f_t(\boldsymbol{x}_t)$ and pays cost $c_t(\boldsymbol{x}_t)[i], \forall i \in [m]$
10: **end for**
---

which the total cost associated to any resource $i \in [m]$ exceeds its budget $B_i$. The goal of the learner is to maximize the cumulative rewards attained during the learning process. As is standard in the literature (see, *e.g.*, [14]), we assume that there exists a void action $\boldsymbol{x}^\varnothing \in \mathcal{X}$ such that $f_t(\boldsymbol{x}^\varnothing) = 0$ for all $t \in [T]$, and $c_t(\boldsymbol{x}^\varnothing)[i] = 0$, for every $i \in [m]$ and $t \in [T]$. This is done in order to guarantee that a feasible solution exists, namely, a sequence of decisions $\{\boldsymbol{x}_t\}_{t=1}^T$ that does not violate the budget constraints. In the following, we will refer to the set of probability measures on $\mathcal{X}$ as $\Xi$, and we will call it the set of strategy mixtures. Moreover, we will denote by $\boldsymbol{\xi}^\varnothing$ the Dirac strategy that deterministically plays the void action $\boldsymbol{x}^\varnothing$.

Protocol 1 depicts the learner-environment interaction in the ORA setting, in the OLRC setting with *full feedback*, and in the OLRC with *bandit feedback*.

**Remark 2.1** (Relation to the adversarial setting). *In the standard adversarial setting, $f_t$ and $c_t$ are directly selected adversarially (refer to [14] for adversarial ORA and to [32] for adversarial OLRC). Our framework generalizes the adversarial setting. While the reward functions $f_t$ and the cost functions $c_t$ are sampled from a distribution at each round, the distributions are allowed to change over the rounds. Thus, the standard adversarial setting can be recovered by assuming that $\mathcal{F}_t$ and $\mathcal{C}_t$ always put their mass on a single point.*

We now introduce the notion of a *spending plan*. The learner is provided with a sequence $\mathcal{B}_T^{(i)} := \{B_1^{(i)}, \ldots, B_T^{(i)}\}$ of per-round budgets for each resource $i \in [m]$, where $\sum_{t=1}^T B_t^{(i)} = B$ for all $i \in [m]$ and $B_t^{(i)} \in [0,1]$ for all $t \in [T], i \in [m]$. We refer to $\mathcal{B}_T^{(i)}$ as the spending plan for the $i$-th resource, as it defines the maximum budget the learner can allocate *at each time step* $t \in [T]$, in expectation. Notice that the spending plan does not define hard budget constraints, differently from

---

[3]We denote by $[n]$ the set $\{1, \ldots, n\}$ with $n \in \mathbb{N}_{>0}$.

[4]Given a vector $\boldsymbol{v}$, we will denote by $\boldsymbol{v}[j]$ its $j$-th component.

[5]Notice that $\mathcal{F}_t$ and $\mathcal{C}_t$ may be correlated.

[6]A problem with arbitrary budgets can be reduced to one with equal budgets by dividing, for every resource $i \in [m]$, all per-round resource consumptions $c_t(\cdot)[i]$ by $B_i / \min_j B_j$.

the overall budget constraint defined in Section 2. Indeed, the learner is allowed to deviate from the prescribed spending plan $\mathcal{B}_T^{(i)}$, as long as the overall budget constraint is satisfied.

We introduce two baselines to evaluate the performance of our algorithms. The *dynamic* optimal solution is defined by means of the following optimization problem:

$$\text{OPT}_{\mathcal{D}} := \begin{cases} \sup_{\boldsymbol{\xi} \in \Xi^T} & \mathbb{E}_{\boldsymbol{x}_t \sim \boldsymbol{\xi}_t} \left[ \sum_{t=1}^T \bar{f}_t(\boldsymbol{x}_t) \right] \\ \text{s.t.} & \mathbb{E}_{\boldsymbol{x}_t \sim \boldsymbol{\xi}_t} \left[ \bar{c}_t(\boldsymbol{x}_t)[i] \right] \leq B_t^{(i)} \ \ \forall i \in [m], \forall t \in [T] \end{cases}, \tag{1}$$

where $\Xi^T := \bigtimes_{t=1}^T \Xi$ and $\boldsymbol{\xi}_t$ is the $t$-th component of $\boldsymbol{\xi}$. Problem (1) computes the expected value attained by the optimal *dynamic* strategy mixture that satisfies the spending plan. The *dynamic* baseline is common in the ORA literature [14].

Similarly, we introduce the *fixed* optimum in hindsight by means of the optimization problem:

$$\text{OPT}_{\mathcal{H}} := \begin{cases} \sup_{\boldsymbol{\xi} \in \Xi} & \mathbb{E}_{\boldsymbol{x} \sim \boldsymbol{\xi}} \left[ \sum_{t=1}^T \bar{f}_t(\boldsymbol{x}) \right] \\ \text{s.t.} & \mathbb{E}_{\boldsymbol{x} \sim \boldsymbol{\xi}} \left[ \bar{c}_t(\boldsymbol{x})[i] \right] \leq B_t^{(i)} \ \ \forall i \in [m], \forall t \in [T] \end{cases}. \tag{2}$$

Problem (2) computes the expected value attained by the hindsight-optimal fixed strategy mixture that satisfies the spending plan. The fixed baseline is standard in OLRC [32, 19].

We define the minimum per-round budget in the spending plan as $\rho_{\min} := \min_{i \in [m]} \min_{t \in [T]} B_t^{(i)}$. Notice that $\rho_{\min}$ is the Slater parameter of both Problems (1) and (2). Indeed, $\rho_{\min}$ denotes the minimum margin by which the void action $\boldsymbol{x}^{\varnothing}$ satisfies the constraints defined by the spending plan. Clearly, when $\rho_{\min} = 0$, Slater's constraint qualification does not hold, meaning that the problem does not admit a strictly feasible solution. Intuitively, when $\rho_{\min}$ is large, the spending plan distributes the budget $B$ reasonably well across the rounds. This will lead to better performance during the learning dynamic.

**Performance Metrics**  In the ORA setting, we evaluate the performance of our algorithm via its *dynamic* cumulative regret $\mathfrak{R}_T := \text{OPT}_{\mathcal{D}} - \sum_{t=1}^T f_t(\boldsymbol{x}_t)$, which compares the total reward attained by the algorithm with the optimal *dynamic* solution that follows the spending plan recommendations. In the OLRC setting, we evaluate the performance of our algorithm via its *static* cumulative regret $R_T := \text{OPT}_{\mathcal{H}} - \sum_{t=1}^T f_t(\boldsymbol{x}_t)$, which compares the total reward attained by the algorithm with the optimal *fixed* solution that follows the spending plan recommendations. The aim of our work is to develop algorithms that attain sublinear regret bounds in their specific setting: $\mathfrak{R}_T = o(T)$ in ORA, and $R_T = o(T)$ in OLRC.

**Remark 2.2** (On the impossibility result of learning with budget constraints)**.** *It is well known that when no spending plan is available, it is impossible to achieve sublinear regret bounds in all the settings presented in this work, that is, when rewards and costs are allowed to vary over time (see, e.g., [12] for ORA and [32] for OLRC). We will show that exploiting the information given by the spending plan is enough to achieve sublinear regret.*

**Regret Minimizers**  We will employ regret minimizers (that is, no-regret algorithms) as black-box tools. This is done to make the results as general as possible. Specifically, a *regret minimizer* $\mathcal{R}^A$ for a decision space $\mathcal{W}$ is an abstract model of a decision maker that repeatedly interacts with a black-box environment. At each time step $t \in [T]$, $\mathcal{R}^A$ may perform two operations: (i) $\mathcal{R}^A.\texttt{SelectDecision}()$, which outputs an element $w_t \in W$; (ii) $\mathcal{R}^A.\texttt{ReceiveFeedback}(r_t)$ (alternatively, $\mathcal{R}^A.\texttt{ReceiveFeedback}(r_t(w_t))$), which updates the regret minimizer's internal state using feedback received from the environment. The feedback is given in terms of a reward function $r_t : \mathcal{W} \to [a, b]$, where $[a, b] \subset \mathbb{R}$, when the regret minimizer is tailored for *full feedback*. When the regret minimizer is tailored for *bandit feedback*, the feedback is given in terms of the reward attained in the previous timestep $r_t(w_t)$. The reward function $r_t$ may depend adversarially on the previous decisions $w_1, \ldots, w_{t-1}$. The goal of $\mathcal{R}^A$ is to output a sequence $w_1, \ldots, w_T$ such that its cumulative regret, defined as $\sup_{w \in \mathcal{W}} \sum_{t=1}^T (r_t(w) - r_t(w_t))$, grows sublinearly with the time horizon $T$, that is, it is $o(T)$.[7] For convenience, we introduce the notion of a *regret minimizer constructor*, a procedure

---

[7]We underline that regret minimizers with *bandit feedback* generally attain sublinear regret bounds which hold with probability at least $1 - O(\delta)$, for all $\delta \in (0, 1)$.

denoted as $\mathcal{R}^A.\texttt{Initialize}(\mathcal{W}, [a, b])$ that builds a regret minimizer based on two input parameters: the decision set $\mathcal{W}$ and the payoff range $[a, b]$. This constructor returns a regret minimizer tailored to the given inputs, with the guarantee that its cumulative regret grows sublinearly in $T$. We will refer to the regret upper bound of the regret minimizer $\mathcal{R}^A$ at time $t \in [T]$ as $R_t^A$. Throughout the paper, we will assume that $R_t^A \leq R_{t'}^A$ for all $t < t'$.

## 3 Online Resource Allocation

In this section, we provide the algorithm and analysis for the ORA setting.

Following the ORA literature, our algorithm works with the Lagrangian formulation of the problem where the budget constraints are relaxed with a Lagrange multiplier vector $\boldsymbol{\lambda}$, namely $\mathfrak{L}_{f,c,\mathfrak{B}}(\boldsymbol{\xi}, \boldsymbol{\lambda}) := \left[ \mathbb{E}_{\boldsymbol{x} \sim \boldsymbol{\xi}}[f(\boldsymbol{x})] + \sum_{i \in [m]} \boldsymbol{\lambda}[i] \cdot \left( \mathfrak{B}^{(i)} - \mathbb{E}_{\boldsymbol{x} \sim \boldsymbol{\xi}}[c(\boldsymbol{x})[i]] \right) \right]$, where $f, c$ are arbitrary reward and cost functions, and $\mathfrak{B}$ is the per-round expenditure goal. Since both the reward function and the cost one are observed at the beginning of the round we have that the optimal strategy mixture $\boldsymbol{\xi}_t$ of the Lagrangian function at time $t \in [T]$ can be computed directly for a fixed $\boldsymbol{\lambda}_t$ by solving $\arg\max_{\boldsymbol{\xi}} \mathfrak{L}_{f_t, c_t, \mathfrak{B}}(\boldsymbol{\xi}, \boldsymbol{\lambda}_t)$. On the other hand, the choice of $\boldsymbol{\lambda}_t$ is not as obvious. This choice is handled by employing a dual regret minimizer $\mathcal{R}^D$.

---

**Algorithm 1** Dual Algorithm for ORA

---

**Require:** Horizon $T$, budget $B$, spending plans $\mathcal{B}_T^{(i)}$ for all $i \in [m]$, dual regret minimizer $\mathcal{R}^D$ (*full feedback*)
1: Set $B_{i,1} = B$, for all $i \in [m]$
2: Define $\mathcal{L} := \left\{ \boldsymbol{\lambda} \in \mathbb{R}_{\geq 0}^m : \|\boldsymbol{\lambda}\|_1 \leq 1/\rho_{\min} \right\}$
3: $\mathcal{R}^D.\texttt{Initialize}\left( \mathcal{L}, [-1/\rho_{\min}, 1/\rho_{\min}] \right)$
4: $\boldsymbol{\lambda}_1 \leftarrow \mathcal{R}^D.\texttt{SelectDecision}()$
5: **for** $t \in [T]$ **do**
6:      Observe reward function $f_t$ and costs function $c_t$
7:      $\boldsymbol{\xi}_t \leftarrow \arg\max_{\boldsymbol{\xi} \in \Xi} \left[ \mathbb{E}_{\boldsymbol{x} \sim \boldsymbol{\xi}}[f_t(\boldsymbol{x})] - \sum_{i \in [m]} \boldsymbol{\lambda}_t[i] \cdot \mathbb{E}_{\boldsymbol{x} \sim \boldsymbol{\xi}}[c_t(\boldsymbol{x})[i]] \right]$
8:      Play strategy: $\boldsymbol{x}_t \leftarrow \begin{cases} \boldsymbol{x} \sim \boldsymbol{\xi}_t & \text{if } B_{i,t} \geq 1, \ \forall i \in [m] \\ \boldsymbol{x}^\varnothing & \text{otherwise} \end{cases}$
9:      Update budget availability $B_{i,t+1} \leftarrow B_{i,t} - c_t(\boldsymbol{x}_t)[i], \forall i \in [m]$
10:      $r_t^D : \mathcal{L} \ni \boldsymbol{\lambda} \mapsto -\sum_{i \in [m]} \boldsymbol{\lambda}[i] \cdot \left( B_t^{(i)} - \mathbb{E}_{\boldsymbol{x} \sim \boldsymbol{\xi}_t}[c_t(\boldsymbol{x})[i]] \right)$
11:      $\mathcal{R}^D.\texttt{ReceiveFeedback}(r_t^D)$
12:      Update dual variable $\boldsymbol{\lambda}_{t+1} \leftarrow \mathcal{R}^D.\texttt{SelectDecision}()$
13: **end for**

---

In Algorithm 1, we provide the pseudocode of our algorithm. Specifically, the algorithm requires as input a dual regret minimizer $\mathcal{R}^D$ with *full feedback*. Notice that the Lagrangian variable receives full feedback at each episode, as the reward $r_t^D$ associated to each possible $\boldsymbol{\lambda} \in \mathcal{L}$, where $\mathcal{L}$ is the Lagrangian space, can easily be computed. Line 1 initializes the budget available to the learner. Then, the Lagrangian space is built according to $\rho_{\min}$, the Slater parameter of Program (1). It is well-known that in the standard ORA problem, it is sufficient to set $\mathcal{L} := \left\{ \boldsymbol{\lambda} \in \mathbb{R}_{\geq 0}^m : \|\boldsymbol{\lambda}\|_1 \leq 1/\rho \right\}$ to obtain the optimal competitive ratio[8]. In the ORA with spending plan setting, we extend this idea by bounding the Lagrangian space given the minimum per-round budget given by the spending plan. The dual algorithm is initialized to work on the $\mathcal{L}$ decision space and with a payoff range $[-1/\rho_{\min}, 1/\rho_{\min}]$, while the first Lagrangian vector $\boldsymbol{\lambda}_1$ is selected given the initialization of the algorithm (Line 3 - 4). At each round $t \in [T]$, the algorithm observes the feedback (Line 6), selects the strategy mixture $\boldsymbol{\xi} \in \Xi$ which maximizes the Lagrangian $\mathfrak{L}_{f_t, c_t, B_t}(\boldsymbol{\xi}, \boldsymbol{\lambda}_t)$ (Line 7) and accordingly plays the strategy when enough budget is available (Line 8). Then, the algorithm updates the remaining budget (Line 9), builds the dual reward $r_t^D$ for all $\boldsymbol{\lambda} \in \mathcal{L}$ (Line 10) and inputs it to the dual regret minimizer (Line 11). Finally, the Lagrangian multipliers are updated given the feedback (Line 12).

---

[8]In the ORA setting, when online mirror descent with fixed learning rate is employed, the dual regret minimizer can be directly instantiated in the positive orthant [14]. We explicitly bound the Lagrangian space to allow the choice of dual regret minimizers with time-varying learning rates, which generally need explicit bounds on the "diameter" of the decision space.

We highlight a subtle distinction between deterministic and randomized adversarial rewards in the spending-plan setting. In standard adversarial ORA settings where both the reward and the costs are deterministically chosen at each round, the spending-plan ORA problem can trivially be solved by setting $\boldsymbol{\xi}_t \leftarrow \operatorname{argmax}_{\boldsymbol{\xi} \in \Xi} \mathbb{E}_{\boldsymbol{x} \sim \boldsymbol{\xi}} [f_t(\boldsymbol{x})]$ s.t. $\mathbb{E}_{\boldsymbol{x} \sim \boldsymbol{\xi}} [c_t(\boldsymbol{x})[i]] \leq B_t^{(i)}, \forall i \in [m]$ (for the sake of simplicity, we are assuming that this problem admits a solution). This is *not* the case for our setting, because the baseline satisfies the spending plan only in expectation, whereas we observe only a sample from the per-round reward and cost distributions.

## 3.1 Theoretical Results

In this section we show theoretical guarantees attained by Algorithm 1. For additional lemmas and omitted proofs, we refer to Appendix B.1. We first provide the following lemma which lower bounds the expected utility attained by Algorithm 1.

**Lemma 3.1.** *For any $\delta \in (0, 1)$, Algorithm 1, when instantiated with a dual regret minimizer which attains a regret upper bound $R_T^D$, guarantees, with probability at least $1 - \delta$, $\sum_{t=1}^{T} \mathbb{E}_{\boldsymbol{x} \sim \boldsymbol{\xi}_t} [f_t(\boldsymbol{x})] \geq$ $\mathrm{OPT}_{\mathcal{D}} - (T - \tau) - (4 + 4 \max_{\boldsymbol{\lambda} \in \mathcal{L}} \|\boldsymbol{\lambda}\|_1) \sqrt{2\tau \ln \frac{T}{\delta}} - \sum_{t=1}^{\tau} \sum_{i \in [m]} \boldsymbol{\lambda}[i] \cdot (B_t^{(i)} - \mathbb{E}_{\boldsymbol{x} \sim \boldsymbol{\xi}_t} [c_t(\boldsymbol{x})[i]]) - \frac{2}{\rho_{\min}} R_\tau^D$, where $\boldsymbol{\lambda} \in \mathcal{L}$ is an arbitrary Lagrange multiplier and $\tau$ is the stopping time of the algorithm.*

Lemma 3.1 states that the expected cumulative reward attained by the algorithm until the stopping time $\tau$ is lower bounded by the *dynamic* optimum $\mathrm{OPT}_{\mathcal{D}}$ minus the following quantities. The second term $T - \tau$ measures early stopping time. The third term arises from a concentration argument. The fourth term captures the *violation* of the spending plan constraint by the algorithm during the learning dynamic. Finally, the last term is the regret guarantee attained by the dual algorithm. The multiplicative factor $(2/\rho_{\min})$ is due to the payoff range given to the dual regret minimizer.

We are now ready to show the final *dynamic* regret bound of Algorithm 1.

**Theorem 3.2.** *For any $\delta \in (0, 1)$, Algorithm 1 instantiated with a dual regret minimizer which attains a regret upper bound $R_T^D$, guarantees $\mathfrak{R}_T \leq 1 + \frac{1}{\rho_{\min}} + \frac{2}{\rho_{\min}} R_T^D + \left(8 + \frac{8}{\rho_{\min}}\right) \sqrt{2T \ln \frac{T}{\delta}}$, with probability at least $1 - 2\delta$.*

Theorem 3.2 has a linear dependence on the quantity $1/\rho_{\min}$. We underline that it is standard for (primal-)dual methods to have a dependence on the inverse of the Slater parameter of the offline problem in the regret bound (*e.g.* [25, 19, 14, 51]). From a technical perspective this occurs because when $T > \tau$ (*i.e.* the algorithm has depleted the budget associated with some resource), the Lagrangian variable associated with that specific resource must be set at $1/\rho_{\min}$ to compensate the loss in terms of rewards given by the difference $T - \tau$. Given Theorem 3.2, it is easy to see that by employing online mirror descent [46] as a dual regret minimizer, the dynamic regret guarantee of Algorithm 1 is of order $\mathfrak{R}_T \leq \widetilde{O}(1/\rho_{\min} \sqrt{T})$.

## 4 Online Learning with Resource Constraints

In this section we study the OLRC setting under both *full feedback* and *bandit feedback*. Notice that when the the reward and cost functions cannot be observed at the beginning of the round, it is provably not possible to achieve $\mathfrak{R}_T = o(T)$. Thus, we will focus on the less challenging objective of attaining $R_T = o(T)$. In Algorithm 2, we provide the pseudocode of our primal-dual method for OLRC. The parts highlighted in red are specific to OLRC with *full feedback* while the ones highlighted in blue are specific to OLRC with *bandit feedback*. The key differences between Algorithm 2 and our dual algorithm for the *online resource allocation* problem are as follows. First, since $f_t$ and $c_t$ are not revealed in advance, the per-round optimal strategy mixture $\boldsymbol{\xi}_t$, which maximizes the Lagrangian function $\mathfrak{L}_{f_t, c_t, B_t}(\boldsymbol{\xi}, \boldsymbol{\lambda}_t)$ at time $t \in [T]$, *cannot* be computed. We address this problem by employing a primal regret minimizer $\mathcal{R}^P$ which optimizes over the decision space $\Xi$. In OLRC with *full feedback*, it is sufficient to employ a primal regret minimizer tailored for full feedback. In the *bandit feedback* case, a primal regret minimizer tailored for bandit feedback is necessary (*e.g.*, EXP-3 IX [45], when $\mathcal{X}$ is a discrete number of arms). The primal regret minimizer is initialized with the strategy mixture decision space and the payoff range defined by the per-round Lagrangian (Line 3). At each round $t \in [T]$, the strategy mixture is selected by the primal regret minimizer $\mathcal{R}^P$ (Line 6). In

---

**Algorithm 2** Primal-Dual Algorithm for OLRC

---

**Require:** Horizon $T$, budget $B$, spending plans $\mathcal{B}_T^{(i)}$ for all $i \in [m]$, primal regret minimizer $\mathcal{R}^P$ (*full feedback/bandit feedback*), dual regret minimizer $\mathcal{R}^D$ (*full feedback*)

1: Set $B_{i,1} = B$, for all $i \in [m]$
2: Define $\mathcal{L} := \left\{ \boldsymbol{\lambda} \in \mathbb{R}_{\geq 0}^m : \|\boldsymbol{\lambda}\|_1 \leq 1/\rho_{\min} \right\}$
3: $\mathcal{R}^P.\texttt{Initialize}\left(\Xi, [-1/\rho_{\min}, 1 + 1/\rho_{\min}]\right)$
4: $\mathcal{R}^D.\texttt{Initialize}\left(\mathcal{L}, [-1/\rho_{\min}, 1/\rho_{\min}]\right)$
5: **for** $t \in [T]$ **do**
6:     Select strategy mixture $\boldsymbol{\xi}_t \leftarrow \mathcal{R}^P.\texttt{SelectDecision}()$
7:     Play strategy:

$$\boldsymbol{x}_t \leftarrow \begin{cases} \boldsymbol{x} \sim \boldsymbol{\xi}_t & \text{if } B_{i,t} \geq 1, \ \forall i \in [m] \\ \boldsymbol{x}^\varnothing & \text{otherwise} \end{cases}$$

8:     Observe the *feedback* as prescribed in Protocol 1
9:     Update budget availability $B_{i,t+1} \leftarrow B_{i,t} - c_t(\boldsymbol{x}_t)[i], \forall i \in [m]$
10:     Update dual variable $\boldsymbol{\lambda}_t \leftarrow \mathcal{R}^D.\texttt{SelectDecision}()$
11:     $\boxed{r_t^P : \Xi \ni \boldsymbol{\xi} \mapsto \mathbb{E}_{\boldsymbol{x} \sim \boldsymbol{\xi}}\left[f_t(\boldsymbol{x})\right] - \sum_{i \in [m]} \boldsymbol{\lambda}_t[i] \cdot \mathbb{E}_{\boldsymbol{x} \sim \boldsymbol{\xi}}\left[c_t(\boldsymbol{x})[i]\right]}$

12:     $\boxed{r_t^P(\boldsymbol{x}_t) \leftarrow f_t(\boldsymbol{x}_t) - \sum_{i \in [m]} \boldsymbol{\lambda}_t[i] \cdot c_t(\boldsymbol{x}_t)[i]}$

13:     $\mathcal{R}^P.\texttt{ReceiveFeedback}(r_t^P)$
14:     $\boxed{r_t^D : \mathcal{L} \ni \boldsymbol{\lambda} \mapsto -\sum_{i \in [m]} \boldsymbol{\lambda}[i] \cdot \left(B_t^{(i)} - \mathbb{E}_{\boldsymbol{x} \sim \boldsymbol{\xi}_t}\left[c_t(\boldsymbol{x})[i]\right]\right)}$

15:     $\boxed{r_t^D : \mathcal{L} \ni \boldsymbol{\lambda} \mapsto -\sum_{i \in [m]} \boldsymbol{\lambda}[i] \cdot \left(B_t^{(i)} - c_t(\boldsymbol{x}_t)[i]\right)}$

16:     $\mathcal{R}^D.\texttt{ReceiveFeedback}(r_t^D)$
17: **end for**

---

the full feedback case, the primal reward function—given for all possible strategy mixtures $\boldsymbol{\xi} \in \Xi$—is fed back into $\mathcal{R}^P$ (Line 11). In the bandit feedback case, the per round Lagrangian to be fed into the primal regret minimizer is computed as $r_t^P(\boldsymbol{x}_t) \leftarrow f_t(\boldsymbol{x}_t) - \sum_{i \in [m]} \boldsymbol{\lambda}_t[i] \cdot c_t(\boldsymbol{x}_t)[i]$ (Line 12 - 13). Since the complete reward and costs function are unknown with bandit feedback, it is not possible to build the Lagrangian for any strategy mixture $\boldsymbol{\xi} \in \Xi$. Finally, we underline that since the dual reward is built as $r_t^D : \mathcal{L} \ni \boldsymbol{\lambda} \mapsto -\sum_{i \in [m]} \boldsymbol{\lambda}[i] \cdot \left(B_t^{(i)} - \mathbb{E}_{\boldsymbol{x} \sim \boldsymbol{\xi}_t}\left[c_t(\boldsymbol{x})[i]\right]\right)$ for the full feedback case (Line 14) and as $r_t^D : \mathcal{L} \ni \boldsymbol{\lambda} \mapsto -\sum_{i \in [m]} \boldsymbol{\lambda}[i] \cdot \left(B_t^{(i)} - c_t(\boldsymbol{x}_t)[i]\right)$ in the bandit feedback case (Line 15), it is still sufficient to employ a dual regret minimizer tailored for full feedback in the bandit case.

## 4.1 Theoretical Results

In this section, we provide the theoretical guarantees attained by Algorithm 2. We mainly focus on the *full feedback* case; we show later that almost identical theoretical results can be obtained for *bandit feedback*. For additional lemmas and omitted proofs with *full feedback*, we refer to Appendix C.1. Similarly, for additional lemmas and omitted proofs with *bandit feedback*, we refer to Appendix D.1.

We start by lower bounding the expected reward attained by Algorithm 2.

**Lemma 4.1.** *For any $\delta \in (0,1)$, Algorithm 2 instantiated in the full feedback setting with a primal regret minimizer which attains a regret upper bound $R_T^P$ and a dual regret minimizer which attains a regret upper bound $R_T^D$, guarantees the following bound, with probability at least $1 - \delta$:*
$\sum_{t=1}^T \mathbb{E}_{\boldsymbol{x} \sim \boldsymbol{\xi}_t}\left[f_t(\boldsymbol{x})\right] \geq \text{OPT}_{\mathcal{H}} - (T - \tau) - (4 + 4\max_{\boldsymbol{\lambda} \in \mathcal{L}} \|\boldsymbol{\lambda}\|_1) \sqrt{2\tau \ln \frac{T}{\delta}} - \sum_{t=1}^\tau \sum_{i \in [m]} \boldsymbol{\lambda}[i] \cdot \left(B_t^{(i)} - \mathbb{E}_{\boldsymbol{x} \sim \boldsymbol{\xi}_t}\left[c_t(\boldsymbol{x})[i]\right]\right) - \frac{2}{\rho_{\min}} R_\tau^D - \left(1 + \frac{2}{\rho_{\min}}\right) R_\tau^P$, *where $\boldsymbol{\lambda} \in \mathcal{L}$ is an arbitrary Lagrange multiplier and $\tau$ is the stopping time of the algorithm.*

Lemma 4.1 shares many similarities with Lemma 3.1. Nonetheless, there are fundamental differences. First, the baseline is the *fixed* optimum $\text{OPT}_{\mathcal{H}}$. This is a consequence of the primal update performed

by Algorithm 2, which is no-regret with respect to fixed strategy mixtures only. Furthermore, in Lemma 4.1 the expected reward lower bounded has an additional term which depends on the theoretical guarantees of the primal regret minimizer with a multiplicative factor $(1 + 2/\rho_{\min})$ due to payoff range given to the primal algorithm.

We now present the final regret bound.

**Theorem 4.2.** *For any $\delta \in (0,1)$, Algorithm 2, when instantiated in the full feedback setting with a primal regret minimizer which attains a regret upper bound $R_T^P$ and a dual regret minimizer which attains a regret upper bound $R_T^D$, guarantees, with probability at least $1 - 2\delta$, the following regret bound $R_T \le 1 + \frac{1}{\rho_{\min}} + \frac{2}{\rho_{\min}} R_T^D + \left(1 + \frac{2}{\rho_{\min}}\right) R_T^P + \left(8 + \frac{8}{\rho_{\min}}\right) \sqrt{2T \ln \frac{T}{\delta}}.$*

Theorem 4.2 shows that learning with a spending plan is still possible when both the reward and the costs functions are not observed beforehand. The main difference with the online resource allocation setting is that we focus on the standard regret definition $R_T$–and not dynamic regret–and we pay an additional factor given by the primal no-regret guarantees. Finally notice that when $R_T^P \le \widetilde{O}(\sqrt{T})$ and $R_T^D \le \widetilde{O}(\sqrt{T})$, the cumulative regret is of order $R_T \le \widetilde{O}(1/\rho_{\min}\sqrt{T})$.

To extend the results above to the *bandit feedback* case, it is sufficient to notice that the optimal strategy mixture with respect to the Lagrangian is a pure strategy, that is:

$$\sup_{\boldsymbol{x} \in \mathcal{X}} \sum_{t=1}^{\tau} \left[ f_t(\boldsymbol{x}) + \sum_{i \in [m]} \boldsymbol{\lambda}_t[i] \cdot \left( B_t^{(i)} - c_t(\boldsymbol{x})[i] \right) \right]$$

$$= \sup_{\boldsymbol{\xi} \in \Xi} \sum_{t=1}^{\tau} \left[ \mathop{\mathbb{E}}_{\boldsymbol{x} \sim \boldsymbol{\xi}} [f_t(\boldsymbol{x})] + \sum_{i \in [m]} \boldsymbol{\lambda}_t[i] \cdot \left( B_t^{(i)} - \mathop{\mathbb{E}}_{\boldsymbol{x} \sim \boldsymbol{\xi}} [c_t(\boldsymbol{x})[i]] \right) \right].$$

The equation above allows us to relate the regret guarantees attained by the primal regret minimizer with bandit feedback to the ones attained by the primal regret minimizer in the *full feedback* setting and thus, obtaining almost identical results. Finally, we remark that algorithms designed for bandit feedback typically guarantee performance only with high probability. Consequently, the *bandit feedback* regret bound holds with probability at least $1 - (\delta + \delta_P)$, where $\delta_P$ is the confidence of algorithm $\mathcal{R}^P$.

**Remark 4.3** (Applications to multi-armed bandits with knapsacks). *Consider the bandits with knapsacks problem with $K$ arms (i.e. $\mathcal{X} = [K]$). We employ EXP-3 IX [45] as primal regret minimizer and online mirror descent with the negative entropy regularizer [46] as dual regret minimizer, and we assume that $\rho_{\min}$ is a constant independent of $T$. Then, for any $\delta \in (0,1)$, Algorithm 2 attains, with probability at least $1 - 2\delta$, $R_T \le O(\sqrt{KT \log(Tm/\delta)})$.*

## 5 Extensions

We provide extensions to our algorithms and analysis. We first show how to modify our algorithms in order to ensure sublinear regret when $\rho_{\min}$ is arbitrarily small. Then, we show performance guarantees for our algorithms against baselines which *do not* follow the spending plan.

### 5.1 Dealing with Arbitrarily Small $\rho_{\min}$

In this section we show how our algorithms may be modified to attain sublinear regret when $\rho_{\min}$ is arbitrarily small. Specifically, we show that a regret of order $\widetilde{O}(T^{3/4})$ is still attainable for all $\rho_{\min} \le \rho/T^{1/4}$. Due to space constraints, we will focus on the *online resource allocation* problem. Nevertheless, the same ideas apply to the OLRC setting. We refer to Appendix B.2, C.2, D.2 for the complete analysis, and the results for all three settings.

In Algorithm 3, we provide a meta-procedure which, by suitably modifying the input parameters given to Algorithm 1, achieves the desired theoretical guarantees. The meta-procedure is designed to be used when $\rho_{\min} \le \rho/T^{1/4}$—a condition that can be checked in advance—since in this regime, the dependence on $1/\rho_{\min}$ leads to suboptimal bounds compared to those we obtain below. Specifically,

---

**Algorithm 3** Meta-algorithm for arbitrarily small $\rho_{\min}$

---

**Require:** Horizon $T$, budget $B$, spending plans $\mathcal{B}_T^{(i)}$ for all $i \in [m]$, dual regret minimizer $\mathcal{R}^D$ (*full feedback*)

1: Define $\hat{\rho} \coloneqq \rho/T^{1/4}$, where $\rho \coloneqq B/T$
2: Define $\overline{B}_t^{(i)} \coloneqq B_t^{(i)} \left(1 - T^{-1/4}\right) \ \forall t \in [T], i \in [m]$
3: Run Algorithm 1 with $\rho_{\min} \leftarrow \hat{\rho}, B_t^{(i)} \leftarrow \overline{B}_t^{(i)}$

---

Algorithm 3 defines the capped minimum per-round budget $\hat{\rho} \coloneqq \rho/T^{1/4}$ (Line 1). This is done to prevent Algorithm 1 from instantiating the dual regret minimizer over a decision space $\mathcal{L}$ which scales as $1/\rho_{\min}$, leading to the same dependence in the regret bound. Then, the algorithm rescales the spending plan by a $(1 - T^{-1/4})$ factor (Line 2). Intuitively, this is necessary because the modified Lagrangian space $\mathcal{L} \coloneqq \left\{ \boldsymbol{\lambda} \in \mathbb{R}_{\geq 0}^m : \|\boldsymbol{\lambda}\|_1 \leq 1/\hat{\rho} \right\}$ is not large enough to compensate the spending plan violations. Thus, we force the algorithm to learn a harder constraint associated to the rescaled spending plan. Finally, Algorithm 1 is instantiated employing $\hat{\rho}$ in place of $\rho_{\min}$ and $\overline{B}_t^{(i)}$ in place of $B_t^{(i)}$, for all $t \in [T], i \in [m]$ (Line 3).

Employing Algorithm 3, it is possible to prove a similar result to Lemma 3.1. The key difference is that the payoff range of the dual and, in general, all the quantities which depend on the Lagrangian decision space, no longer scale as $O(1/\rho_{\min})$, but as $O(T^{1/4}/\rho)$. Moreover, while in the analysis of Algorithm 1 we compensate the $(T - \tau)$ term with $\sum_{t=1}^{\tau} \sum_{i \in [m]} \boldsymbol{\lambda}[i] \cdot (B_t^{(i)} - \mathbb{E}_{\boldsymbol{x} \sim \boldsymbol{\xi}_t} [c_t(\boldsymbol{x})[i]])$, in this case, we bound $(T - \tau)$ directly. This is done in the following lemma.

**Lemma 5.1.** *For any $\delta \in (0, 1)$, Algorithm 3 instantiated with a dual regret minimizer which attains a regret upper bound $R_T^D$, guarantees with probability at least $1 - \delta$, $T - \tau \leq \frac{14}{\rho} \left( \sqrt{\ln \frac{T}{\delta}} + \frac{R_T^D}{\sqrt{T}} \right) T^{\frac{3}{4}}$.*

Lemma 5.1 is proved by contradiction. Intuitively, we show that scaling the spending plan by a $(1 - T^{-1/4})$ factor results in the impossibility to have both $T - \tau > C T^{3/4}$, where $C$ is a constant to be chosen, and the optimality of the strategy $\boldsymbol{\xi}_t$ with respect to the Lagrangian $\mathfrak{L}_{f_t, c_t, \overline{B}_t}(\boldsymbol{\xi}, \boldsymbol{\lambda}_t)$, which is true by definition. We underline that the $R_T^D/\sqrt{T}$ factor is independent of $T$ for a dual regret minimizer which attains regret of order $O(\sqrt{T})$.

We are now ready to provide the final *dynamic* regret bound.

**Theorem 5.2.** *For any $\delta \in (0, 1)$, Algorithm 3, when instantiated with a dual regret minimizer which attains a regret upper bound $R_T^D$, guarantees, with probability at least $1 - 3\delta$, $\mathfrak{R}_T \leq \frac{14}{\rho} \left( \sqrt{\ln \frac{T}{\delta}} + \frac{R_T^D}{\sqrt{T}} \right) T^{\frac{3}{4}} + T^{\frac{3}{4}} + \left( 8 + \frac{4T^{\frac{1}{4}}}{\rho} \right) \sqrt{2T \ln \frac{T}{\delta}} + \frac{2T^{\frac{1}{4}}}{\rho} R_T^D$.*

Theorem 5.2 is proved employing Lemma 5.1 and bounding the loss in performance given by the fact the dynamic optimal baseline is not guaranteed to satisfy the scaled budgets $\overline{B}_t^{(i)}$, for all $t \in [T], i \in [m]$. Finally, notice that employing online mirror descent [46] as a dual regret minimizer, the dynamic regret guarantees of Algorithm 1 when compared with a baseline which follows the spending plans are of order $\mathfrak{R}_T \leq \widetilde{O}(T^{3/4})$, since $1/\rho$ is constant by definition.

As a final remark, we point out that if the number of rounds in which the per-round spending plan budget is smaller than $T^{-1/4}\rho$ is at most $\sqrt{T}$, then employing the meta-procedure defined by Algorithm 3 can be avoided. Instead, one can directly modify Algorithm 1 so that it plays the void action $\boldsymbol{x}^{\varnothing}$ in each of those rounds. Since the entire spending plan is known in advance, this preprocessing step can be performed before the start of the learning dynamic. This modification allows us to obtain the same theoretical guarantees of Section 3 (*i.e.*, when $\rho_{\min} > T^{-1/4}\rho$). Intuitively, since the number of such rounds is relatively small compared to the time horizon, their contribution to the regret remains negligible.

## 5.2 Robustness to Baselines Deviating from the Spending Plan

In this section, we study the performance of our algorithms compared to baselines which do not strictly follow the spending plan. Due to space constraints, we will focus on the ORA set-

ting. Nevertheless, the same ideas apply to both the OLRC case. We refer to the Appendix (B.1.1, B.2.1, C.1.1 C.2.3, D.1.1, D.2.3) for the analysis and the results for all three settings.

Suppose we give the baseline an error budget $\epsilon_t^{(i)} \geq 0$ for each $i \in [m], t \in [T]$, and allow the baseline to violate the per-round constraint by the given error. We define the *dynamic* optimal solution parametrized by the errors as the following optimization problem:

$$\text{OPT}_{\mathcal{D}}(\epsilon_t) \coloneqq \begin{cases} \sup_{\boldsymbol{\xi} \in \Xi^T} & \mathbb{E}_{\boldsymbol{x}_t \sim \boldsymbol{\xi}_t} \left[ \sum_{t=1}^T \bar{f}_t(\boldsymbol{x}_t) \right] \\ \text{s.t.} & \mathbb{E}_{\boldsymbol{x}_t \sim \boldsymbol{\xi}_t} \left[ \bar{c}_t(\boldsymbol{x}_t)[i] \right] \leq B_t^{(i)} + \epsilon_t^{(i)} \ \forall i \in [m], \forall t \in [T] \\ & \sum_{t=1}^T \mathbb{E}_{\boldsymbol{x}_t \sim \boldsymbol{\xi}_t} \left[ \bar{c}_t(\boldsymbol{x}_t)[i] \right] \leq B \ \forall i \in [m] \end{cases} \cdot \quad (3)$$

Problem (3) computes the expected value attained by the optimal dynamic strategy mixture that satisfies the spending plan at each round $t \in [T]$ up to the error term $\epsilon_t^{(i)}$, defined for all $i \in [m]$ and $t \in [T]$. The performance of our algorithms will smoothly degrade with the magnitude of these errors. We remark that the last group of constraints in Problem (3) ensures that the error terms do not allow the optimal solution to violate the aggregate budget constraints. Observe that when $\epsilon_t^{(i)} = 0$ for all $i \in [m]$ and $t \in [T]$—meaning that the spending plan is strictly followed by the optimal solution—the aggregate budget constraint is satisfied by the definition of the spending plan.

Similarly, we define the following notion of cumulative dynamic regret $\mathfrak{R}_T(\epsilon_t) \coloneqq \text{OPT}_{\mathcal{D}}(\epsilon_t) - \sum_{t=1}^T f_t(\boldsymbol{x}_t)$, which simply compares the rewards attained by the algorithm with respect to the optimal dynamic solution which follows the spending plan recommendations up to the errors.

We first provide the performance of Algorithm 1.

**Theorem 5.3.** *For any $\delta \in (0,1)$, Algorithm 1 instantiated with a dual regret minimizer which attains a regret upper bound $R_T^D$, guarantees, with probability at least $1 - 2\delta$, $\mathfrak{R}_T(\epsilon_t) \leq 1 + \frac{1}{\rho_{\min}} + \frac{2}{\rho_{\min}} R_T^D + \left( 8 + \frac{8}{\rho_{\min}} \right) \sqrt{2T \ln \frac{T}{\delta}} + \frac{1}{\rho_{\min}} \sum_{t=1}^T \sum_{i \in [m]} \epsilon_t^{(i)}$.*

Theorem 5.3 provides a similar regret bound to the one of Theorem 3.2. The main difference is that Algorithm 1 suffers an additional $\frac{1}{\rho_{\min}} \sum_{t=1}^T \sum_{i \in [m]} \epsilon_t^{(i)}$ term due to the error in the baseline definition and where the $1/\rho_{\min}$ factor follows from the Lagrangian space definition. As is easy to see, the regret bound in Theorem 5.3 remains sublinear as long as the sequence of errors is itself sublinear.

We conclude by showing the *dynamic* regret of Algorithm 3 for arbitrarily small $\rho_{\min}$.

**Theorem 5.4.** *For any $\delta \in (0,1)$, Algorithm 3, when instantiated with a dual regret minimizer which attains a regret upper bound $R_T^D$, guarantees, with probability at least $1 - 3\delta$, $\mathfrak{R}_T(\epsilon_t) \leq \frac{14}{\rho} \left( \sqrt{\ln \frac{T}{\delta}} + \frac{R_T^D}{\sqrt{T}} \right) T^{\frac{3}{4}} + T^{\frac{3}{4}} + \left( 8 + \frac{4T^{\frac{1}{4}}}{\rho} \right) \sqrt{2T \ln \frac{T}{\delta}} + \frac{2T^{\frac{1}{4}}}{\rho} R_T^D + \frac{T^{\frac{1}{4}}}{\rho} \sum_{t=1}^T \sum_{i \in [m]} \epsilon_t^{(i)}$.*

Theorem 5.4 shows that it is possible to be robust against a baseline which does not strictly follow the spending plan when $\rho_{\min}$ is arbitrarily small. In such a case, the algorithm pays an additional $\frac{T^{\frac{1}{4}}}{\rho} \cdot \sum_{t=1}^T \sum_{i \in [m]} \epsilon_t^{(i)}$ term, which follows from the definition of the Lagrangian space based on $\hat{\rho}$.

## Acknowledgments

This paper is supported by the Italian MIUR PRIN 2022 Project "Targeted Learning Dynamics: Computing Efficient and Fair Equilibria through No-Regret Algorithms", by the FAIR (Future Artificial Intelligence Research) project, funded by the NextGenerationEU program within the PNRR-PE-AI scheme (M4C2, Investment 1.3, Line on Artificial Intelligence), and by the EU Horizon project ELIAS (European Lighthouse of AI for Sustainability, No. 101120237). Christian Kroer is supported by the Office of Naval Research awards N00014-22-1-2530 and N00014-23-1-2374, and the National Science Foundation awards IIS-2147361 and IIS-2238960.

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

# Appendix

The appendix is structured as follows:

- In Appendix A, we provide further discussion on related works.
- In Appendix B, we provide additional lemmas and the omitted proof for the ORA setting.
- In Appendix C, we provide additional lemmas and the omitted proof for the OLRC setting under *full feedback*.
- In Appendix D, we provide additional lemmas and the omitted proof for the OLRC setting under *bandit feedback*.
- In Appendix E, we provide some technical lemmas which are necessary to prove the main results of our work.

# A   Related Works

In the following, we highlight the works that are mainly related to ours. Specifically, we will focus on the ORA literature, on the OLRC (in particular, bandits with knapsacks) one and, finally, on the learning with general constraints literature.

**Online Resource Allocation**   Early works on online allocation mostly focus on settings where the reward and costs/resource functions are linear in the decision variables, especially in the *random permutation model*. In this setup, an adversary chooses a fixed list of requests, which are then shown in a random order. Devanur and Hayes [21] studied the *AdWords* problem and introduced a two-step method using dual variables. Their method has regret of order $O(T^{2/3})$. Feldman et al. [27] proposed similar training-based methods for a wider range of linear allocation problems, attaining comparable regret bounds. Later, Agrawal et al. [5], Devanur et al. [22], and Kesselheim et al. [37] improved these ideas by designing algorithms that update decisions over time by solving linear programs repeatedly using accumulated data. These methods brought the regret down to $O(T^{1/2})$ and scale better with more resources. Gupta and Molinaro [30] extended this type of approach to the random permutation model. Agrawal and Devanur [2] suggested a similar algorithm that keeps and updates dual variables, but it either needs prior knowledge of a benchmark value or must solve extra optimization problems to estimate it. Balseiro et al. [11] proposed a dual mirror descent algorithm for problems with concave rewards and stochastic rewards/costs, getting $O(T^{1/2})$ regret. Sun et al. [56] showed an algorithm that gets better than $O(T^{1/2})$ regret when the request distribution is known. Kanoria and Qian [35] used online dual mirror descent in managing circulating resources in closed systems, such as those used in ride-hailing platforms. Differently, in the adversarial online allocation problem, sublinear regret is not possible, so the goal becomes designing algorithms with constant-factor guarantees compared to the offline optimum. Mehta et al. [43] and Buchbinder et al. [18] studied the *AdWords* problem—a special case of online matching where rewards are proportional to resource usage. They proposed primal-dual algorithms that get a $(1 - 1/e)$ competitive ratio, which is known to be the best possible. But when rewards are not proportional to the resource consumption, a fixed competitive ratio cannot be ensured. To handle this, Feldman et al. [26] suggested a version with free disposal, where resource limits can be exceeded, and only the highest-reward allocations count in the objective. Their method also achieves a $(1 - 1/e)$ competitive ratio. Gaitonde et al. [28] study bidding algorithms with aggregate guarantees in terms of overall market efficiency, without relying on convergence. Balseiro et al. [14] develop the first best-of-many-worlds algorithm for the online allocation problem. Their algorithm simultaneously handle stochastic, adversarial and non-stationary rewards/costs. In a recent work, Balseiro et al. [13] study the online allocation problem where a sample from each adversarially changing distributions $\mathcal{F}_t, \mathcal{C}_t$ is known beforehand. To show that, in such a setting, it is possible to achieve sublinear regret–this is possible when the samples are slightly corrupted, too–, the authors develop a dual algorithm which works with an estimated spending plan. When this spending plan is correct, their algorithm shares different similarities with Algorithm 1. Nonetheless, their analysis relies on the assumption that there exists a $\kappa \in \mathbb{R}_{\geq 0}$ s.t. $f_t(\boldsymbol{x}) \leq \kappa c_t(\boldsymbol{x})$[9] for all $\boldsymbol{x} \in \mathcal{X}, t \in [T]$, and their regret bound scales as $\widetilde{O}(\kappa\sqrt{T})$. This is not the case in our work, where we study the standard online allocation problem as presented in Balseiro et al. [14].

---

[9]Balseiro et al. [13] focus on online allocation with a single resource.

**Online Learning with Resource Constraints**  The stochastic bandits with knapsacks framework was originally introduced and optimally solved by Badanidiyuru et al. [7, 9]. Subsequently, other algorithms achieving optimal regret in the stochastic setting were developed by Agrawal and Devanur [3, 4] and Immorlica et al. [31]. Over time, the bandits with knapsack framework has been extended to a variety of scenarios, including more general types of constraints [3, 4], contextual bandits [24, 8, 1, 6, 50], and combinatorial semi-bandits [48]. The adversarial version of bandits with knapsacks was introduced by Immorlica et al. [31, 32], who showed that it is possible to achieve a $O(m \log T)$ competitive ratio when an oblivious adversary selects the rewards and costs. Immorlica et al. [31, 32] also establish a matching lower bound, proving that no algorithm can achieve better than a $O(\log T)$ competitive ratio on all instances, even with just two arms and one resource. This lower bound was further tightened by Kesselheim and Singla [36], who show an $O(\log m \log T)$ competitive ratio, and demonstrate that it is optimal up to constant factors for the general adversarial bandits with knapsacks setting. Castiglioni et al. [19] propose the first best-of-both-worlds algorithm for bandits with knapsacks. The authors propose a primal-dual algorithm which simultaneously handle stochastic and adversarial rewards/cost function. They propose an algorithm for the *full feedback* setting and one for the *bandit feedback* case. In a contemporaneous work, Braverman et al. [17] show how it is possible to overcome the impossibility result for adversarial bandits with knapsacks for specific benchmarks, which, intuitively, are not too far from the solution who spends the budget uniformly over the rounds.

**Learning with General Constraints**  There exists an extended literature on online learning problem with general constraints (*e.g.*, [42, 39]). Two main settings are usually studied. In the soft constraints setting (*e.g.*, [20]), the aim is to guarantee that the constraint violations incurred by the algorithm grow sub-linearly. In the hard constraints setting, the algorithms must satisfy the constraints at every round, by assuming knowledge of a strictly feasible decision (*e.g.*, [47]). Both soft and hard constraints have been generalized to settings that are more challenging than multi-armed bandits, such as linear bandits (*e.g.*, [29]). We acknowledge that online learning with constraints has been studies even in multi-state environment such as constrained Markov decision processes (CMDPs) [58, 25, 60, 10, 16, 40, 23, 51, 53, 44, 52, 54, 55]. Some works focus on constrained online convex optimization settings (*e.g.*, [41, 33, 59, 49]).

# B  Omitted Proofs for Online Resource Allocation

In this section, we provide the results and the omitted proofs for the *online resource allocation setting*.

## B.1  Theoretical Guarantees of Algorithm 1

We start by providing the results for Algorithm 1. Specifically, we lower bound the expected rewards attained by the algorithm as follows.

**Lemma B.1.** *Algorithm 1, when instantiated with a dual regret minimizer which attains a regret upper bound $R_T^D$, guarantees the following bound:*

$$\sum_{t=1}^{\tau} \mathop{\mathbb{E}}_{\boldsymbol{x} \sim \boldsymbol{\xi}_t} [f_t(\boldsymbol{x})] \geq \sum_{t=1}^{\tau} \left( \mathop{\mathbb{E}}_{\boldsymbol{x} \sim \boldsymbol{\xi}^{(t)}} [f_t(\boldsymbol{x})] - \sum_{i \in [m]} \boldsymbol{\lambda}_t[i] \cdot \left( B_t^{(i)} - \mathop{\mathbb{E}}_{\boldsymbol{x} \sim \boldsymbol{\xi}^{(t)}} [c_t(\boldsymbol{x})[i]] \right) \right)$$
$$- \sum_{t=1}^{\tau} \sum_{i \in [m]} \boldsymbol{\lambda}[i] \cdot \left( B_t^{(i)} - \mathop{\mathbb{E}}_{\boldsymbol{x} \sim \boldsymbol{\xi}_t} [c_t(\boldsymbol{x})[i]] \right) - \frac{2}{\rho_{\min}} R_\tau^D.$$

*where $\boldsymbol{\lambda} \in \mathcal{L}$ is an arbitrary Lagrange multiplier, $\{\boldsymbol{\xi}^{(t)} \in \Xi\}_{t=1}^{\tau}$ is an arbitrary sequence of strategy mixtures and $\tau$ is the stopping time of the algorithm.*

*Proof.*  In the following, we aim at bounding the rewards attained by Algorithm 1 during the learning dynamic, that is, until the round $\tau$ the algorithm has depleted its budget.

First, by the strategy mixture selection criterion, it holds:

$$\mathop{\mathbb{E}}_{\boldsymbol{x} \sim \boldsymbol{\xi}} [f_t(\boldsymbol{x})] - \sum_{i \in [m]} \boldsymbol{\lambda}_t[i] \cdot \mathop{\mathbb{E}}_{\boldsymbol{x} \sim \boldsymbol{\xi}} [c_t(\boldsymbol{x})[i]] \leq \mathop{\mathbb{E}}_{\boldsymbol{x} \sim \boldsymbol{\xi}_t} [f_t(\boldsymbol{x})] - \sum_{i \in [m]} \boldsymbol{\lambda}_t[i] \cdot \mathop{\mathbb{E}}_{\boldsymbol{x} \sim \boldsymbol{\xi}_t} [c_t(\boldsymbol{x})[i]], \quad (4)$$

for all $\boldsymbol{\xi} \in \Xi$ and for all $t \in [\tau]$.

Summing Equation (4) over $t$ we obtain, for all sequences $\{\boldsymbol{\xi}^{(t)} \in \Xi\}_{t=1}^{\tau}$, the following bound:

$$\sum_{t=1}^{\tau} \left( \mathop{\mathbb{E}}_{\boldsymbol{x} \sim \boldsymbol{\xi}^{(t)}} [f_t(\boldsymbol{x})] - \sum_{i \in [m]} \boldsymbol{\lambda}_t[i] \cdot \mathop{\mathbb{E}}_{\boldsymbol{x} \sim \boldsymbol{\xi}^{(t)}} [c_t(\boldsymbol{x})[i]] \right)$$

$$\leq \sum_{t=1}^{\tau} \left( \mathop{\mathbb{E}}_{\boldsymbol{x} \sim \boldsymbol{\xi}_t} [f_t(\boldsymbol{x})] - \sum_{i \in [m]} \boldsymbol{\lambda}_t[i] \cdot \mathop{\mathbb{E}}_{\boldsymbol{x} \sim \boldsymbol{\xi}_t} [c_t(\boldsymbol{x})[i]] \right),$$

which in turn implies:

$$\sum_{t=1}^{\tau} \mathop{\mathbb{E}}_{\boldsymbol{x} \sim \boldsymbol{\xi}_t} [f_t(\boldsymbol{x})] \geq \sum_{t=1}^{\tau} \left( \mathop{\mathbb{E}}_{\boldsymbol{x} \sim \boldsymbol{\xi}^{(t)}} [f_t(\boldsymbol{x})] - \sum_{i \in [m]} \boldsymbol{\lambda}_t[i] \cdot \mathop{\mathbb{E}}_{\boldsymbol{x} \sim \boldsymbol{\xi}^{(t)}} [c_t(\boldsymbol{x})[i]] + \sum_{i \in [m]} \boldsymbol{\lambda}_t[i] \cdot \mathop{\mathbb{E}}_{\boldsymbol{x} \sim \boldsymbol{\xi}_t} [c_t(\boldsymbol{x})[i]] \right).$$

Employing the dual regret minimizer guarantees, we have, for any Lagrange variable $\boldsymbol{\lambda} \in \mathcal{L}$:

$$\sum_{t=1}^{\tau} \sum_{i \in [m]} \boldsymbol{\lambda}_t[i] \cdot \left( B_t^{(i)} - \mathop{\mathbb{E}}_{\boldsymbol{x} \sim \boldsymbol{\xi}_t} [c_t(\boldsymbol{x})[i]] \right) - \sum_{t=1}^{\tau} \sum_{i \in [m]} \boldsymbol{\lambda}[i] \cdot \left( B_t^{(i)} - \mathop{\mathbb{E}}_{\boldsymbol{x} \sim \boldsymbol{\xi}_t} [c_t(\boldsymbol{x})[i]] \right) \leq \frac{2}{\rho_{\min}} R_\tau^D,$$

(5)

where the $\frac{2}{\rho_{\min}}$ dependence follows from the payoffs' range of the dual regret minimizer.

Thus, substituting Equation (5) in the previous bound, we get:

$$\sum_{t=1}^{\tau} \mathop{\mathbb{E}}_{\boldsymbol{x} \sim \boldsymbol{\xi}_t} [f_t(\boldsymbol{x})] \geq \sum_{t=1}^{\tau} \left( \mathop{\mathbb{E}}_{\boldsymbol{x} \sim \boldsymbol{\xi}^{(t)}} [f_t(\boldsymbol{x})] - \sum_{i \in [m]} \boldsymbol{\lambda}_t[i] \cdot \left( B_t^{(i)} - \mathop{\mathbb{E}}_{\boldsymbol{x} \sim \boldsymbol{\xi}^{(t)}} [c_t(\boldsymbol{x})[i]] \right) \right)$$

$$- \sum_{t=1}^{\tau} \sum_{i \in [m]} \boldsymbol{\lambda}[i] \cdot \left( B_t^{(i)} - \mathop{\mathbb{E}}_{\boldsymbol{x} \sim \boldsymbol{\xi}_t} [c_t(\boldsymbol{x})[i]] \right) - \frac{2}{\rho_{\min}} R_\tau^D.$$

This concludes the proof. $\qquad\square$

We proceed by relating the previous lower bound to the *dynamic* optimum.

**Lemma 3.1.** *For any $\delta \in (0, 1)$, Algorithm 1, when instantiated with a dual regret minimizer which attains a regret upper bound $R_T^D$, guarantees, with probability at least $1 - \delta$, $\sum_{t=1}^{T} \mathbb{E}_{\boldsymbol{x} \sim \boldsymbol{\xi}_t} [f_t(\boldsymbol{x})] \geq$ $\mathrm{OPT}_{\mathcal{D}} - (T - \tau) - (4 + 4 \max_{\boldsymbol{\lambda} \in \mathcal{L}} \|\boldsymbol{\lambda}\|_1) \sqrt{2\tau \ln \frac{T}{\delta}} - \sum_{t=1}^{\tau} \sum_{i \in [m]} \boldsymbol{\lambda}[i] \cdot (B_t^{(i)} - \mathbb{E}_{\boldsymbol{x} \sim \boldsymbol{\xi}_t} [c_t(\boldsymbol{x})[i]]) - \frac{2}{\rho_{\min}} R_\tau^D$, where $\boldsymbol{\lambda} \in \mathcal{L}$ is an arbitrary Lagrange multiplier and $\tau$ is the stopping time of the algorithm.*

*Proof.* We first employ Lemma B.1 to obtain, for all strategy mixtures sequences $\{\boldsymbol{\xi}^{(t)} \in \Xi\}_{t=1}^{\tau}$ and for all Lagrangian variables $\boldsymbol{\lambda} \in \mathcal{L}$, the following lower bound:

$$\sum_{t=1}^{\tau} \mathop{\mathbb{E}}_{\boldsymbol{x} \sim \boldsymbol{\xi}_t} [f_t(\boldsymbol{x})] \geq \sum_{t=1}^{\tau} \left( \mathop{\mathbb{E}}_{\boldsymbol{x} \sim \boldsymbol{\xi}^{(t)}} [f_t(\boldsymbol{x})] - \sum_{i \in [m]} \boldsymbol{\lambda}_t[i] \cdot \left( B_t^{(i)} - \mathop{\mathbb{E}}_{\boldsymbol{x} \sim \boldsymbol{\xi}^{(t)}} [c_t(\boldsymbol{x})[i]] \right) \right)$$

$$- \sum_{t=1}^{\tau} \sum_{i \in [m]} \boldsymbol{\lambda}[i] \cdot \left( B_t^{(i)} - \mathop{\mathbb{E}}_{\boldsymbol{x} \sim \boldsymbol{\xi}_t} [c_t(\boldsymbol{x})[i]] \right) - \frac{2}{\rho_{\min}} R_\tau^D,$$

We now focus on lower bounding the following term:

$$\sum_{t=1}^{\tau} \left( \mathop{\mathbb{E}}_{\boldsymbol{x} \sim \boldsymbol{\xi}^{(t)}} [f_t(\boldsymbol{x})] - \sum_{i \in [m]} \boldsymbol{\lambda}_t[i] \cdot \left( B_t^{(i)} - \mathop{\mathbb{E}}_{\boldsymbol{x} \sim \boldsymbol{\xi}^{(t)}} [c_t(\boldsymbol{x})[i]] \right) \right)$$

Thus, notice that by the definition of Program (1), there exists a sequence strategy mixture $\{\boldsymbol{\xi}_t^*\}_{t=1}^{T}$ such that $\mathbb{E}_{\boldsymbol{x} \sim \boldsymbol{\xi}_t^*} [\bar{c}_t(\boldsymbol{x})] \leq B_t^{(i)}$ for all $t \in [T], i \in [m]$ and $\sum_{t=1}^{T} \mathbb{E}_{\boldsymbol{x} \sim \boldsymbol{\xi}_t^*} [\bar{f}_t(\boldsymbol{x})] \geq \mathrm{OPT}_{\mathcal{D}} - \gamma$, for

all $\gamma > 0$. In the rest of the proof, we will omit the the dependence on $\gamma$, since it can be chosen arbitrarily small, thus being negligible in the regret bound.

Selecting $\{\boldsymbol{\xi}^{(t)}\}_{t=1}^{\tau} = \{\boldsymbol{\xi}_t^*\}_{t=1}^{\tau}$ and employing Lemma E.2, the quantity of interest is lower bounded as:

$$\sum_{t=1}^{\tau} \left( \mathop{\mathbb{E}}_{\boldsymbol{x} \sim \boldsymbol{\xi}_t^*} [f_t(\boldsymbol{x})] - \sum_{i \in [m]} \boldsymbol{\lambda}_t[i] \cdot \left( B_t^{(i)} - \mathop{\mathbb{E}}_{\boldsymbol{x} \sim \boldsymbol{\xi}_t^*} [c_t(\boldsymbol{x})[i]] \right) \right)$$

$$\geq \sum_{t=1}^{\tau} \left( \mathop{\mathbb{E}}_{\boldsymbol{x} \sim \boldsymbol{\xi}_t^*} [\bar{f}_t(\boldsymbol{x})] - \sum_{i \in [m]} \boldsymbol{\lambda}_t[i] \cdot \left( B_t^{(i)} - \mathop{\mathbb{E}}_{\boldsymbol{x} \sim \boldsymbol{\xi}_t^*} [\bar{c}_t(\boldsymbol{x})[i]] \right) \right)$$

$$- \left( 4 + 4 \max_{\boldsymbol{\lambda} \in \mathcal{L}} \|\boldsymbol{\lambda}\|_1 \right) \sqrt{2\tau \ln \frac{T}{\delta}},$$

which holds with probability at least $1 - \delta$. Moreover, by the baseline definition, it holds:

$$\sum_{t=1}^{\tau} \sum_{i \in [m]} \boldsymbol{\lambda}_t[i] \cdot \left( B_t^{(i)} - \mathop{\mathbb{E}}_{\boldsymbol{x} \sim \boldsymbol{\xi}_t^*} [\bar{c}_t(\boldsymbol{x})[i]] \right) \geq 0,$$

since the *dynamic* optimum satisfies the learning plan at each round.

Combining everything, we get, with probability at least $1 - \delta$, the following lower bound:

$$\sum_{t=1}^{\tau} \left( \mathop{\mathbb{E}}_{\boldsymbol{x}_t \sim \boldsymbol{\xi}_t^*} [f_t(\boldsymbol{x})] - \sum_{i \in [m]} \boldsymbol{\lambda}_t[i] \cdot \left( B_t^{(i)} - \mathop{\mathbb{E}}_{\boldsymbol{x} \sim \boldsymbol{\xi}_t^*} [c_t(\boldsymbol{x})[i]] \right) \right)$$

$$\geq \sum_{t=1}^{\tau} \left( \mathop{\mathbb{E}}_{\boldsymbol{x} \sim \boldsymbol{\xi}_t^*} [\bar{f}_t(\boldsymbol{x})] - \sum_{i \in [m]} \boldsymbol{\lambda}_t[i] \cdot \left( B_t^{(i)} - \mathop{\mathbb{E}}_{\boldsymbol{x} \sim \boldsymbol{\xi}_t^*} [\bar{c}_t(\boldsymbol{x})[i]] \right) \right)$$

$$- \left( 4 + 4 \max_{\boldsymbol{\lambda} \in \mathcal{L}} \|\boldsymbol{\lambda}\|_1 \right) \sqrt{2\tau \ln \frac{T}{\delta}}$$

$$\geq \sum_{t=1}^{\tau} \mathop{\mathbb{E}}_{\boldsymbol{x} \sim \boldsymbol{\xi}_t^*} [\bar{f}_t(\boldsymbol{x})] - \left( 4 + 4 \max_{\boldsymbol{\lambda} \in \mathcal{L}} \|\boldsymbol{\lambda}\|_1 \right) \sqrt{2\tau \ln \frac{T}{\delta}}$$

$$\geq \mathrm{OPT}_{\mathcal{D}} - (T - \tau) - \left( 4 + 4 \max_{\boldsymbol{\lambda} \in \mathcal{L}} \|\boldsymbol{\lambda}\|_1 \right) \sqrt{2\tau \ln \frac{T}{\delta}}.$$

Noticing that by the update of Algorithm 1, it holds:

$$\sum_{t=1}^{\tau} \mathop{\mathbb{E}}_{\boldsymbol{x} \sim \boldsymbol{\xi}_t} [f_t(\boldsymbol{x})] = \sum_{t=1}^{T} \mathop{\mathbb{E}}_{\boldsymbol{x} \sim \boldsymbol{\xi}_t} [f_t(\boldsymbol{x})],$$

which concludes the proof. $\qquad\square$

We are now ready to prove the final regret bound for Algorithm 1.

**Theorem 3.2.** *For any $\delta \in (0, 1)$, Algorithm 1 instantiated with a dual regret minimizer which attains a regret upper bound $R_T^D$, guarantees $\mathfrak{R}_T \leq 1 + \frac{1}{\rho_{\min}} + \frac{2}{\rho_{\min}} R_T^D + \left( 8 + \frac{8}{\rho_{\min}} \right) \sqrt{2T \ln \frac{T}{\delta}}$, with probability at least $1 - 2\delta$.*

*Proof.* We start by employing Lemma 3.1 to get, with probability at least $1 - \delta$, the following bound:

$$\sum_{t=1}^{T} \mathop{\mathbb{E}}_{\boldsymbol{x} \sim \boldsymbol{\xi}_t} [f_t(\boldsymbol{x})] \geq \mathrm{OPT}_{\mathcal{D}} - (T - \tau) - \left( 4 + 4 \max_{\boldsymbol{\lambda} \in \mathcal{L}} \|\boldsymbol{\lambda}\|_1 \right) \sqrt{2\tau \ln \frac{T}{\delta}}$$

$$- \sum_{t=1}^{\tau} \sum_{i \in [m]} \boldsymbol{\lambda}[i] \cdot \left( B_t^{(i)} - \mathop{\mathbb{E}}_{\boldsymbol{x} \sim \boldsymbol{\xi}_t} [c_t(\boldsymbol{x})[i]] \right) - \frac{2}{\rho_{\min}} R_\tau^D,$$

Thus, we employ Lemma E.1, to recover the following result, which holds with probability at least $1 - 2\delta$, by Union Bound:

$$\sum_{t=1}^{T} f_t(\boldsymbol{x}_t) \geq \text{OPT}_{\mathcal{D}} - (T - \tau) - \left(4 + 4 \max_{\boldsymbol{\lambda} \in \mathcal{L}} \|\boldsymbol{\lambda}\|_1\right) \sqrt{2\tau \ln \frac{T}{\delta}}$$
$$- \sum_{t=1}^{\tau} \sum_{i \in [m]} \boldsymbol{\lambda}[i] \cdot \left(B_t^{(i)} - c_t(\boldsymbol{x}_t)[i]\right) - \frac{2}{\rho_{\min}} R_\tau^D - (4 + 4\|\boldsymbol{\lambda}\|_1) \sqrt{2\tau \ln \frac{T}{\delta}},$$

which in turn implies the following *dynamic* regret bound:

$$\mathfrak{R}_T \leq (T - \tau) + \sum_{t=1}^{\tau} \sum_{i \in [m]} \boldsymbol{\lambda}[i] \cdot \left(B_t^{(i)} - c_t(\boldsymbol{x}_t)[i]\right) + \frac{2}{\rho_{\min}} R_\tau^D + \left(8 + 8 \max_{\boldsymbol{\lambda} \in \mathcal{L}} \|\boldsymbol{\lambda}\|_1\right) \sqrt{2\tau \ln \frac{T}{\delta}}.$$

which holds with probability at least $1 - 2\delta$. We now split the analysis in two cases. Specifically, in case $(i)$, it holds $T = \tau$, namely, the algorithm has not depleted the budget during the learning dynamic, while in case $(ii)$, it holds $T > \tau$.

**Bound for case $(i)$.** When $T = \tau$, we choose $\boldsymbol{\lambda} = \boldsymbol{0}$ to obtain, with probability at least $1 - 2\delta$:

$$\mathfrak{R}_T \leq (T - \tau) + \frac{2}{\rho_{\min}} R_\tau^D + \left(8 + 8 \max_{\boldsymbol{\lambda} \in \mathcal{L}} \|\boldsymbol{\lambda}\|_1\right) \sqrt{2\tau \ln \frac{T}{\delta}}.$$

**Bound for case $(ii)$.** When $T > \tau$, we notice that, due the budget constraints, there exists a resource $i^* \in [m]$ such that the following holds:

$$\sum_{t=1}^{\tau} c_t(\boldsymbol{x}_t)[i^*] + 1 \geq B \geq \sum_{t=1}^{T} B_t^{(i^*)}.$$

Thus, the *dynamic* regret can be bounded with probability at least $1 - 2\delta$ as follows:

$$\mathfrak{R}_T \leq (T - \tau) + \left(8 + \frac{8}{\rho_{\min}}\right) \sqrt{2\tau \ln \frac{T}{\delta}} + \sum_{t=1}^{\tau} \sum_{i \in [m]} \boldsymbol{\lambda}[i] \cdot \left(B_t^{(i)} - c_t(\boldsymbol{x}_t)[i]\right) + \frac{2}{\rho_{\min}} R_\tau^D$$

$$\leq (T - \tau) + \left(8 + \frac{8}{\rho_{\min}}\right) \sqrt{2\tau \ln \frac{T}{\delta}} + \frac{1}{\rho_{\min}} \left(\sum_{t=1}^{\tau} B_t^{(i)} - \sum_{t=1}^{T} B_t^{(i^*)} + 1\right) + \frac{2}{\rho_{\min}} R_\tau^D$$

$$= (T - \tau) + \left(8 + \frac{8}{\rho_{\min}}\right) \sqrt{2\tau \ln \frac{T}{\delta}} - \frac{1}{\rho_{\min}} \left(\sum_{t=\tau+1}^{T} B_t^{(i^*)} + 1\right) + \frac{2}{\rho_{\min}} R_\tau^D$$

$$\leq (T - \tau) + \left(8 + \frac{8}{\rho_{\min}}\right) \sqrt{2\tau \ln \frac{T}{\delta}} - \frac{1}{\rho_{\min}} \left(\sum_{t=\tau+1}^{T} \rho_{\min} + 1\right) + \frac{2}{\rho_{\min}} R_\tau^D$$

$$\leq (T - \tau) + \left(8 + \frac{8}{\rho_{\min}}\right) \sqrt{2\tau \ln \frac{T}{\delta}} - (T - \tau) + 1 + \frac{1}{\rho_{\min}} + \frac{2}{\rho_{\min}} R_\tau^D$$

$$\leq 1 + \frac{1}{\rho_{\min}} + \left(8 + \frac{8}{\rho_{\min}}\right) \sqrt{2\tau \ln \frac{T}{\delta}} + \frac{2}{\rho_{\min}} R_\tau^D,$$

where the first inequality holds since $\max_{\boldsymbol{\lambda} \in \mathcal{L}} \|\boldsymbol{\lambda}\|_1 \leq \frac{1}{\rho_{\min}}$ and the second inequality holds by selecting $\boldsymbol{\lambda}$ such that $\boldsymbol{\lambda}[i^*] = \frac{1}{\rho_{\min}}$ and $\boldsymbol{\lambda}[i] = 0$ for all others $i \in [m]$.

Finally, we notice the following trivial bounds:

$$R_\tau^D \leq R_T^D, \quad \left(8 + \frac{8}{\rho_{\min}}\right) \sqrt{2\tau \ln \frac{T}{\delta}} \leq \left(8 + \frac{8}{\rho_{\min}}\right) \sqrt{2T \ln \frac{T}{\delta}}.$$

This concludes the proof. $\qquad\square$

### B.1.1 Robustness to Baselines Deviating from the Spending Plan

We provide the regret of Algorithm 1 with respect to a baseline which deviates from the spending plan.

**Theorem 5.3.** *For any $\delta \in (0,1)$, Algorithm 1 instantiated with a dual regret minimizer which attains a regret upper bound $R_T^D$, guarantees, with probability at least $1 - 2\delta$, $\mathfrak{R}_T(\epsilon_t) \leq 1 + \frac{1}{\rho_{\min}} + \frac{2}{\rho_{\min}} R_T^D + \left(8 + \frac{8}{\rho_{\min}}\right) \sqrt{2T \ln \frac{T}{\delta}} + \frac{1}{\rho_{\min}} \sum_{t=1}^T \sum_{i \in [m]} \epsilon_t^{(i)}$.*

*Proof.* Similarly to Lemma 3.1, we employ Lemma B.1 to obtain, for all strategy mixtures sequences $\{\boldsymbol{\xi}^{(t)} \in \Xi\}_{t=1}^\tau$ and for all Lagrangian variables $\boldsymbol{\lambda} \in \mathcal{L}$, the following lower bound:

$$
\sum_{t=1}^\tau \mathop{\mathbb{E}}_{\boldsymbol{x} \sim \boldsymbol{\xi}_t} [f_t(\boldsymbol{x})] \geq \sum_{t=1}^\tau \left( \mathop{\mathbb{E}}_{\boldsymbol{x} \sim \boldsymbol{\xi}^{(t)}} [f_t(\boldsymbol{x})] - \sum_{i \in [m]} \boldsymbol{\lambda}_t[i] \cdot \left( B_t^{(i)} - \mathop{\mathbb{E}}_{\boldsymbol{x} \sim \boldsymbol{\xi}^{(t)}} [c_t(\boldsymbol{x})[i]] \right) \right)
$$
$$
- \sum_{t=1}^\tau \sum_{i \in [m]} \boldsymbol{\lambda}[i] \cdot \left( B_t^{(i)} - \mathop{\mathbb{E}}_{\boldsymbol{x} \sim \boldsymbol{\xi}_t} [c_t(\boldsymbol{x})[i]] \right) - \frac{2}{\rho_{\min}} R_\tau^D,
$$

The key difference with respect to the analysis of Lemma 3.1 is how to bound the following term:

$$
\sum_{t=1}^\tau \left( \mathop{\mathbb{E}}_{\boldsymbol{x} \sim \boldsymbol{\xi}^{(t)}} [f_t(\boldsymbol{x})] - \sum_{i \in [m]} \boldsymbol{\lambda}_t[i] \cdot \left( B_t^{(i)} - \mathop{\mathbb{E}}_{\boldsymbol{x} \sim \boldsymbol{\xi}^{(t)}} [c_t(\boldsymbol{x})[i]] \right) \right)
$$

Thus, notice that by the definition of Program (3), there exists a sequence strategy mixture $\{\boldsymbol{\xi}_t^*\}_{t=1}^T$ such that $\mathbb{E}_{\boldsymbol{x} \sim \boldsymbol{\xi}_t^*} [\bar{c}_t(\boldsymbol{x})] \leq B_t^{(i)} - \epsilon_t^{(i)}$ for all $t \in [T], i \in [m]$ and $\sum_{t=1}^T \mathbb{E}_{\boldsymbol{x} \sim \boldsymbol{\xi}_t^*} [\bar{f}_t(\boldsymbol{x})] \geq \text{OPT}_\mathcal{D}(\epsilon_t) - \gamma$, for all $\gamma > 0$. In the rest of the proof, we will omit the the dependence on $\gamma$, since it can be chosen arbitrarily small, thus being negligible in the regret bound.

Selecting $\{\boldsymbol{\xi}^{(t)}\}_{t=1}^\tau = \{\boldsymbol{\xi}_t^*\}_{t=1}^\tau$ and employing Lemma E.2, the quantity of interest is lower bounded as:

$$
\sum_{t=1}^\tau \left( \mathop{\mathbb{E}}_{\boldsymbol{x} \sim \boldsymbol{\xi}_t^*} [f_t(\boldsymbol{x})] - \sum_{i \in [m]} \boldsymbol{\lambda}_t[i] \cdot \left( B_t^{(i)} - \mathop{\mathbb{E}}_{\boldsymbol{x} \sim \boldsymbol{\xi}_t^*} [c_t(\boldsymbol{x})[i]] \right) \right)
$$
$$
\geq \sum_{t=1}^\tau \left( \mathop{\mathbb{E}}_{\boldsymbol{x} \sim \boldsymbol{\xi}_t^*} [\bar{f}_t(\boldsymbol{x})] - \sum_{i \in [m]} \boldsymbol{\lambda}_t[i] \cdot \left( B_t^{(i)} - \mathop{\mathbb{E}}_{\boldsymbol{x} \sim \boldsymbol{\xi}_t^*} [\bar{c}_t(\boldsymbol{x})[i]] \right) \right)
$$
$$
- \left( 4 + 4 \max_{\boldsymbol{\lambda} \in \mathcal{L}} \|\boldsymbol{\lambda}\|_1 \right) \sqrt{2\tau \ln \frac{T}{\delta}},
$$

which holds with probability at least $1 - \delta$. Moreover, by the baseline definition, it holds:

$$
\sum_{t=1}^\tau \sum_{i \in [m]} \boldsymbol{\lambda}_t[i] \cdot \left( B_t^{(i)} - \mathop{\mathbb{E}}_{\boldsymbol{x} \sim \boldsymbol{\xi}_t^*} [\bar{c}_t(\boldsymbol{x})[i]] \right) \geq -\frac{1}{\rho_{\min}} \sum_{t=1}^\tau \sum_{i \in [m]} \epsilon_t^{(i)},
$$

since the *dynamic* optimum satisfies the learning plan at each round, up to the error terms.

Combining everything, we get, with probability at least $1 - \delta$, the following lower bound:

$$
\sum_{t=1}^\tau \left( \mathop{\mathbb{E}}_{\boldsymbol{x}_t \sim \boldsymbol{\xi}_t^*} [f_t(\boldsymbol{x})] - \sum_{i \in [m]} \boldsymbol{\lambda}_t[i] \cdot \left( B_t^{(i)} - \mathop{\mathbb{E}}_{\boldsymbol{x} \sim \boldsymbol{\xi}_t^*} [c_t(\boldsymbol{x})[i]] \right) \right)
$$
$$
\geq \sum_{t=1}^\tau \left( \mathop{\mathbb{E}}_{\boldsymbol{x} \sim \boldsymbol{\xi}_t^*} [\bar{f}_t(\boldsymbol{x})] - \sum_{i \in [m]} \boldsymbol{\lambda}_t[i] \cdot \left( B_t^{(i)} - \mathop{\mathbb{E}}_{\boldsymbol{x} \sim \boldsymbol{\xi}_t^*} [\bar{c}_t(\boldsymbol{x})[i]] \right) \right) - \left( 4 + 4 \max_{\boldsymbol{\lambda} \in \mathcal{L}} \|\boldsymbol{\lambda}\|_1 \right) \sqrt{2\tau \ln \frac{T}{\delta}}
$$
$$
\geq \sum_{t=1}^\tau \mathop{\mathbb{E}}_{\boldsymbol{x} \sim \boldsymbol{\xi}_t^*} [\bar{f}_t(\boldsymbol{x})] - \frac{1}{\rho_{\min}} \sum_{t=1}^\tau \sum_{i \in [m]} \epsilon_t^{(i)} - \left( 4 + 4 \max_{\boldsymbol{\lambda} \in \mathcal{L}} \|\boldsymbol{\lambda}\|_1 \right) \sqrt{2\tau \ln \frac{T}{\delta}}
$$

$$\geq \mathrm{OPT}_{\mathcal{D}}(\epsilon_t) - (T - \tau) - \frac{1}{\rho_{\min}} \sum_{t=1}^{\tau} \sum_{i \in [m]} \epsilon_t^{(i)} - \left(4 + 4 \max_{\boldsymbol{\lambda} \in \mathcal{L}} \|\boldsymbol{\lambda}\|_1\right) \sqrt{2\tau \ln \frac{T}{\delta}}.$$

Noticing that by the update of Algorithm 1, it holds:

$$\sum_{t=1}^{\tau} \mathbb{E}_{\boldsymbol{x} \sim \boldsymbol{\xi}_t} [f_t(\boldsymbol{x})] = \sum_{t=1}^{T} \mathbb{E}_{\boldsymbol{x} \sim \boldsymbol{\xi}_t} [f_t(\boldsymbol{x})],$$

and employing Lemma E.1, we get the following bound:

$$\sum_{t=1}^{T} f_t(\boldsymbol{x}_t) \geq \mathrm{OPT}_{\mathcal{D}}(\epsilon_t) - (T - \tau) - \frac{1}{\rho_{\min}} \sum_{t=1}^{\tau} \sum_{i \in [m]} \epsilon_t^{(i)} - \left(4 + 4 \max_{\boldsymbol{\lambda} \in \mathcal{L}} \|\boldsymbol{\lambda}\|_1\right) \sqrt{2\tau \ln \frac{T}{\delta}}$$

$$- \sum_{t=1}^{\tau} \sum_{i \in [m]} \boldsymbol{\lambda}[i] \cdot \left(B_t^{(i)} - c_t(\boldsymbol{x}_t)[i]\right) - \frac{2}{\rho_{\min}} R_\tau^D - (4 + 4\|\boldsymbol{\lambda}\|_1) \sqrt{2\tau \ln \frac{T}{\delta}},$$

which holds with probability at least $1 - 2\delta$, by Union Bound. Thus, employing the same analysis of Theorem 3.2 and noticing that:

$$\sum_{t=1}^{\tau} \sum_{i \in [m]} \epsilon_t^{(i)} \leq \sum_{t=1}^{T} \sum_{i \in [m]} \epsilon_t^{(i)},$$

concludes the proof. $\qquad \square$

## B.2 Theoretical Guarantees of Algorithm 3

In this section, we present the results attained by the meta procedure provided in Algorithm 3.

We start by the following lemma.

**Lemma B.2.** *Algorithm 3, when instantiated with a dual regret minimizer which attains a regret upper bound $R_T^D$, guarantees the following bound:*

$$\sum_{t=1}^{\tau} \mathbb{E}_{\boldsymbol{x} \sim \boldsymbol{\xi}_t} [f_t(\boldsymbol{x})] \geq \sum_{t=1}^{\tau} \left( \mathbb{E}_{\boldsymbol{x} \sim \boldsymbol{\xi}^{(t)}} [f_t(\boldsymbol{x})] - \sum_{i \in [m]} \boldsymbol{\lambda}_t[i] \cdot \left(\overline{B}_t^{(i)} - \mathbb{E}_{\boldsymbol{x} \sim \boldsymbol{\xi}^{(t)}} [c_t(\boldsymbol{x})[i]]\right) \right)$$

$$- \sum_{t=1}^{\tau} \sum_{i \in [m]} \boldsymbol{\lambda}[i] \cdot \left(\overline{B}_t^{(i)} - \mathbb{E}_{\boldsymbol{x} \sim \boldsymbol{\xi}_t} [c_t(\boldsymbol{x})[i]]\right) - \frac{2}{\hat{\rho}} R_\tau^D.$$

*where $\boldsymbol{\lambda} \in \mathcal{L}$ is an arbitrary Lagrange multiplier, $\{\boldsymbol{\xi}^{(t)} \in \Xi\}_{t=1}^{\tau}$ is an arbitrary sequence of strategy mixtures and $\tau$ is the stopping time of the algorithm.*

*Proof.* The proof is equivalent to the one of Lemma B.1 after substituting $\rho_{\min}$ with $\hat{\rho}$ and $B_t^{(i)}$ with $\overline{B}_t^{(i)}$, for all $i \in [m], t \in [T]$. $\qquad \square$

**Lemma B.3.** *For any $\delta \in (0, 1)$, Algorithm 3, when instantiated with a dual regret minimizer which attains a regret upper bound $R_T^D$, guarantees, with probability at least $1 - \delta$:*

$$\sum_{t=1}^{\tau} \mathbb{E}_{\boldsymbol{x} \sim \boldsymbol{\xi}_t} [f_t(\boldsymbol{x})] \geq \mathrm{OPT}_{\mathcal{D}} - (T - \tau) - T^{\frac{3}{4}} - \left(4 + \frac{4T^{\frac{1}{4}}}{\rho}\right) \sqrt{2\tau \ln \frac{T}{\delta}} - \frac{2T^{\frac{1}{4}}}{\rho} R_T^D,$$

*where $\tau$ is the stopping time of the algorithm.*

*Proof.* We first employ Lemma B.2 and the definition of $\hat{\rho}$ to obtain:

$$\sum_{t=1}^{\tau} \mathbb{E}_{\boldsymbol{x} \sim \boldsymbol{\xi}_t} [f_t(\boldsymbol{x})] \geq \sum_{t=1}^{\tau} \left( \mathbb{E}_{\boldsymbol{x} \sim \boldsymbol{\xi}^{(t)}} [f_t(\boldsymbol{x})] - \sum_{i \in [m]} \boldsymbol{\lambda}_t[i] \cdot \left(\overline{B}_t^{(i)} - \mathbb{E}_{\boldsymbol{x} \sim \boldsymbol{\xi}^{(t)}} [c_t(\boldsymbol{x})[i]]\right) \right)$$

$$-\sum_{t=1}^{\tau}\sum_{i\in[m]}\boldsymbol{\lambda}[i]\cdot\left(\overline{B}_t^{(i)}-\mathop{\mathbb{E}}_{\boldsymbol{x}\sim\boldsymbol{\xi}_t}[c_t(\boldsymbol{x})[i]]\right)-\frac{2T^{\frac{1}{4}}}{\rho}R_\tau^D.$$

Hence, we select the Lagrange variable as $\boldsymbol{\lambda}=\boldsymbol{0}$ vector to get the following bound:

$$\sum_{t=1}^{\tau}\mathop{\mathbb{E}}_{\boldsymbol{x}\sim\boldsymbol{\xi}_t}[f_t(\boldsymbol{x})]\geq\sum_{t=1}^{\tau}\left(\mathop{\mathbb{E}}_{\boldsymbol{x}\sim\boldsymbol{\xi}^{(t)}}[f_t(\boldsymbol{x})]-\sum_{i\in[m]}\boldsymbol{\lambda}_t[i]\cdot\left(\overline{B}_t^{(i)}-\mathop{\mathbb{E}}_{\boldsymbol{x}\sim\boldsymbol{\xi}^{(t)}}[c_t(\boldsymbol{x})[i]]\right)\right)-\frac{2T^{\frac{1}{4}}}{\rho}R_\tau^D.$$

We now focus on bounding the term:

$$\sum_{t=1}^{\tau}\left(\mathop{\mathbb{E}}_{\boldsymbol{x}\sim\boldsymbol{\xi}^{(t)}}[f_t(\boldsymbol{x})]-\sum_{i\in[m]}\boldsymbol{\lambda}_t[i]\cdot\left(\overline{B}_t^{(i)}-\mathop{\mathbb{E}}_{\boldsymbol{x}\sim\boldsymbol{\xi}^{(t)}}[c_t(\boldsymbol{x})[i]]\right)\right).$$

Similarly to Lemma 3.1, we notice that by the definition of Program (1), there exists a sequence strategy mixture $\{\boldsymbol{\xi}_t^*\}_{t=1}^T$ such that $\mathbb{E}_{\boldsymbol{x}\sim\boldsymbol{\xi}_t^*}[\bar{c}_t(\boldsymbol{x})]\leq B_t^{(i)}$ for all $t\in[T],i\in[m]$ and $\sum_{t=1}^T\mathbb{E}_{\boldsymbol{x}\sim\boldsymbol{\xi}_t^*}[\bar{f}_t(\boldsymbol{x})]\geq\mathrm{OPT}_{\mathcal{D}}-\gamma$, for all $\gamma>0$. In the rest of the proof, we will omit the the dependence on $\gamma$, since it can be chosen arbitrarily small, thus being negligible in the regret bound.

We then define the following strategy mixture $\boldsymbol{\xi}_t^\diamond$ for all $t\in[T]$ as follows:

$$\boldsymbol{\xi}_t^\diamond:=\begin{cases}\boldsymbol{x}^\varnothing & \text{w.p. } 1/T^{1/4}\\ \boldsymbol{\xi}_t^* & \text{w.p. } 1-1/T^{1/4}\end{cases}.$$

Thus, we first show that $\boldsymbol{\xi}_t^\diamond$ satisfies the per-round expected constraints defined by $\overline{B}_t^{(i)}$. Indeed it holds, for all $i\in[m]$:

$$\overline{B}_t^{(i)}-\mathop{\mathbb{E}}_{\boldsymbol{x}\sim\boldsymbol{\xi}_t^\diamond}[\bar{c}_t(\boldsymbol{x})]=\overline{B}_t^{(i)}-\left(1-\frac{1}{T^{1/4}}\right)\mathop{\mathbb{E}}_{\boldsymbol{x}\sim\boldsymbol{\xi}_t^*}[\bar{c}_t(\boldsymbol{x})]-\frac{1}{T^{1/4}}\mathop{\mathbb{E}}_{\boldsymbol{x}\sim\boldsymbol{\xi}^\varnothing}[\bar{c}_t(\boldsymbol{x})]$$

$$=\left(1-\frac{1}{T^{1/4}}\right)B_t^{(i)}-\left(1-\frac{1}{T^{1/4}}\right)\mathop{\mathbb{E}}_{\boldsymbol{x}\sim\boldsymbol{\xi}_t^*}[\bar{c}_t(\boldsymbol{x})]-\frac{1}{T^{1/4}}\mathop{\mathbb{E}}_{\boldsymbol{x}\sim\boldsymbol{\xi}^\varnothing}[\bar{c}_t(\boldsymbol{x})]$$

$$=\left(1-\frac{1}{T^{1/4}}\right)B_t^{(i)}-\left(1-\frac{1}{T^{1/4}}\right)\mathop{\mathbb{E}}_{\boldsymbol{x}\sim\boldsymbol{\xi}_t^*}[\bar{c}_t(\boldsymbol{x})]$$

$$\geq 0,$$

where we employed the definition of $\boldsymbol{\xi}_t^*$ and $\boldsymbol{\xi}^\varnothing$.

Thus, returning to the quantity of interest and selecting $\{\boldsymbol{\xi}^{(t)}\}_{t=1}^\tau=\{\boldsymbol{\xi}_t^\diamond\}_{t=1}^\tau$, it holds:

$$\sum_{t=1}^{\tau}\left[\mathop{\mathbb{E}}_{\boldsymbol{x}\sim\boldsymbol{\xi}^{(t)}}[f_t(\boldsymbol{x})]+\sum_{i\in[m]}\boldsymbol{\lambda}_t[i]\cdot\left(\overline{B}_t^{(i)}-\mathop{\mathbb{E}}_{\boldsymbol{x}\sim\boldsymbol{\xi}^{(t)}}[c_t(\boldsymbol{x})[i]]\right)\right]$$

$$=\sum_{t=1}^{\tau}\left[\mathop{\mathbb{E}}_{\boldsymbol{x}\sim\boldsymbol{\xi}_t^\diamond}[f_t(\boldsymbol{x})]+\sum_{i\in[m]}\boldsymbol{\lambda}_t[i]\cdot\left(\overline{B}_t^{(i)}-\mathop{\mathbb{E}}_{\boldsymbol{x}\sim\boldsymbol{\xi}_t^\diamond}[c_t(\boldsymbol{x})[i]]\right)\right]$$

$$\geq\sum_{t=1}^{\tau}\left[\mathop{\mathbb{E}}_{\boldsymbol{x}\sim\boldsymbol{\xi}_t^\diamond}[\bar{f}_t(\boldsymbol{x})]+\sum_{i\in[m]}\boldsymbol{\lambda}_t[i]\cdot\left(\overline{B}_t^{(i)}-\mathop{\mathbb{E}}_{\boldsymbol{x}\sim\boldsymbol{\xi}_t^\diamond}[\bar{c}_t(\boldsymbol{x})[i]]\right)\right]-\left(4+\frac{4T^{1/4}}{\rho}\right)\sqrt{2\tau\ln\frac{T}{\delta}}$$

$$\geq\sum_{t=1}^{\tau}\left[\mathop{\mathbb{E}}_{\boldsymbol{x}\sim\boldsymbol{\xi}_t^\diamond}[\bar{f}_t(\boldsymbol{x})]\right]-\left(4+\frac{4T^{1/4}}{\rho}\right)\sqrt{2\tau\ln\frac{T}{\delta}}$$

$$=\sum_{t=1}^{\tau}\left[\left(1-\frac{1}{T^{1/4}}\right)\mathop{\mathbb{E}}_{\boldsymbol{x}\sim\boldsymbol{\xi}_t^*}[\bar{f}_t(\boldsymbol{x})]+\frac{1}{T^{1/4}}\mathop{\mathbb{E}}_{\boldsymbol{x}\sim\boldsymbol{\xi}^\varnothing}[\bar{f}_t(\boldsymbol{x})]\right]-\left(4+\frac{4T^{1/4}}{\rho}\right)\sqrt{2\tau\ln\frac{T}{\delta}}$$

$$=\sum_{t=1}^{\tau}\left[\left(1-\frac{1}{T^{1/4}}\right)\mathop{\mathbb{E}}_{\boldsymbol{x}\sim\boldsymbol{\xi}_t^*}[\bar{f}_t(\boldsymbol{x})]\right]-\left(4+\frac{4T^{1/4}}{\rho}\right)\sqrt{2\tau\ln\frac{T}{\delta}}$$

$$\geq \sum_{t=1}^{\tau} \mathop{\mathbb{E}}_{\boldsymbol{x} \sim \boldsymbol{\xi}_t^*} \left[ \bar{f}_t(\boldsymbol{x}) \right] - T^{3/4} - \left( 4 + \frac{4T^{1/4}}{\rho} \right) \sqrt{2\tau \ln \frac{T}{\delta}}$$

$$\geq \mathrm{OPT}_{\mathcal{D}} - T^{3/4} - (T - \tau) - \left( 4 + \frac{4T^{1/4}}{\rho} \right) \sqrt{2\tau \ln \frac{T}{\delta}},$$

where the second inequality holds, with probability at least $1 - \delta$, by Lemma E.2 and upper bounding the Lagrangian multiplier with $T^{\frac{1}{4}}/\rho$. This concludes the proof. $\qquad\square$

Hence, we proceed upper bounding the difference between the horizon $T$ and the stopping time $\tau$.

**Lemma 5.1.** *For any $\delta \in (0, 1)$, Algorithm 3 instantiated with a dual regret minimizer which attains a regret upper bound $R_T^D$, guarantees with probability at least $1 - \delta$, $T - \tau \leq \frac{14}{\rho} \left( \sqrt{\ln \frac{T}{\delta}} + \frac{R_T^D}{\sqrt{T}} \right) T^{\frac{3}{4}}$.*

*Proof.* Suppose by contradiction that $T - \tau > CT^{3/4}$, thus, it holds $\tau < T - CT^{3/4}$.

We proceed upper and lower bounding the value of the Lagrangian. We first lower bound the quantity of interest employing the strategy mixture selection of Algorithm 3. Given that, it holds:

$$\sum_{t=1}^{\tau} \left[ \mathop{\mathbb{E}}_{\boldsymbol{x} \sim \boldsymbol{\xi}_t} \left[ f_t(\boldsymbol{x}) \right] - \sum_{i \in [m]} \boldsymbol{\lambda}_t[i] \cdot \mathop{\mathbb{E}}_{\boldsymbol{x} \sim \boldsymbol{\xi}_t} \left[ c_t(\boldsymbol{x})[i] \right] \right] \geq \sum_{t=1}^{\tau} \left[ \mathop{\mathbb{E}}_{\boldsymbol{x} \sim \boldsymbol{\xi}^\varnothing} \left[ f_t(\boldsymbol{x}) \right] - \sum_{i \in [m]} \boldsymbol{\lambda}_t[i] \cdot \mathop{\mathbb{E}}_{\boldsymbol{x} \sim \boldsymbol{\xi}^\varnothing} \left[ c_t(\boldsymbol{x})[i] \right] \right]$$
$$= 0.$$

Differently, to upper bound the same quantity, we employ the no-regret property of the dual algorithm $\mathcal{R}^D$. Hence, it holds:

$$\sum_{t=1}^{\tau} \left[ \mathop{\mathbb{E}}_{\boldsymbol{x} \sim \boldsymbol{\xi}_t} \left[ f_t(\boldsymbol{x}) \right] - \sum_{i \in [m]} \boldsymbol{\lambda}_t[i] \cdot \mathop{\mathbb{E}}_{\boldsymbol{x} \sim \boldsymbol{\xi}_t} \left[ c_t(\boldsymbol{x})[i] \right] \right]$$

$$\leq \tau - \sum_{t=1}^{\tau} \sum_{i \in [m]} \boldsymbol{\lambda}_t[i] \cdot \mathop{\mathbb{E}}_{\boldsymbol{x} \sim \boldsymbol{\xi}_t} \left[ c_t(\boldsymbol{x})[i] \right]$$

$$\leq \tau + \sum_{t=1}^{\tau} \sum_{i \in [m]} \boldsymbol{\lambda}[i] \cdot \left( \overline{B}_t^{(i)} - \mathop{\mathbb{E}}_{\boldsymbol{x} \sim \boldsymbol{\xi}_t} \left[ c_t(\boldsymbol{x})[i] \right] \right) + \frac{2T^{\frac{1}{4}}}{\rho} R_\tau^D - \sum_{t=1}^{\tau} \sum_{i \in [m]} \boldsymbol{\lambda}_t[i] \cdot \overline{B}_t^{(i)} \qquad (6a)$$

$$\leq \tau + \sum_{t=1}^{\tau} \sum_{i \in [m]} \boldsymbol{\lambda}[i] \cdot \left( \overline{B}_t^{(i)} - \mathop{\mathbb{E}}_{\boldsymbol{x} \sim \boldsymbol{\xi}_t} \left[ c_t(\boldsymbol{x})[i] \right] \right) + \frac{2T^{\frac{1}{4}}}{\rho} R_\tau^D$$

$$= \tau + \sum_{t=1}^{\tau} \sum_{i \in [m]} \boldsymbol{\lambda}[i] \cdot \left( \left( 1 - \frac{1}{T^{1/4}} \right) B_t^{(i)} - \mathop{\mathbb{E}}_{\boldsymbol{x} \sim \boldsymbol{\xi}_t} \left[ c_t(\boldsymbol{x})[i] \right] \right) + \frac{2T^{\frac{1}{4}}}{\rho} R_\tau^D$$

$$= \tau + \frac{T^{1/4}}{\rho} \cdot \sum_{t=1}^{\tau} \left( \left( 1 - \frac{1}{T^{1/4}} \right) B_t^{(i^*)} - \mathop{\mathbb{E}}_{\boldsymbol{x} \sim \boldsymbol{\xi}_t} \left[ c_t(\boldsymbol{x})[i^*] \right] \right) + \frac{2T^{\frac{1}{4}}}{\rho} R_\tau^D \qquad (6b)$$

$$\leq \tau + \frac{T^{1/4}}{\rho} \cdot \sum_{t=1}^{\tau} \left( \left( 1 - \frac{1}{T^{1/4}} \right) B_t^{(i^*)} - c_t(\boldsymbol{x}_t)[i^*] \right) + \frac{8T^{3/4}}{\rho} \sqrt{\ln \frac{T}{\delta}} + \frac{2T^{\frac{1}{4}}}{\rho} R_\tau^D \qquad (6c)$$

$$\leq \tau + \frac{T^{1/4}}{\rho} \cdot \left( \left( 1 - \frac{1}{T^{1/4}} \right) T\rho - T\rho + 1 \right) + \frac{8T^{3/4}}{\rho} \sqrt{\ln \frac{T}{\delta}} + \frac{2T^{\frac{1}{4}}}{\rho} R_\tau^D \qquad (6d)$$

$$= \tau + \frac{T^{1/4}}{\rho} - T + \frac{8T^{3/4}}{\rho} \sqrt{\ln \frac{T}{\delta}} + \frac{2T^{\frac{1}{4}}}{\rho} R_\tau^D$$

$$< T - CT^{3/4} + \frac{T^{1/4}}{\rho} - T + \frac{8T^{3/4}}{\rho} \sqrt{\ln \frac{T}{\delta}} + \frac{2T^{\frac{1}{4}}}{\rho} R_\tau^D \qquad (6e)$$

$$= -CT^{3/4} + \frac{T^{1/4}}{\rho} + \frac{8T^{3/4}}{\rho} \sqrt{\ln \frac{T}{\delta}} + \frac{2T^{\frac{1}{4}}}{\rho} R_\tau^D,$$

where Inequality (6a) holds by the no-regret property of the dual regret minimizer $\mathcal{R}^D$, Equation (6b) holds selecting $\boldsymbol{\lambda}$ s.t. $\boldsymbol{\lambda}[i^*] = T^{1/4}/\rho$ and $\boldsymbol{\lambda}[i] = 0$ for all other $i \in [m]$, where $i^*$ is the depleted resource – notice that, there must exists a depleted resource since $T - \tau > 0$ –, Inequality (6c) holds with probability at least $1 - \delta$ employing the Azuma-Höeffding inequality, Inequality (6d) holds since the following holds for the resource $i^* \in [m]$:

$$\sum_{t=1}^{\tau} c_t(\boldsymbol{x}_t)[i^*] + 1 \geq B = T\rho,$$

and finally Inequality (6e) holds since $T - \tau > CT^{3/4}$.

Setting $C \geq \frac{14}{\rho}\left(\sqrt{\ln\frac{T}{\delta}} + \frac{R_T^D}{\sqrt{T}}\right)$ we reach the contradiction. This concludes the proof. $\qquad\square$

We conclude the section by proving the final *dynamic* regret bound of Algorithm 3.

**Theorem 5.2.** *For any $\delta \in (0, 1)$, Algorithm 3, when instantiated with a dual regret minimizer which attains a regret upper bound $R_T^D$, guarantees, with probability at least $1 - 3\delta$, $\mathfrak{R}_T \leq \frac{14}{\rho}\left(\sqrt{\ln\frac{T}{\delta}} + \frac{R_T^D}{\sqrt{T}}\right)T^{\frac{3}{4}} + T^{\frac{3}{4}} + \left(8 + \frac{4T^{\frac{1}{4}}}{\rho}\right)\sqrt{2T\ln\frac{T}{\delta}} + \frac{2T^{\frac{1}{4}}}{\rho}R_T^D$.*

*Proof.* We first employ Lemma B.3 to get the following bound, with probability at least $1 - \delta$:

$$\sum_{t=1}^{\tau}\mathbb{E}_{\boldsymbol{x}\sim\boldsymbol{\xi}_t}[f_t(\boldsymbol{x})] \geq \text{OPT}_{\mathcal{D}} - (T - \tau) - T^{\frac{3}{4}} - \left(4 + \frac{4T^{\frac{1}{4}}}{\rho}\right)\sqrt{2\tau\ln\frac{T}{\delta}} - \frac{2T^{\frac{1}{4}}}{\rho}R_\tau^D,$$

Thus, we employ the Azuma-Höeffding inequality to get, with probability at least $1 - 2\delta$ by Union Bound:

$$\sum_{t=1}^{\tau}f_t(\boldsymbol{x}_t) \geq \text{OPT}_{\mathcal{D}} - (T - \tau) - T^{\frac{3}{4}} - \left(4 + \frac{4T^{\frac{1}{4}}}{\rho}\right)\sqrt{2\tau\ln\frac{T}{\delta}} - \frac{2T^{\frac{1}{4}}}{\rho}R_\tau^D - 4\sqrt{2\tau\ln\frac{T}{\delta}},$$

which in turn implies, with probability at least $1 - 2\delta$:

$$\mathfrak{R}_T \leq (T - \tau) + T^{\frac{3}{4}} + \left(4 + \frac{4T^{\frac{1}{4}}}{\rho}\right)\sqrt{2\tau\ln\frac{T}{\delta}} + \frac{2T^{\frac{1}{4}}}{\rho}R_\tau^D + 4\sqrt{2\tau\ln\frac{T}{\delta}}.$$

We then apply Lemma 5.1 to obtain, with probability at least $1 - 3\delta$, by Union Bound, the following regret upper bound:

$$\mathfrak{R}_T \leq \frac{14}{\rho}\left(\sqrt{\ln\frac{T}{\delta}} + \frac{R_T^D}{\sqrt{T}}\right)T^{\frac{3}{4}} + T^{\frac{3}{4}} + \left(4 + \frac{4T^{\frac{1}{4}}}{\rho}\right)\sqrt{2\tau\ln\frac{T}{\delta}} + \frac{2T^{\frac{1}{4}}}{\rho}R_\tau^D + 4\sqrt{2\tau\ln\frac{T}{\delta}}.$$

To get the final regret bound we notice the following trivial upper bounds:

$$R_\tau^D \leq R_T^D, \quad \sqrt{2\tau\ln\frac{T}{\delta}} \leq \sqrt{2T\ln\frac{T}{\delta}}.$$

This concludes the proof. $\qquad\square$

### B.2.1 Robustness to Baselines Deviating from the Spending Plan

We provide the regret of Algorithm 3 with respect to a baseline which deviates from the spending plan.

**Theorem 5.4.** *For any $\delta \in (0, 1)$, Algorithm 3, when instantiated with a dual regret minimizer which attains a regret upper bound $R_T^D$, guarantees, with probability at least $1 - 3\delta$, $\mathfrak{R}_T(\epsilon_t) \leq \frac{14}{\rho}\left(\sqrt{\ln\frac{T}{\delta}} + \frac{R_T^D}{\sqrt{T}}\right)T^{\frac{3}{4}} + T^{\frac{3}{4}} + \left(8 + \frac{4T^{\frac{1}{4}}}{\rho}\right)\sqrt{2T\ln\frac{T}{\delta}} + \frac{2T^{\frac{1}{4}}}{\rho}R_T^D + \frac{T^{\frac{1}{4}}}{\rho}\sum_{t=1}^{T}\sum_{i\in[m]}\epsilon_t^{(i)}$.*

*Proof.* Similarly to Lemma B.3, we first employ Lemma B.2 and the definition of $\hat{\rho}$:

$$\sum_{t=1}^{\tau} \mathop{\mathbb{E}}_{\boldsymbol{x} \sim \boldsymbol{\xi}_t} [f_t(\boldsymbol{x})] \geq \sum_{t=1}^{\tau} \left( \mathop{\mathbb{E}}_{\boldsymbol{x} \sim \boldsymbol{\xi}^{(t)}} [f_t(\boldsymbol{x})] - \sum_{i \in [m]} \boldsymbol{\lambda}_t[i] \cdot \left( \overline{B}_t^{(i)} - \mathop{\mathbb{E}}_{\boldsymbol{x} \sim \boldsymbol{\xi}^{(t)}} [c_t(\boldsymbol{x})[i]] \right) \right)$$
$$- \sum_{t=1}^{\tau} \sum_{i \in [m]} \boldsymbol{\lambda}[i] \cdot \left( \overline{B}_t^{(i)} - \mathop{\mathbb{E}}_{\boldsymbol{x} \sim \boldsymbol{\xi}_t} [c_t(\boldsymbol{x})[i]] \right) - \frac{2T^{\frac{1}{4}}}{\rho} R_\tau^D.$$

Hence, we select the Lagrange variable as $\boldsymbol{\lambda} = \mathbf{0}$ vector to get the following bound:

$$\sum_{t=1}^{\tau} \mathop{\mathbb{E}}_{\boldsymbol{x} \sim \boldsymbol{\xi}_t} [f_t(\boldsymbol{x})] \geq \sum_{t=1}^{\tau} \left( \mathop{\mathbb{E}}_{\boldsymbol{x} \sim \boldsymbol{\xi}^{(t)}} [f_t(\boldsymbol{x})] - \sum_{i \in [m]} \boldsymbol{\lambda}_t[i] \cdot \left( \overline{B}_t^{(i)} - \mathop{\mathbb{E}}_{\boldsymbol{x} \sim \boldsymbol{\xi}^{(t)}} [c_t(\boldsymbol{x})[i]] \right) \right) - \frac{2T^{\frac{1}{4}}}{\rho} R_\tau^D.$$

We now focus on bounding the following term when a baseline deviating from the spending plan is employed:

$$\sum_{t=1}^{\tau} \left( \mathop{\mathbb{E}}_{\boldsymbol{x} \sim \boldsymbol{\xi}^{(t)}} [f_t(\boldsymbol{x})] - \sum_{i \in [m]} \boldsymbol{\lambda}_t[i] \cdot \left( \overline{B}_t^{(i)} - \mathop{\mathbb{E}}_{\boldsymbol{x} \sim \boldsymbol{\xi}^{(t)}} [c_t(\boldsymbol{x})[i]] \right) \right).$$

We notice that by the definition of Program (3), there exists a sequence strategy mixture $\{\boldsymbol{\xi}_t^*\}_{t=1}^T$ such that $\mathbb{E}_{\boldsymbol{x} \sim \boldsymbol{\xi}_t^*} [\bar{c}_t(\boldsymbol{x})] \leq B_t^{(i)} - \epsilon_t^{(i)}$ for all $t \in [T], i \in [m]$ and $\sum_{t=1}^T \mathbb{E}_{\boldsymbol{x} \sim \boldsymbol{\xi}_t^*} [\bar{f}_t(\boldsymbol{x})] \geq \mathrm{OPT}_{\mathcal{D}}(\epsilon_t) - \gamma$, for all $\gamma > 0$. In the rest of the proof, we will omit the the dependence on $\gamma$, since it can be chosen arbitrarily small, thus being negligible in the regret bound.

We then define the following strategy mixture $\boldsymbol{\xi}_t^\diamond$ for all $t \in [T]$ as follows:

$$\boldsymbol{\xi}_t^\diamond := \begin{cases} \boldsymbol{x}^\varnothing & \text{w.p. } 1/T^{1/4} \\ \boldsymbol{\xi}_t^* & \text{w.p. } 1 - 1/T^{1/4} \end{cases}.$$

Thus, we first show that $\boldsymbol{\xi}_t^\diamond$ satisfies the per-round expected constraints defined by $\overline{B}_t^{(i)}$, up to the errors terms. Indeed it holds, for all $i \in [m]$:

$$\overline{B}_t^{(i)} - \mathop{\mathbb{E}}_{\boldsymbol{x} \sim \boldsymbol{\xi}_t^\diamond} [\bar{c}_t(\boldsymbol{x})] = \overline{B}_t^{(i)} - \left( 1 - \frac{1}{T^{1/4}} \right) \mathop{\mathbb{E}}_{\boldsymbol{x} \sim \boldsymbol{\xi}_t^*} [\bar{c}_t(\boldsymbol{x})] - \frac{1}{T^{1/4}} \mathop{\mathbb{E}}_{\boldsymbol{x} \sim \boldsymbol{\xi}^\varnothing} [\bar{c}_t(\boldsymbol{x})]$$
$$= \left( 1 - \frac{1}{T^{1/4}} \right) B_t^{(i)} - \left( 1 - \frac{1}{T^{1/4}} \right) \mathop{\mathbb{E}}_{\boldsymbol{x} \sim \boldsymbol{\xi}_t^*} [\bar{c}_t(\boldsymbol{x})] - \frac{1}{T^{1/4}} \mathop{\mathbb{E}}_{\boldsymbol{x} \sim \boldsymbol{\xi}^\varnothing} [\bar{c}_t(\boldsymbol{x})]$$
$$= \left( 1 - \frac{1}{T^{1/4}} \right) B_t^{(i)} - \left( 1 - \frac{1}{T^{1/4}} \right) \mathop{\mathbb{E}}_{\boldsymbol{x} \sim \boldsymbol{\xi}_t^*} [\bar{c}_t(\boldsymbol{x})]$$
$$\geq - \left( 1 - \frac{1}{T^{1/4}} \right) \epsilon_t^{(i)}$$
$$\geq -\epsilon_t^{(i)},$$

where we employed the definition of $\boldsymbol{\xi}_t^*$ and $\boldsymbol{\xi}^\varnothing$.

Thus, returning to the quantity of interest and selecting $\{\boldsymbol{\xi}^{(t)}\}_{t=1}^{\tau} = \{\boldsymbol{\xi}_t^\diamond\}_{t=1}^{\tau}$, it holds:

$$\sum_{t=1}^{\tau} \left[ \mathop{\mathbb{E}}_{\boldsymbol{x} \sim \boldsymbol{\xi}^{(t)}} [f_t(\boldsymbol{x})] + \sum_{i \in [m]} \boldsymbol{\lambda}_t[i] \cdot \left( \overline{B}_t^{(i)} - \mathop{\mathbb{E}}_{\boldsymbol{x} \sim \boldsymbol{\xi}^{(t)}} [c_t(\boldsymbol{x})[i]] \right) \right]$$
$$= \sum_{t=1}^{\tau} \left[ \mathop{\mathbb{E}}_{\boldsymbol{x} \sim \boldsymbol{\xi}_t^\diamond} [f_t(\boldsymbol{x})] + \sum_{i \in [m]} \boldsymbol{\lambda}_t[i] \cdot \left( \overline{B}_t^{(i)} - \mathop{\mathbb{E}}_{\boldsymbol{x} \sim \boldsymbol{\xi}_t^\diamond} [c_t(\boldsymbol{x})[i]] \right) \right]$$
$$\geq \sum_{t=1}^{\tau} \left[ \mathop{\mathbb{E}}_{\boldsymbol{x} \sim \boldsymbol{\xi}_t^\diamond} [\bar{f}_t(\boldsymbol{x})] + \sum_{i \in [m]} \boldsymbol{\lambda}_t[i] \cdot \left( \overline{B}_t^{(i)} - \mathop{\mathbb{E}}_{\boldsymbol{x} \sim \boldsymbol{\xi}_t^\diamond} [\bar{c}_t(\boldsymbol{x})[i]] \right) \right] - \left( 4 + \frac{4T^{1/4}}{\rho} \right) \sqrt{2\tau \ln \frac{T}{\delta}}$$

$$\geq \sum_{t=1}^{\tau} \left[ \underset{\boldsymbol{x} \sim \boldsymbol{\xi}_t^{\diamond}}{\mathbb{E}} \left[ \bar{f}_t(\boldsymbol{x}) \right] \right] - \frac{T^{1/4}}{\rho} \sum_{t=1}^{\tau} \sum_{i \in [m]} \epsilon_t^{(i)} - \left( 4 + \frac{4T^{1/4}}{\rho} \right) \sqrt{2\tau \ln \frac{T}{\delta}}$$

$$= \sum_{t=1}^{\tau} \left[ \left( 1 - \frac{1}{T^{1/4}} \right) \underset{\boldsymbol{x} \sim \boldsymbol{\xi}_t^{*}}{\mathbb{E}} \left[ \bar{f}_t(\boldsymbol{x}) \right] + \frac{1}{T^{1/4}} \underset{\boldsymbol{x} \sim \boldsymbol{\xi}^{\varnothing}}{\mathbb{E}} \left[ \bar{f}_t(\boldsymbol{x}) \right] \right] - \frac{T^{1/4}}{\rho} \sum_{t=1}^{\tau} \sum_{i \in [m]} \epsilon_t^{(i)}$$

$$- \left( 4 + \frac{4T^{1/4}}{\rho} \right) \sqrt{2\tau \ln \frac{T}{\delta}}$$

$$= \sum_{t=1}^{\tau} \left[ \left( 1 - \frac{1}{T^{1/4}} \right) \underset{\boldsymbol{x} \sim \boldsymbol{\xi}_t^{*}}{\mathbb{E}} \left[ \bar{f}_t(\boldsymbol{x}) \right] \right] - \frac{T^{1/4}}{\rho} \sum_{t=1}^{\tau} \sum_{i \in [m]} \epsilon_t^{(i)} - \left( 4 + \frac{4T^{1/4}}{\rho} \right) \sqrt{2\tau \ln \frac{T}{\delta}}$$

$$\geq \sum_{t=1}^{\tau} \underset{\boldsymbol{x} \sim \boldsymbol{\xi}_t^{*}}{\mathbb{E}} \left[ \bar{f}_t(\boldsymbol{x}) \right] - T^{3/4} - \frac{T^{1/4}}{\rho} \sum_{t=1}^{\tau} \sum_{i \in [m]} \epsilon_t^{(i)} - \left( 4 + \frac{4T^{1/4}}{\rho} \right) \sqrt{2\tau \ln \frac{T}{\delta}}$$

$$\geq \mathrm{OPT}_{\mathcal{D}}(\epsilon_t) - T^{3/4} - (T - \tau) - \frac{T^{1/4}}{\rho} \sum_{t=1}^{\tau} \sum_{i \in [m]} \epsilon_t^{(i)} - \left( 4 + \frac{4T^{1/4}}{\rho} \right) \sqrt{2\tau \ln \frac{T}{\delta}},$$

where the second inequality holds, with probability at least $1 - \delta$, by Lemma E.2 and upper bounding the Lagrangian multiplier with $T^{\frac{1}{4}}/\rho$. Employing the same analysis of Theorem 5.2 and noticing that:

$$\sum_{t=1}^{\tau} \sum_{i \in [m]} \epsilon_t^{(i)} \leq \sum_{t=1}^{\tau} \sum_{i \in [m]} \epsilon_t^{(i)},$$

concludes the proof. $\qquad \square$

## C  Omitted Proofs for OLRC with Full Feedback

In this section, we provide the results and the omitted proofs for the OLRC with *full feedback* setting.

### C.1  Theoretical Guarantees of Algorithm 2

We start by providing a lower bound to the expected rewards attained by Algorithm 2.

**Lemma C.1.** *Algorithm 2, when instantiated in the full feedback setting with a primal regret minimizer which attains a regret upper bound $R_T^P$ and a dual regret minimizer which attains a regret upper bound $R_T^D$ for all $t \in [T]$, guarantees the following bound:*

$$\sum_{t=1}^{\tau} \underset{\boldsymbol{x} \sim \boldsymbol{\xi}_t}{\mathbb{E}} \left[ f_t(\boldsymbol{x}) \right] \geq \sup_{\boldsymbol{\xi} \in \Xi} \sum_{t=1}^{\tau} \left[ \underset{\boldsymbol{x} \sim \boldsymbol{\xi}}{\mathbb{E}} \left[ f_t(\boldsymbol{x}) \right] + \sum_{i \in [m]} \boldsymbol{\lambda}_t[i] \cdot \left( B_t^{(i)} - \underset{\boldsymbol{x} \sim \boldsymbol{\xi}}{\mathbb{E}} \left[ c_t(\boldsymbol{x})[i] \right] \right) \right]$$

$$- \sum_{t=1}^{\tau} \sum_{i \in [m]} \boldsymbol{\lambda}[i] \cdot \left( B_t^{(i)} - \underset{\boldsymbol{x} \sim \boldsymbol{\xi}_t}{\mathbb{E}} \left[ c_t(\boldsymbol{x})[i] \right] \right) - \frac{2}{\rho_{\min}} R_{\tau}^D - \left( 1 + \frac{2}{\rho_{\min}} \right) R_{\tau}^P,$$

*where $\boldsymbol{\lambda} \in \mathcal{L}$ is an arbitrary Lagrange multiplier and $\tau$ is the stopping time of the algorithm.*

*Proof.* In the following proof, we will refer to the stopping time of Algorithm 2 as $\tau$. Notice that there are two possible stopping criterion for Algorithm 2: $(i)$ the budget is depleted before reaching $T$, that is, at time $\tau < T$, $(ii)$ the algorithm stops at the end of the learning horizon, namely, $\tau = T$.

We first employ the no-regret property of the primal regret minimizer $\mathcal{R}^P$. Given that, it holds:

$$\sup_{\boldsymbol{\xi} \in \Xi} \sum_{t=1}^{\tau} \left[ \underset{\boldsymbol{x} \sim \boldsymbol{\xi}}{\mathbb{E}} \left[ f_t(\boldsymbol{x}) \right] - \sum_{i \in [m]} \boldsymbol{\lambda}_t[i] \cdot \underset{\boldsymbol{x} \sim \boldsymbol{\xi}}{\mathbb{E}} \left[ c_t(\boldsymbol{x})[i] \right] \right] -$$

$$\sum_{t=1}^{\tau} \left[ \underset{\boldsymbol{x} \sim \boldsymbol{\xi}_t}{\mathbb{E}} \left[ f_t(\boldsymbol{x}) \right] - \sum_{i \in [m]} \boldsymbol{\lambda}_t[i] \cdot \underset{\boldsymbol{x} \sim \boldsymbol{\xi}_t}{\mathbb{E}} \left[ c_t(\boldsymbol{x})[i] \right] \right] \leq \left( 1 + \frac{2}{\rho_{\min}} \right) R_{\tau}^P, \quad (7)$$

where the $(1 + 2/\rho_{\min})$ factor is the dependence on the payoffs range given as feedback to the primal regret minimizer.

Thus, we can rearrange Equation (7) to obtain the following lower bound the expected reward attained by Algorithm 2:

$$\sum_{t=1}^{\tau} \mathop{\mathbb{E}}_{\boldsymbol{x} \sim \boldsymbol{\xi}_t} [f_t(\boldsymbol{x})] \geq \sup_{\boldsymbol{\xi} \in \Xi} \sum_{t=1}^{\tau} \left[ \mathop{\mathbb{E}}_{\boldsymbol{x} \sim \boldsymbol{\xi}} [f_t(\boldsymbol{x})] - \sum_{i \in [m]} \boldsymbol{\lambda}_t[i] \cdot \mathop{\mathbb{E}}_{\boldsymbol{x} \sim \boldsymbol{\xi}} [c_t(\boldsymbol{x})[i]] \right]$$

$$+ \sum_{t=1}^{\tau} \sum_{i \in [m]} \boldsymbol{\lambda}_t[i] \cdot \mathop{\mathbb{E}}_{\boldsymbol{x} \sim \boldsymbol{\xi}_t} [c_t(\boldsymbol{x})[i]] - \left(1 + \frac{2}{\rho_{\min}}\right) R_\tau^P. \tag{8}$$

Thus, we make a similar reasoning for the dual regret minimizer. Specifically, we have, for any Lagrange multiplier $\boldsymbol{\lambda} \in \mathcal{L}$ the following bound:

$$\sum_{t=1}^{\tau} \sum_{i \in [m]} \boldsymbol{\lambda}_t[i] \cdot \left( B_t^{(i)} - \mathop{\mathbb{E}}_{\boldsymbol{x} \sim \boldsymbol{\xi}_t} [c_t(\boldsymbol{x})[i]] \right) - \sum_{t=1}^{\tau} \sum_{i \in [m]} \boldsymbol{\lambda}[i] \cdot \left( B_t^{(i)} - \mathop{\mathbb{E}}_{\boldsymbol{x} \sim \boldsymbol{\xi}_t} [c_t(\boldsymbol{x})[i]] \right) \leq \frac{2}{\rho_{\min}} R_\tau^D,$$

which in turn implies:

$$\sum_{t=1}^{\tau} \sum_{i \in [m]} \boldsymbol{\lambda}_t[i] \cdot \mathop{\mathbb{E}}_{\boldsymbol{x} \sim \boldsymbol{\xi}_t} [c_t(\boldsymbol{x})[i]] \geq$$

$$\sum_{t=1}^{\tau} \sum_{i \in [m]} \boldsymbol{\lambda}_t[i] \cdot B_t^{(i)} - \sum_{t=1}^{\tau} \sum_{i \in [m]} \boldsymbol{\lambda}[i] \cdot \left( B_t^{(i)} - \mathop{\mathbb{E}}_{\boldsymbol{x} \sim \boldsymbol{\xi}_t} [c_t(\boldsymbol{x})[i]] \right) - \frac{2}{\rho_{\min}} R_\tau^D, \tag{9}$$

where the $2/\rho_{\min}$ factor follows from the bound on the payoffs range of the dual regret minimizer.

We then substitute Equation (9) in Equation (8) to obtain:

$$\sum_{t=1}^{\tau} \mathop{\mathbb{E}}_{\boldsymbol{x} \sim \boldsymbol{\xi}_t} [f_t(\boldsymbol{x})] \geq \sup_{\boldsymbol{\xi} \in \Xi} \sum_{t=1}^{\tau} \left[ \mathop{\mathbb{E}}_{\boldsymbol{x} \sim \boldsymbol{\xi}} [f_t(\boldsymbol{x})] + \sum_{i \in [m]} \boldsymbol{\lambda}_t[i] \cdot \left( B_t^{(i)} - \mathop{\mathbb{E}}_{\boldsymbol{x} \sim \boldsymbol{\xi}} [c_t(\boldsymbol{x})[i]] \right) \right]$$

$$- \sum_{t=1}^{\tau} \sum_{i \in [m]} \boldsymbol{\lambda}[i] \cdot \left( B_t^{(i)} - \mathop{\mathbb{E}}_{\boldsymbol{x} \sim \boldsymbol{\xi}_t} [c_t(\boldsymbol{x})[i]] \right) - \frac{2}{\rho_{\min}} R_\tau^D - \left(1 + \frac{2}{\rho_{\min}}\right) R_\tau^P.$$

This concludes the proof. □

Thus, we refine the lower bound to the expected rewards by means of the following lemma.

**Lemma 4.1.** *For any $\delta \in (0, 1)$, Algorithm 2 instantiated in the full feedback setting with a primal regret minimizer which attains a regret upper bound $R_T^P$ and a dual regret minimizer which attains a regret upper bound $R_T^D$, guarantees the following bound, with probability at least $1 - \delta$:*
$\sum_{t=1}^{T} \mathbb{E}_{\boldsymbol{x} \sim \boldsymbol{\xi}_t} [f_t(\boldsymbol{x})] \geq \text{OPT}_{\mathcal{H}} - (T - \tau) - (4 + 4 \max_{\boldsymbol{\lambda} \in \mathcal{L}} \|\boldsymbol{\lambda}\|_1) \sqrt{2\tau \ln \frac{T}{\delta}} - \sum_{t=1}^{\tau} \sum_{i \in [m]} \boldsymbol{\lambda}[i] \cdot$
$\left( B_t^{(i)} - \mathbb{E}_{\boldsymbol{x} \sim \boldsymbol{\xi}_t} [c_t(\boldsymbol{x})[i]] \right) - \frac{2}{\rho_{\min}} R_\tau^D - \left(1 + \frac{2}{\rho_{\min}}\right) R_\tau^P$, *where $\boldsymbol{\lambda} \in \mathcal{L}$ is an arbitrary Lagrange multiplier and $\tau$ is the stopping time of the algorithm.*

*Proof.* We first employ Lemma C.1 to obtain:

$$\sum_{t=1}^{\tau} \mathop{\mathbb{E}}_{\boldsymbol{x} \sim \boldsymbol{\xi}_t} [f_t(\boldsymbol{x})] \geq \sup_{\boldsymbol{\xi} \in \Xi} \sum_{t=1}^{\tau} \left[ \mathop{\mathbb{E}}_{\boldsymbol{x} \sim \boldsymbol{\xi}} [f_t(\boldsymbol{x})] + \sum_{i \in [m]} \boldsymbol{\lambda}_t[i] \cdot \left( B_t^{(i)} - \mathop{\mathbb{E}}_{\boldsymbol{x} \sim \boldsymbol{\xi}} [c_t(\boldsymbol{x})[i]] \right) \right]$$

$$- \sum_{t=1}^{\tau} \sum_{i \in [m]} \boldsymbol{\lambda}[i] \cdot \left( B_t^{(i)} - \mathop{\mathbb{E}}_{\boldsymbol{x} \sim \boldsymbol{\xi}_t} [c_t(\boldsymbol{x})[i]] \right) - \frac{2}{\rho_{\min}} R_\tau^D - \left(1 + \frac{2}{\rho_{\min}}\right) R_\tau^P.$$

To conclude the proof, we focus on bounding the term:

$$\sup_{\boldsymbol{\xi} \in \Xi} \sum_{t=1}^{\tau} \left[ \mathbb{E}_{\boldsymbol{x} \sim \boldsymbol{\xi}} [f_t(\boldsymbol{x})] + \sum_{i \in [m]} \boldsymbol{\lambda}_t[i] \cdot \left( B_t^{(i)} - \mathbb{E}_{\boldsymbol{x} \sim \boldsymbol{\xi}} [c_t(\boldsymbol{x})[i]] \right) \right].$$

First we define $\Xi^\circ$ as the set which encompasses all safe strategy mixtures during the learning dynamic. Specifically, we let $\Xi^\circ := \{ \boldsymbol{\xi} \in \Xi : \mathbb{E}_{\boldsymbol{x} \sim \boldsymbol{\xi}} [\bar{c}_t(x)] \leq B_t^{(i)}, \forall i \in [m], t \in [T] \}$. Now, notice that by definition of Problem (2), there exists a strategy $\boldsymbol{\xi}^* \in \Xi^\circ$ such that $\sum_{t=1}^{T} \mathbb{E}_{\boldsymbol{x} \sim \boldsymbol{\xi}^*} \left[ \bar{f}_t(\boldsymbol{x}) \right] \geq \text{OPT}_{\mathcal{H}} - \gamma$, for all $\gamma > 0$. In the rest of the paper, we omit the factor $\gamma$, since it can be chosen arbitrarily small, thus being a negligible factor in the regret bound. Moreover, since the strategy belongs to $\Xi^\circ$, thus, it satisfies the budget constraints imposed by the spending plan, we additionally have:

$$\sum_{t=1}^{\tau} \sum_{i \in [m]} \boldsymbol{\lambda}_t[i] \cdot \left( B_t^{(i)} - \mathbb{E}_{\boldsymbol{x} \sim \boldsymbol{\xi}^*} [\bar{c}_t(\boldsymbol{x})[i]] \right) \geq 0.$$

Thus, we can conclude that:

$$\sup_{\boldsymbol{\xi} \in \Xi} \sum_{t=1}^{\tau} \left[ \mathbb{E}_{\boldsymbol{x} \sim \boldsymbol{\xi}} [f_t(\boldsymbol{x})] + \sum_{i \in [m]} \boldsymbol{\lambda}_t[i] \cdot \left( B_t^{(i)} - \mathbb{E}_{\boldsymbol{x} \sim \boldsymbol{\xi}} [c_t(\boldsymbol{x})[i]] \right) \right]$$

$$\geq \sum_{t=1}^{\tau} \left[ \mathbb{E}_{\boldsymbol{x} \sim \boldsymbol{\xi}^*} [f_t(\boldsymbol{x})] + \sum_{i \in [m]} \boldsymbol{\lambda}_t[i] \cdot \left( B_t^{(i)} - \mathbb{E}_{\boldsymbol{x} \sim \boldsymbol{\xi}^*} [c_t(\boldsymbol{x})[i]] \right) \right]$$

$$\geq \sum_{t=1}^{\tau} \left[ \mathbb{E}_{\boldsymbol{x} \sim \boldsymbol{\xi}^*} \left[ \bar{f}_t(\boldsymbol{x}) \right] + \sum_{i \in [m]} \boldsymbol{\lambda}_t[i] \cdot \left( B_t^{(i)} - \mathbb{E}_{\boldsymbol{x} \sim \boldsymbol{\xi}^*} [\bar{c}_t(\boldsymbol{x})[i]] \right) \right] - \left( 4 + 4 \max_{\boldsymbol{\lambda} \in \mathcal{L}} \|\boldsymbol{\lambda}\|_1 \right) \sqrt{2\tau \ln \frac{T}{\delta}}$$

$$\geq \sum_{t=1}^{\tau} \mathbb{E}_{\boldsymbol{x} \sim \boldsymbol{\xi}^*} \left[ \bar{f}_t(\boldsymbol{x}) \right] - \left( 4 + 4 \max_{\boldsymbol{\lambda} \in \mathcal{L}} \|\boldsymbol{\lambda}\|_1 \right) \sqrt{2\tau \ln \frac{T}{\delta}}$$

$$\geq \text{OPT}_{\mathcal{H}} - (T - \tau) - \left( 4 + 4 \max_{\boldsymbol{\lambda} \in \mathcal{L}} \|\boldsymbol{\lambda}\|_1 \right) \sqrt{2\tau \ln \frac{T}{\delta}},$$

where the second inequality holds with probability at least $1 - \delta$ by Lemma E.2.

Noticing that by the update of Algorithm 2, it holds:

$$\sum_{t=1}^{\tau} \mathbb{E}_{\boldsymbol{x} \sim \boldsymbol{\xi}_t} [f_t(\boldsymbol{x})] = \sum_{t=1}^{T} \mathbb{E}_{\boldsymbol{x} \sim \boldsymbol{\xi}_t} [f_t(\boldsymbol{x})],$$

concludes the proof. $\qquad \square$

We are now ready to prove the final regret bound.

**Theorem 4.2.** *For any $\delta \in (0, 1)$, Algorithm 2, when instantiated in the full feedback setting with a primal regret minimizer which attains a regret upper bound $R_T^P$ and a dual regret minimizer which attains a regret upper bound $R_T^D$, guarantees, with probability at least $1 - 2\delta$, the following regret bound $R_T \leq 1 + \frac{1}{\rho_{\min}} + \frac{2}{\rho_{\min}} R_T^D + \left( 1 + \frac{2}{\rho_{\min}} \right) R_T^P + \left( 8 + \frac{8}{\rho_{\min}} \right) \sqrt{2T \ln \frac{T}{\delta}}.$*

*Proof.* We employ Lemma 4.1 to recover the following bound, which holds with probability at least $1 - \delta$:

$$\sum_{t=1}^{\tau} \mathbb{E}_{\boldsymbol{x} \sim \boldsymbol{\xi}_t} [f_t(\boldsymbol{x})] \geq \text{OPT}_{\mathcal{H}} - (T - \tau) - \left( 4 + 4 \max_{\boldsymbol{\lambda} \in \mathcal{L}} \|\boldsymbol{\lambda}\|_1 \right) \sqrt{2\tau \ln \frac{T}{\delta}}$$

$$- \sum_{t=1}^{\tau} \sum_{i \in [m]} \boldsymbol{\lambda}[i] \cdot \left( B_t^{(i)} - \mathbb{E}_{\boldsymbol{x} \sim \boldsymbol{\xi}_t} [c_t(\boldsymbol{x})[i]] \right) - \frac{2}{\rho_{\min}} R_\tau^D - \left( 1 + \frac{2}{\rho_{\min}} \right) R_\tau^P.$$

Thus, we apply Lemma E.1 to obtain:

$$\sum_{t=1}^{\tau} f_t(\boldsymbol{x}_t) \geq \text{OPT}_{\mathcal{H}} - (T - \tau) - \left(4 + 4\max_{\boldsymbol{\lambda} \in \mathcal{L}} \|\boldsymbol{\lambda}\|_1\right)\sqrt{2\tau \ln\frac{T}{\delta}}$$

$$- \sum_{t=1}^{\tau}\sum_{i \in [m]} \boldsymbol{\lambda}[i] \cdot \left(B_t^{(i)} - c_t(\boldsymbol{x}_t)[i]\right) - \frac{2}{\rho_{\min}} R_\tau^D - \left(1 + \frac{2}{\rho_{\min}}\right) R_\tau^P - (4 + 4\|\boldsymbol{\lambda}\|_1)\sqrt{2\tau \ln\frac{T}{\delta}},$$

which holds with probability at least $1 - 2\delta$ by Union Bound, form which we have, with same probability, the following regret bound:

$$R_T \leq (T - \tau) + \sum_{t=1}^{\tau}\sum_{i \in [m]} \boldsymbol{\lambda}[i] \cdot \left(B_t^{(i)} - c_t(\boldsymbol{x}_t)[i]\right) + \frac{2}{\rho_{\min}} R_\tau^D$$

$$+ \left(1 + \frac{2}{\rho_{\min}}\right) R_\tau^P + \left(8 + 8\max_{\boldsymbol{\lambda} \in \mathcal{L}} \|\boldsymbol{\lambda}\|_1\right)\sqrt{2\tau \ln\frac{T}{\delta}}.$$

We finally focus on bounding the term $\sum_{t=1}^{\tau}\sum_{i \in [m]} \boldsymbol{\lambda}[i] \cdot \left(B_t^{(i)} - c_t(\boldsymbol{x}_t)[i]\right)$. We split the analysis in two cases: $(i)$ $\tau = T$ and $(ii)$ $\tau < T$.

**Bound for case $(i)$.** When $\tau = T$, we select the Lagrange multiplier $\boldsymbol{\lambda}$ as the zero vector $\boldsymbol{0}$ to get the following regret bound:

$$R_T \leq \frac{2}{\rho_{\min}} R_\tau^D + \left(1 + \frac{2}{\rho_{\min}}\right) R_\tau^P + \left(8 + 8\max_{\boldsymbol{\lambda} \in \mathcal{L}} \|\boldsymbol{\lambda}\|_1\right)\sqrt{2\tau \ln\frac{T}{\delta}},$$

which holds with probability at least $1 - 2\delta$.

**Bound for case $(ii)$.** In such a scenario, the budget has been depleted before the spending plans suggestions. Hence, the following holds for a resource $i^* \in [m]$:

$$\sum_{t=1}^{\tau} c_t(\boldsymbol{x}_t)[i^*] + 1 \geq B \geq \sum_{t=1}^{T} B_t^{(i^*)}.$$

Thus, we can conclude that:

$$\sum_{t=1}^{\tau}\sum_{i \in [m]} \boldsymbol{\lambda}[i] \cdot \left(B_t^{(i)} - c_t(\boldsymbol{x}_t)[i]\right) = \sum_{t=1}^{\tau} \frac{1}{\rho_{\min}} \left(B_t^{(i^*)} - c_t(\boldsymbol{x}_t)[i^*]\right) \tag{10}$$

$$\leq \frac{1}{\rho_{\min}} \left(\sum_{t=1}^{\tau} B_t^{(i^*)} - \sum_{t=1}^{T} B_t^{(i^*)}\right) + \frac{1}{\rho_{\min}}$$

$$= -\frac{1}{\rho_{\min}} \sum_{t=\tau+1}^{T} B_t^{(i^*)} + \frac{1}{\rho_{\min}}$$

$$\leq -\frac{1}{\rho_{\min}} \rho_{\min}(T - \tau - 1) + \frac{1}{\rho_{\min}}$$

$$= -(T - \tau - 1) + \frac{1}{\rho_{\min}},$$

where Equation (10) holds selecting $\boldsymbol{\lambda}$ s.t. $\boldsymbol{\lambda}[i^*] = \frac{1}{\rho_{\min}}$ and $\boldsymbol{\lambda}[i] = 0$ for all others $i \in [m]$. Thus, substituting the result in the regret bound, we obtain:

$$R_T \leq 1 + \frac{1}{\rho_{\min}} + \frac{2}{\rho_{\min}} R_\tau^D + \left(1 + \frac{2}{\rho_{\min}}\right) R_\tau^P + \left(8 + 8\max_{\boldsymbol{\lambda} \in \mathcal{L}} \|\boldsymbol{\lambda}\|_1\right)\sqrt{2\tau \ln\frac{T}{\delta}},$$

which holds with probability at least $1 - 2\delta$.

To get the final regret bound we notice the following trivial upper bounds:

$$R_\tau^D \leq R_T^D, \ R_\tau^P \leq R_T^P, \ \left(8 + 8\max_{\boldsymbol{\lambda} \in \mathcal{L}} \|\boldsymbol{\lambda}\|_1\right)\sqrt{2\tau \ln\frac{T}{\delta}} \leq \left(8 + \frac{8}{\rho_{\min}}\right)\sqrt{2T \ln\frac{T}{\delta}}.$$

This concludes the proof. $\qquad\square$

### C.1.1 Robustness to Baselines Deviating from the Spending Plan

In this section, we first provide the definition of the *fixed* baseline which deviates from the spending plan. Specifically, we first define the *fixed* optimal solution parametrized given the errors $\epsilon_t^{(i)}$ specified for all $t \in [T], i \in [m]$, by means of the following optimization problem:

$$
\text{OPT}_{\mathcal{H}}(\epsilon_t) \coloneqq \begin{cases} \sup_{\boldsymbol{\xi} \in \Xi} & \mathbb{E}_{\boldsymbol{x} \sim \boldsymbol{\xi}} \left[ \sum_{t=1}^{T} \bar{f}_t(\boldsymbol{x}) \right] \\ \text{s.t.} & \mathbb{E}_{\boldsymbol{x} \sim \boldsymbol{\xi}} \left[ \bar{c}_t(\boldsymbol{x})[i] \right] \leq B_t^{(i)} + \epsilon_t^{(i)} \ \ \forall i \in [m], \forall t \in [T] \\ & \sum_{t=1}^{T} \mathbb{E}_{\boldsymbol{x} \sim \boldsymbol{\xi}} \left[ \bar{c}_t(\boldsymbol{x})[i] \right] \leq B \ \ \forall i \in [m] \end{cases} \cdot \tag{11}
$$

Problem (11) computes the expected value attained by the optimal *fixed* strategy mixture that satisfies the *spending plans* at each round $t \in [T]$, up to error terms $\epsilon_t^{(i)} \geq 0$, defined for all $i \in [m]$ and $t \in [T]$. Similarly to the dynamic baseline, we do *not* make any assumption on the error terms $\epsilon_t^{(i)}$. Nonetheless, the performance of our algorithms will smoothly degrade with the magnitude of these errors. Moreover, the errors must allow Problem (11) to be feasible. We remark that the last group of constraints in Problem (11) ensures that the error terms do not allow the optimal solution to violate the general budget constraint. Observe that when $\epsilon_t^{(i)} = 0$ for all $i \in [m]$ and $t \in [T]$—meaning that the spending plan is strictly followed by the optimal solution—the general budget constraint is satisfied by the definition of the spending plan.

Similarly, we define the following notion of cumulative *static* regret $R_T(\epsilon_t) \coloneqq \text{OPT}_{\mathcal{H}}(\epsilon_t) - \sum_{t=1}^{T} f_t(\boldsymbol{x}_t)$, which simply compares the rewards attained by the algorithm with respect to the optimal *fixed* solution which follows the spending plans recommendations up to the errors.

We provide the regret of Algorithm 2 with respect to a baseline which deviates from the spending plan.

**Theorem C.2.** *For any $\delta \in (0, 1)$, Algorithm 2, when instantiated in the full feedback setting with a primal regret minimizer which attains a regret upper bound $R_T^P$ and a dual regret minimizer which attains a regret upper bound $R_T^D$, guarantees, with probability at least $1 - 2\delta$, the following regret bound:*

$$
R_T(\epsilon_t) \leq 1 + \frac{1}{\rho_{\min}} + \frac{2}{\rho_{\min}} R_T^D + \left( 1 + \frac{2}{\rho_{\min}} \right) R_T^P + \left( 8 + \frac{8}{\rho_{\min}} \right) \sqrt{2T \ln \frac{T}{\delta}} + \frac{1}{\rho_{\min}} \sum_{t=1}^{T} \sum_{i \in [m]} \epsilon_t^{(i)}.
$$

*Proof.* Similarly to Lemma 4.1, we employ Lemma C.1 to obtain:

$$
\sum_{t=1}^{\tau} \mathbb{E}_{\boldsymbol{x} \sim \boldsymbol{\xi}_t} [f_t(\boldsymbol{x})] \geq \sup_{\boldsymbol{\xi} \in \Xi} \sum_{t=1}^{\tau} \left[ \mathbb{E}_{\boldsymbol{x} \sim \boldsymbol{\xi}} [f_t(\boldsymbol{x})] + \sum_{i \in [m]} \boldsymbol{\lambda}_t[i] \cdot \left( B_t^{(i)} - \mathbb{E}_{\boldsymbol{x} \sim \boldsymbol{\xi}} [c_t(\boldsymbol{x})[i]] \right) \right]
$$
$$
- \sum_{t=1}^{\tau} \sum_{i \in [m]} \boldsymbol{\lambda}[i] \cdot \left( B_t^{(i)} - \mathbb{E}_{\boldsymbol{x} \sim \boldsymbol{\xi}_t} [c_t(\boldsymbol{x})[i]] \right) - \frac{2}{\rho_{\min}} R_\tau^D - \left( 1 + \frac{2}{\rho_{\min}} \right) R_\tau^P.
$$

We now bound the term:

$$
\sup_{\boldsymbol{\xi} \in \Xi} \sum_{t=1}^{\tau} \left[ \mathbb{E}_{\boldsymbol{x} \sim \boldsymbol{\xi}} [f_t(\boldsymbol{x})] + \sum_{i \in [m]} \boldsymbol{\lambda}_t[i] \cdot \left( B_t^{(i)} - \mathbb{E}_{\boldsymbol{x} \sim \boldsymbol{\xi}} [c_t(\boldsymbol{x})[i]] \right) \right],
$$

which is done taking into account the possible error. First we define $\Xi^\circ$ as the set which encompasses all safe strategy mixtures during the learning dynamic. Specifically, we let $\Xi^\circ \coloneqq \{ \boldsymbol{\xi} \in \Xi : \mathbb{E}_{\boldsymbol{x} \sim \boldsymbol{\xi}} [\bar{c}_t(x)] \leq B_t^{(i)} - \epsilon_t^{(i)}, \forall i \in [m], t \in [T] \}$. Now, notice that by definition of Problem (11), there exists a strategy $\boldsymbol{\xi}^* \in \Xi^\circ$ such that $\sum_{t=1}^{T} \mathbb{E}_{\boldsymbol{x} \sim \boldsymbol{\xi}^*} [\bar{f}_t(\boldsymbol{x})] \geq \text{OPT}_{\mathcal{H}}(\epsilon_t) - \gamma$, for all $\gamma > 0$. In the rest of the paper, we omit the factor $\gamma$, since it can be chosen arbitrarily small, thus being a negligible factor in the regret bound. Moreover, since the strategy belongs to $\Xi^\circ$, thus, it satisfies the budget constraints imposed by the spending plan, up to the error term, we additionally have:

$$
\sum_{t=1}^{\tau} \sum_{i \in [m]} \boldsymbol{\lambda}_t[i] \cdot \left( B_t^{(i)} - \mathbb{E}_{\boldsymbol{x} \sim \boldsymbol{\xi}^*} [\bar{c}_t(\boldsymbol{x})[i]] \right) \geq - \frac{1}{\rho_{\min}} \sum_{t=1}^{\tau} \sum_{i \in [m]} \epsilon_t^{(i)}.
$$

Thus, we can conclude that:

$$\sup_{\boldsymbol{\xi}\in\Xi}\sum_{t=1}^{\tau}\left[\mathop{\mathbb{E}}_{\boldsymbol{x}\sim\boldsymbol{\xi}}[f_t(\boldsymbol{x})]+\sum_{i\in[m]}\boldsymbol{\lambda}_t[i]\cdot\left(B_t^{(i)}-\mathop{\mathbb{E}}_{\boldsymbol{x}\sim\boldsymbol{\xi}}[c_t(\boldsymbol{x})[i]]\right)\right]$$

$$\geq\sum_{t=1}^{\tau}\left[\mathop{\mathbb{E}}_{\boldsymbol{x}\sim\boldsymbol{\xi}^*}[f_t(\boldsymbol{x})]+\sum_{i\in[m]}\boldsymbol{\lambda}_t[i]\cdot\left(B_t^{(i)}-\mathop{\mathbb{E}}_{\boldsymbol{x}\sim\boldsymbol{\xi}^*}[c_t(\boldsymbol{x})[i]]\right)\right]$$

$$\geq\sum_{t=1}^{\tau}\left[\mathop{\mathbb{E}}_{\boldsymbol{x}\sim\boldsymbol{\xi}^*}[\bar{f}_t(\boldsymbol{x})]+\sum_{i\in[m]}\boldsymbol{\lambda}_t[i]\cdot\left(B_t^{(i)}-\mathop{\mathbb{E}}_{\boldsymbol{x}\sim\boldsymbol{\xi}^*}[\bar{c}_t(\boldsymbol{x})[i]]\right)\right]-\left(4+4\max_{\boldsymbol{\lambda}\in\mathcal{L}}\|\boldsymbol{\lambda}\|_1\right)\sqrt{2\tau\ln\frac{T}{\delta}}$$

$$\geq\sum_{t=1}^{\tau}\mathop{\mathbb{E}}_{\boldsymbol{x}\sim\boldsymbol{\xi}^*}[\bar{f}_t(\boldsymbol{x})]-\frac{1}{\rho_{\min}}\sum_{t=1}^{\tau}\sum_{i\in[m]}\epsilon_t^{(i)}-\left(4+4\max_{\boldsymbol{\lambda}\in\mathcal{L}}\|\boldsymbol{\lambda}\|_1\right)\sqrt{2\tau\ln\frac{T}{\delta}}$$

$$\geq\mathrm{OPT}_{\mathcal{H}}(\epsilon_t)-(T-\tau)-\frac{1}{\rho_{\min}}\sum_{t=1}^{\tau}\sum_{i\in[m]}\epsilon_t^{(i)}-\left(4+4\max_{\boldsymbol{\lambda}\in\mathcal{L}}\|\boldsymbol{\lambda}\|_1\right)\sqrt{2\tau\ln\frac{T}{\delta}},$$

where the second inequality holds with probability at least $1-\delta$ by Lemma E.2. Thus, we notice that by the update of Algorithm 2, it holds:

$$\sum_{t=1}^{\tau}\mathop{\mathbb{E}}_{\boldsymbol{x}\sim\boldsymbol{\xi}_t}[f_t(\boldsymbol{x})]=\sum_{t=1}^{T}\mathop{\mathbb{E}}_{\boldsymbol{x}\sim\boldsymbol{\xi}_t}[f_t(\boldsymbol{x})].$$

The final result follows from the same analysis of Theorem 4.2, after noticing that:

$$\sum_{t=1}^{\tau}\sum_{i\in[m]}\epsilon_t^{(i)}\leq\sum_{t=1}^{T}\sum_{i\in[m]}\epsilon_t^{(i)}.$$

This concludes the proof. □

## C.2 Theoretical Guarantees of Algorithm 4

In this section, we present the results attained by the meta procedure provided in Algorithm 4.

### C.2.1 Algorithm

We first provide the algorithm for the *full feedback* case.

---
**Algorithm 4** Meta-algorithm for arbitrarily small $\rho_{\min}$ and *full feedback*

---
**Require:** Horizon $T$, budget $B$, spending plans $\mathcal{B}_T^{(i)}$ for all $i\in[m]$, primal regret minimizer $\mathcal{R}^P$ (*full feedback*), dual regret minimizer $\mathcal{R}^D$ (*full feedback*)
 1: Define $\hat{\rho}:=\rho/T^{1/4}$
 2: Define $\overline{B}_t^{(i)}:=B_t^{(i)}\left(1-T^{-1/4}\right)\ \ \forall t\in[T],i\in[m]$
 3: Run Algorithm 2 for *full feedback* with $\rho_{\min}\leftarrow\hat{\rho}$, $B_t^{(i)}\leftarrow\overline{B}_t^{(i)}$

---

### C.2.2 Analysis

In this section, we provide the analysis of Algorithm 4. We start by lower bounding the expected rewards attained by the meta procedure.

**Lemma C.3.** *Algorithm 4, when instantiated with a primal regret minimizer which attains a regret upper bound $R_T^P$ and a dual regret minimizer which attains a regret upper bound $R_T^D$, guarantees the following bound:*

$$\sum_{t=1}^{\tau}\mathop{\mathbb{E}}_{\boldsymbol{x}\sim\boldsymbol{\xi}_t}[f_t(\boldsymbol{x})]\geq\sup_{\boldsymbol{\xi}\in\Xi}\sum_{t=1}^{\tau}\left[\mathop{\mathbb{E}}_{\boldsymbol{x}\sim\boldsymbol{\xi}}[f_t(\boldsymbol{x})]+\sum_{i\in[m]}\boldsymbol{\lambda}_t[i]\cdot\left(\overline{B}_t^{(i)}-\mathop{\mathbb{E}}_{\boldsymbol{x}\sim\boldsymbol{\xi}}[c_t(\boldsymbol{x})[i]]\right)\right]$$

$$-\sum_{t=1}^{\tau}\sum_{i\in[m]}\boldsymbol{\lambda}[i]\cdot\left(\overline{B}_t^{(i)}-\mathbb{E}_{\boldsymbol{x}\sim\boldsymbol{\xi}_t}[c_t(\boldsymbol{x})[i]]\right)-\frac{2}{\hat{\rho}}R_\tau^D-\left(1+\frac{2}{\hat{\rho}}\right)R_\tau^P,$$

*where $\boldsymbol{\lambda}\in\mathcal{L}$ is an arbitrary Lagrange multiplier and $\tau$ is the stopping time of the algorithm.*

*Proof.* The proof is analogous to the one of Lemma C.1 after substituting $\rho_{\min}$ with $\hat{\rho}$ and $B_t^{(i)}$ with $\overline{B}_t^{(i)}$, for all $i\in[m], t\in[T]$. $\qquad\square$

We refine the previous lower bound as follows.

**Lemma C.4.** *For any $\delta\in(0,1)$, Algorithm 4, when instantiated with a primal regret minimizer which attains a regret upper bound $R_T^P$ and a dual regret minimizer which attains a regret upper bound $R_T^D$, guarantees the following bound, with probability at least $1-\delta$:*

$$\sum_{t=1}^{\tau}\mathbb{E}_{\boldsymbol{x}\sim\boldsymbol{\xi}_t}[f_t(\boldsymbol{x})]\geq\mathrm{OPT}_{\mathcal{H}}-(T-\tau)-T^{\frac{3}{4}}-\left(4+\frac{4T^{\frac{1}{4}}}{\rho}\right)\sqrt{2\tau\ln\frac{T}{\delta}}-\frac{2T^{\frac{1}{4}}}{\rho}R_\tau^D-\left(1+\frac{2T^{\frac{1}{4}}}{\rho}\right)R_\tau^P,$$

*where $\tau$ is the stopping time of the algorithm.*

*Proof.* We first employ Lemma C.3 to obtain:

$$\sum_{t=1}^{\tau}\mathbb{E}_{\boldsymbol{x}\sim\boldsymbol{\xi}_t}[f_t(\boldsymbol{x})]\geq\sup_{\boldsymbol{\xi}\in\Xi}\sum_{t=1}^{\tau}\left[\mathbb{E}_{\boldsymbol{x}\sim\boldsymbol{\xi}}[f_t(\boldsymbol{x})]+\sum_{i\in[m]}\boldsymbol{\lambda}_t[i]\cdot\left(\overline{B}_t^{(i)}-\mathbb{E}_{\boldsymbol{x}\sim\boldsymbol{\xi}}[c_t(\boldsymbol{x})[i]]\right)\right]$$
$$-\sum_{t=1}^{\tau}\sum_{i\in[m]}\boldsymbol{\lambda}[i]\cdot\left(\overline{B}_t^{(i)}-\mathbb{E}_{\boldsymbol{x}\sim\boldsymbol{\xi}_t}[c_t(\boldsymbol{x})[i]]\right)-\frac{2T^{\frac{1}{4}}}{\rho}R_\tau^D-\left(1+\frac{2T^{\frac{1}{4}}}{\rho}\right)R_\tau^P.$$

Thus we select the Lagrange variable as $\boldsymbol{\lambda}=\boldsymbol{0}$ vector to establish the following bound:

$$\sum_{t=1}^{\tau}\mathbb{E}_{\boldsymbol{x}\sim\boldsymbol{\xi}_t}[f_t(\boldsymbol{x})]\geq\sup_{\boldsymbol{\xi}\in\Xi}\sum_{t=1}^{\tau}\left[\mathbb{E}_{\boldsymbol{x}\sim\boldsymbol{\xi}}[f_t(\boldsymbol{x})]+\sum_{i\in[m]}\boldsymbol{\lambda}_t[i]\cdot\left(\overline{B}_t^{(i)}-\mathbb{E}_{\boldsymbol{x}\sim\boldsymbol{\xi}}[c_t(\boldsymbol{x})[i]]\right)\right]$$
$$-\frac{2T^{\frac{1}{4}}}{\rho}R_\tau^D-\left(1+\frac{2T^{\frac{1}{4}}}{\rho}\right)R_\tau^P.$$

We now focus on bounding the term:

$$\sup_{\boldsymbol{\xi}\in\Xi}\sum_{t=1}^{\tau}\left[\mathbb{E}_{\boldsymbol{x}\sim\boldsymbol{\xi}}[f_t(\boldsymbol{x})]+\sum_{i\in[m]}\boldsymbol{\lambda}_t[i]\cdot\left(\overline{B}_t^{(i)}-\mathbb{E}_{\boldsymbol{x}\sim\boldsymbol{\xi}}[c_t(\boldsymbol{x})[i]]\right)\right].$$

Similarly to Lemma 4.1, we define $\Xi^\circ$ as the set which encompasses all safe strategy mixtures during the learning dynamic. Specifically, we let $\Xi^\circ\coloneqq\{\boldsymbol{\xi}\in\Xi:\mathbb{E}_{\boldsymbol{x}\sim\boldsymbol{\xi}}[\bar{c}_t(x)]\leq B_t^{(i)},\forall i\in[m],t\in[T]\}$. Now, notice that by definition of Problem (2), there exists a strategy $\boldsymbol{\xi}^*\in\Xi^\circ$ such that $\sum_{t=1}^{T}\mathbb{E}_{\boldsymbol{x}\sim\boldsymbol{\xi}^*}[\bar{f}_t(\boldsymbol{x})]\geq\mathrm{OPT}_{\mathcal{H}}-\gamma$, for all $\gamma>0$. In the rest of the paper, we omit the factor $\gamma$, since it can be chosen arbitrarily small, thus being a negligible factor in the regret bound. We then define the following strategy mixture $\boldsymbol{\xi}^\diamond$ as follows:

$$\boldsymbol{\xi}^\diamond\coloneqq\begin{cases}\boldsymbol{x}^\varnothing & \text{w.p. } 1/T^{1/4}\\\boldsymbol{\xi}^* & \text{w.p. } 1-1/T^{1/4}\end{cases}.$$

Thus, we first show that $\boldsymbol{\xi}^\diamond$ satisfies the per-round expected constraints defined by $\overline{B}_t^{(i)}$. Indeed it holds, for all $i\in[m]$:

$$\overline{B}_t^{(i)}-\mathbb{E}_{\boldsymbol{x}\sim\boldsymbol{\xi}^\diamond}[\bar{c}_t(\boldsymbol{x})]=\overline{B}_t^{(i)}-\left(1-\frac{1}{T^{1/4}}\right)\mathbb{E}_{\boldsymbol{x}\sim\boldsymbol{\xi}^*}[\bar{c}_t(\boldsymbol{x})]-\frac{1}{T^{1/4}}\mathbb{E}_{\boldsymbol{x}\sim\boldsymbol{\xi}^\varnothing}[\bar{c}_t(\boldsymbol{x})]$$
$$=\left(1-\frac{1}{T^{1/4}}\right)B_t^{(i)}-\left(1-\frac{1}{T^{1/4}}\right)\mathbb{E}_{\boldsymbol{x}\sim\boldsymbol{\xi}^*}[\bar{c}_t(\boldsymbol{x})]-\frac{1}{T^{1/4}}\mathbb{E}_{\boldsymbol{x}\sim\boldsymbol{\xi}^\varnothing}[\bar{c}_t(\boldsymbol{x})]$$

$$= \left(1 - \frac{1}{T^{1/4}}\right) B_t^{(i)} - \left(1 - \frac{1}{T^{1/4}}\right) \mathop{\mathbb{E}}_{\boldsymbol{x} \sim \boldsymbol{\xi}^*} [\bar{c}_t(\boldsymbol{x})]$$

$$\geq 0,$$

where we employed the definition of $\boldsymbol{\xi}^*$ and $\boldsymbol{\xi}^\varnothing$.

Thus, returning to the quantity of interest, it holds:

$$\sup_{\boldsymbol{\xi} \in \Xi} \sum_{t=1}^{\tau} \left[ \mathop{\mathbb{E}}_{\boldsymbol{x} \sim \boldsymbol{\xi}} [f_t(\boldsymbol{x})] + \sum_{i \in [m]} \boldsymbol{\lambda}_t[i] \cdot \left( \overline{B}_t^{(i)} - \mathop{\mathbb{E}}_{\boldsymbol{x} \sim \boldsymbol{\xi}} [c_t(\boldsymbol{x})[i]] \right) \right]$$

$$\geq \sum_{t=1}^{\tau} \left[ \mathop{\mathbb{E}}_{\boldsymbol{x} \sim \boldsymbol{\xi}^\diamond} [f_t(\boldsymbol{x})] + \sum_{i \in [m]} \boldsymbol{\lambda}_t[i] \cdot \left( \overline{B}_t^{(i)} - \mathop{\mathbb{E}}_{\boldsymbol{x} \sim \boldsymbol{\xi}^\diamond} [c_t(\boldsymbol{x})[i]] \right) \right]$$

$$\geq \sum_{t=1}^{\tau} \left[ \mathop{\mathbb{E}}_{\boldsymbol{x} \sim \boldsymbol{\xi}^\diamond} [\bar{f}_t(\boldsymbol{x})] + \sum_{i \in [m]} \boldsymbol{\lambda}_t[i] \cdot \left( \overline{B}_t^{(i)} - \mathop{\mathbb{E}}_{\boldsymbol{x} \sim \boldsymbol{\xi}^\diamond} [\bar{c}_t(\boldsymbol{x})[i]] \right) \right] - \left( 4 + \frac{4T^{1/4}}{\rho} \right) \sqrt{2\tau \ln \frac{T}{\delta}}$$

$$\geq \sum_{t=1}^{\tau} \left[ \mathop{\mathbb{E}}_{\boldsymbol{x} \sim \boldsymbol{\xi}^\diamond} [\bar{f}_t(\boldsymbol{x})] \right] - \left( 4 + \frac{4T^{1/4}}{\rho} \right) \sqrt{2\tau \ln \frac{T}{\delta}}$$

$$= \sum_{t=1}^{\tau} \left[ \left( 1 - \frac{1}{T^{1/4}} \right) \mathop{\mathbb{E}}_{\boldsymbol{x} \sim \boldsymbol{\xi}^*} [\bar{f}_t(\boldsymbol{x})] + \frac{1}{T^{1/4}} \mathop{\mathbb{E}}_{\boldsymbol{x} \sim \boldsymbol{\xi}^\varnothing} [\bar{f}_t(\boldsymbol{x})] \right] - \left( 4 + \frac{4T^{1/4}}{\rho} \right) \sqrt{2\tau \ln \frac{T}{\delta}}$$

$$= \sum_{t=1}^{\tau} \left[ \left( 1 - \frac{1}{T^{1/4}} \right) \mathop{\mathbb{E}}_{\boldsymbol{x} \sim \boldsymbol{\xi}^*} [\bar{f}_t(\boldsymbol{x})] \right] - \left( 4 + \frac{4T^{1/4}}{\rho} \right) \sqrt{2\tau \ln \frac{T}{\delta}}$$

$$\geq \sum_{t=1}^{\tau} \mathop{\mathbb{E}}_{\boldsymbol{x} \sim \boldsymbol{\xi}^*} [\bar{f}_t(\boldsymbol{x})] - T^{3/4} - \left( 4 + \frac{4T^{1/4}}{\rho} \right) \sqrt{2\tau \ln \frac{T}{\delta}}$$

$$\geq \mathrm{OPT}_{\mathcal{H}} - T^{3/4} - (T - \tau) - \left( 4 + \frac{4T^{1/4}}{\rho} \right) \sqrt{2\tau \ln \frac{T}{\delta}},$$

where the second inequality holds, with probability at least $1 - \delta$, by Lemma E.2 and upper bounding the Lagrangian multiplier with $T^{\frac{1}{4}}/\rho$. This concludes the proof. $\square$

Hence, we proceed upper bounding the difference between the horizon $T$ and the stopping time $\tau$. This is done by means of the following lemma.

**Lemma C.5.** *For any $\delta \in (0, 1)$, Algorithm 4, when instantiated with a primal regret minimizer which attains a regret upper bound $R_T^P$ and a dual regret minimizer which attains a regret upper bound $R_T^D$, guarantees the following bound with probability at least $1 - \delta$:*

$$T - \tau \leq \frac{14}{\rho} \left( \sqrt{\ln \frac{T}{\delta}} + \frac{R_T^P + R_T^D}{\sqrt{T}} \right) T^{\frac{3}{4}}.$$

*Proof.* Suppose by contradiction that $T - \tau > CT^{3/4}$, thus, it holds $\tau < T - CT^{3/4}$.

We proceed upper and lower bounding the value of the Lagrangian given as feedback to the primal regret minimizer. Thus, we employ the no-regret property of the primal regret minimizer $\mathcal{R}^P$. Given that, it holds:

$$\sum_{t=1}^{\tau} \left[ \mathop{\mathbb{E}}_{\boldsymbol{x} \sim \boldsymbol{\xi}_t} [f_t(\boldsymbol{x})] - \sum_{i \in [m]} \boldsymbol{\lambda}_t[i] \cdot \mathop{\mathbb{E}}_{\boldsymbol{x} \sim \boldsymbol{\xi}_t} [c_t(\boldsymbol{x})[i]] \right]$$

$$\geq \sum_{t=1}^{\tau} \left[ \mathop{\mathbb{E}}_{\boldsymbol{x} \sim \boldsymbol{\xi}^\varnothing} [f_t(\boldsymbol{x})] - \sum_{i \in [m]} \boldsymbol{\lambda}_t[i] \cdot \mathop{\mathbb{E}}_{\boldsymbol{x} \sim \boldsymbol{\xi}^\varnothing} [c_t(\boldsymbol{x})[i]] \right] - \left( 1 + \frac{2T^{1/4}}{\rho} \right) R_\tau^P$$

$$= -\left(1 + \frac{2T^{1/4}}{\rho}\right) R_\tau^P,$$

where we already substituted the value of $\hat{\rho}$.

To upper bound the same quantity, we follow the same analysis of Lemma 5.1. Hence, it holds:

$$\sum_{t=1}^{\tau} \left[ \mathop{\mathbb{E}}_{\boldsymbol{x} \sim \boldsymbol{\xi}_t} [f_t(\boldsymbol{x})] - \sum_{i \in [m]} \boldsymbol{\lambda}_t[i] \cdot \mathop{\mathbb{E}}_{\boldsymbol{x} \sim \boldsymbol{\xi}_t} [c_t(\boldsymbol{x})[i]] \right]$$

$$\leq \tau - \sum_{t=1}^{\tau} \sum_{i \in [m]} \boldsymbol{\lambda}_t[i] \cdot \mathop{\mathbb{E}}_{\boldsymbol{x} \sim \boldsymbol{\xi}_t} [c_t(\boldsymbol{x})[i]]$$

$$\leq \tau + \sum_{t=1}^{\tau} \sum_{i \in [m]} \boldsymbol{\lambda}[i] \cdot \left( \overline{B}_t^{(i)} - \mathop{\mathbb{E}}_{\boldsymbol{x} \sim \boldsymbol{\xi}_t} [c_t(\boldsymbol{x})[i]] \right) + \frac{2T^{\frac{1}{4}}}{\rho} R_\tau^D - \sum_{t=1}^{\tau} \sum_{i \in [m]} \boldsymbol{\lambda}_t[i] \cdot \overline{B}_t^{(i)} \quad (12\mathrm{a})$$

$$\leq \tau + \sum_{t=1}^{\tau} \sum_{i \in [m]} \boldsymbol{\lambda}[i] \cdot \left( \overline{B}_t^{(i)} - \mathop{\mathbb{E}}_{\boldsymbol{x} \sim \boldsymbol{\xi}_t} [c_t(\boldsymbol{x})[i]] \right) + \frac{2T^{\frac{1}{4}}}{\rho} R_\tau^D$$

$$= \tau + \sum_{t=1}^{\tau} \sum_{i \in [m]} \boldsymbol{\lambda}[i] \cdot \left( \left( 1 - \frac{1}{T^{1/4}} \right) B_t^{(i)} - \mathop{\mathbb{E}}_{\boldsymbol{x} \sim \boldsymbol{\xi}_t} [c_t(\boldsymbol{x})[i]] \right) + \frac{2T^{\frac{1}{4}}}{\rho} R_\tau^D$$

$$= \tau + \frac{T^{1/4}}{\rho} \cdot \sum_{t=1}^{\tau} \left( \left( 1 - \frac{1}{T^{1/4}} \right) B_t^{(i^*)} - \mathop{\mathbb{E}}_{\boldsymbol{x} \sim \boldsymbol{\xi}_t} [c_t(\boldsymbol{x})[i^*]] \right) + \frac{2T^{\frac{1}{4}}}{\rho} R_\tau^D \quad (12\mathrm{b})$$

$$\leq \tau + \frac{T^{1/4}}{\rho} \cdot \sum_{t=1}^{\tau} \left( \left( 1 - \frac{1}{T^{1/4}} \right) B_t^{(i^*)} - c_t(\boldsymbol{x}_t)[i^*] \right) + \frac{8T^{3/4}}{\rho} \sqrt{\ln \frac{T}{\delta}} + \frac{2T^{\frac{1}{4}}}{\rho} R_\tau^D \quad (12\mathrm{c})$$

$$\leq \tau + \frac{T^{1/4}}{\rho} \cdot \left( \left( 1 - \frac{1}{T^{1/4}} \right) T\rho - T\rho + 1 \right) + \frac{8T^{3/4}}{\rho} \sqrt{\ln \frac{T}{\delta}} + \frac{2T^{\frac{1}{4}}}{\rho} R_\tau^D \quad (12\mathrm{d})$$

$$= \tau + \frac{T^{1/4}}{\rho} - T + \frac{8T^{3/4}}{\rho} \sqrt{\ln \frac{T}{\delta}} + \frac{2T^{\frac{1}{4}}}{\rho} R_\tau^D$$

$$< T - CT^{3/4} + \frac{T^{1/4}}{\rho} - T + \frac{8T^{3/4}}{\rho} \sqrt{\ln \frac{T}{\delta}} + \frac{2T^{\frac{1}{4}}}{\rho} R_\tau^D \quad (12\mathrm{e})$$

$$= -CT^{3/4} + \frac{T^{1/4}}{\rho} + \frac{8T^{3/4}}{\rho} \sqrt{\ln \frac{T}{\delta}} + \frac{2T^{\frac{1}{4}}}{\rho} R_\tau^D,$$

where Inequality (12a) holds by the no-regret property of the dual regret minimizer $\mathcal{R}^D$, Equation (12b) holds selecting $\boldsymbol{\lambda}$ s.t. $\boldsymbol{\lambda}[i^*] = {T^{1/4}}/{\rho}$ and $\boldsymbol{\lambda}[i] = 0$ for all other $i \in [m]$, where $i^*$ is the depleted resource – notice that, there must exists a depleted resource since $T - \tau > 0$ –, Inequality (12c) holds with probability at least $1 - \delta$ employing the Azuma-Höeffding inequality, Inequality (12d) holds since the following holds for the resource $i^* \in [m]$:

$$\sum_{t=1}^{\tau} c_t(\boldsymbol{x}_t)[i^*] + 1 \geq B = T\rho,$$

and finally Inequality (12e) holds since $T - \tau > CT^{3/4}$.

Setting $C \geq \frac{14}{\rho} \left( \sqrt{\ln \frac{T}{\delta}} + \frac{R_T^P + R_T^D}{\sqrt{T}} \right)$ we reach the contradiction. This concludes the proof. $\qquad \square$

We are now ready to prove the final regret bound of Algorithm 4.

**Theorem C.6.** *For any $\delta \in (0, 1)$, Algorithm 4, when instantiated with a primal regret minimizer which attains a regret upper bound $R_T^P$ and a dual regret minimizer which attains a regret upper bound $R_T^D$, guarantees, with probability at least $1 - 3\delta$, the following regret bound:*

$$R_T \leq \frac{14}{\rho} \left( \sqrt{\ln \frac{T}{\delta}} + \frac{R_T^P + R_T^D}{\sqrt{T}} \right) T^{\frac{3}{4}} + T^{\frac{3}{4}} + \left( 8 + \frac{4T^{\frac{1}{4}}}{\rho} \right) \sqrt{2T \ln \frac{T}{\delta}} + \frac{2T^{\frac{1}{4}}}{\rho} R_T^D + \left( 1 + \frac{2T^{\frac{1}{4}}}{\rho} \right) R_T^P.$$

*Proof.* We first employ Lemma C.4 to get the following bound, with probability at least $1 - \delta$:

$$\sum_{t=1}^{\tau} \mathop{\mathbb{E}}_{\boldsymbol{x} \sim \boldsymbol{\xi}_t} [f_t(\boldsymbol{x})] \geq \text{OPT}_{\mathcal{H}} - (T - \tau) - T^{\frac{3}{4}} - \left(4 + \frac{4T^{\frac{1}{4}}}{\rho}\right) \sqrt{2\tau \ln \frac{T}{\delta}} - \frac{2T^{\frac{1}{4}}}{\rho} R_{\tau}^D - \left(1 + \frac{2T^{\frac{1}{4}}}{\rho}\right) R_{\tau}^P.$$

Thus, we employ the Azuma-Höeffding inequality to get, with probability at least $1 - 2\delta$ by Union Bound:

$$\sum_{t=1}^{\tau} f_t(\boldsymbol{x}_t) \geq \text{OPT}_{\mathcal{H}} - (T - \tau) - T^{\frac{3}{4}} - \left(4 + \frac{4T^{\frac{1}{4}}}{\rho}\right) \sqrt{2\tau \ln \frac{T}{\delta}}$$
$$- \frac{2T^{\frac{1}{4}}}{\rho} R_{\tau}^D - \left(1 + \frac{2T^{\frac{1}{4}}}{\rho}\right) R_{\tau}^P - 4\sqrt{2\tau \ln \frac{T}{\delta}},$$

which in turn implies, with probability at least $1 - 2\delta$:

$$R_T \leq (T - \tau) + T^{\frac{3}{4}} + \left(4 + \frac{4T^{\frac{1}{4}}}{\rho}\right) \sqrt{2\tau \ln \frac{T}{\delta}} + \frac{2T^{\frac{1}{4}}}{\rho} R_{\tau}^D + \left(1 + \frac{2T^{\frac{1}{4}}}{\rho}\right) R_{\tau}^P + 4\sqrt{2\tau \ln \frac{T}{\delta}}.$$

We then apply Lemma C.5 to obtain, with probability at least $1 - 3\delta$, by Union Bound, the following regret upper bound:

$$R_T \leq \frac{14}{\rho} \left(\sqrt{\ln \frac{T}{\delta}} + \frac{R_T^P + R_T^D}{\sqrt{T}}\right) T^{\frac{3}{4}} + T^{\frac{3}{4}} + \left(4 + \frac{4T^{\frac{1}{4}}}{\rho}\right) \sqrt{2\tau \ln \frac{T}{\delta}}$$
$$+ \frac{2T^{\frac{1}{4}}}{\rho} R_{\tau}^D + \left(1 + \frac{2T^{\frac{1}{4}}}{\rho}\right) R_{\tau}^P + 4\sqrt{2\tau \ln \frac{T}{\delta}}.$$

To get the final regret bound we notice the following trivial upper bounds:

$$R_{\tau}^D \leq R_T^D, \quad R_{\tau}^P \leq R_T^P, \quad \sqrt{2\tau \ln \frac{T}{\delta}} \leq \sqrt{2T \ln \frac{T}{\delta}}.$$

This concludes the proof. $\square$

### C.2.3 Robustness to Baselines Deviating from the Spending Plan

We provide the regret of Algorithm 4 with respect to a baseline which deviates from the spending plan.

**Theorem C.7.** *For any $\delta \in (0, 1)$, Algorithm 4, when instantiated with a primal regret minimizer which attains a regret upper bound $R_T^P$ and a dual regret minimizer which attains a regret upper bound $R_T^D$, guarantees, with probability at least $1 - 3\delta$, the following regret bound:*

$$R_T(\epsilon_t) \leq \frac{14}{\rho} \left(\sqrt{\ln \frac{T}{\delta}} + \frac{R_T^P + R_T^D}{\sqrt{T}}\right) T^{\frac{3}{4}} + T^{\frac{3}{4}} + \left(8 + \frac{4T^{\frac{1}{4}}}{\rho}\right) \sqrt{2T \ln \frac{T}{\delta}}$$
$$+ \frac{T^{\frac{1}{4}}}{\rho} \sum_{t=1}^{T} \sum_{i \in [m]} \epsilon_t^{(i)} + \frac{2T^{\frac{1}{4}}}{\rho} R_T^D + \left(1 + \frac{2T^{\frac{1}{4}}}{\rho}\right) R_T^P.$$

*Proof.* Similarly to Lemma C.4, we first employ Lemma C.3 and the definition of $\hat{\rho}$:

$$\sum_{t=1}^{\tau} \mathop{\mathbb{E}}_{\boldsymbol{x} \sim \boldsymbol{\xi}_t} [f_t(\boldsymbol{x})] \geq \sup_{\boldsymbol{\xi} \in \Xi} \sum_{t=1}^{\tau} \left[ \mathop{\mathbb{E}}_{\boldsymbol{x} \sim \boldsymbol{\xi}} [f_t(\boldsymbol{x})] + \sum_{i \in [m]} \boldsymbol{\lambda}_t[i] \cdot \left(\overline{B}_t^{(i)} - \mathop{\mathbb{E}}_{\boldsymbol{x} \sim \boldsymbol{\xi}} [c_t(\boldsymbol{x})[i]]\right) \right]$$
$$- \sum_{t=1}^{\tau} \sum_{i \in [m]} \boldsymbol{\lambda}[i] \cdot \left(\overline{B}_t^{(i)} - \mathop{\mathbb{E}}_{\boldsymbol{x} \sim \boldsymbol{\xi}_t} [c_t(\boldsymbol{x})[i]]\right) - \frac{2T^{\frac{1}{4}}}{\rho} R_{\tau}^D - \left(1 + \frac{2T^{\frac{1}{4}}}{\rho}\right) R_{\tau}^P.$$

Thus we select the Lagrange variable as $\boldsymbol{\lambda} = \mathbf{0}$ vector to establish the following bound:

$$\sum_{t=1}^{\tau} \mathop{\mathbb{E}}_{\boldsymbol{x}\sim\boldsymbol{\xi}_t} [f_t(\boldsymbol{x})] \geq \sup_{\boldsymbol{\xi}\in\Xi} \sum_{t=1}^{\tau} \left[ \mathop{\mathbb{E}}_{\boldsymbol{x}\sim\boldsymbol{\xi}} [f_t(\boldsymbol{x})] + \sum_{i\in[m]} \boldsymbol{\lambda}_t[i] \cdot \left( \overline{B}_t^{(i)} - \mathop{\mathbb{E}}_{\boldsymbol{x}\sim\boldsymbol{\xi}} [c_t(\boldsymbol{x})[i]] \right) \right]$$
$$- \frac{2T^{\frac{1}{4}}}{\rho} R_\tau^D - \left( 1 + \frac{2T^{\frac{1}{4}}}{\rho} \right) R_\tau^P.$$

We now focus on bounding the following term when the baseline deviates from the spending plan:

$$\sup_{\boldsymbol{\xi}\in\Xi} \sum_{t=1}^{\tau} \left[ \mathop{\mathbb{E}}_{\boldsymbol{x}\sim\boldsymbol{\xi}} [f_t(\boldsymbol{x})] + \sum_{i\in[m]} \boldsymbol{\lambda}_t[i] \cdot \left( \overline{B}_t^{(i)} - \mathop{\mathbb{E}}_{\boldsymbol{x}\sim\boldsymbol{\xi}} [c_t(\boldsymbol{x})[i]] \right) \right] .$$

We define $\Xi^\circ$ as the set which encompasses all safe strategy mixtures during the learning dynamic. Specifically, we let $\Xi^\circ := \{ \boldsymbol{\xi} \in \Xi : \mathbb{E}_{\boldsymbol{x}\sim\boldsymbol{\xi}}[\bar{c}_t(x)] \leq B_t^{(i)} - \epsilon_t^{(i)}, \forall i \in [m], t \in [T]\}$. Now, notice that by definition of Problem (11), there exists a strategy $\boldsymbol{\xi}^* \in \Xi^\circ$ such that $\sum_{t=1}^{T} \mathbb{E}_{\boldsymbol{x}\sim\boldsymbol{\xi}^*} [\bar{f}_t(\boldsymbol{x})] \geq \mathrm{OPT}_{\mathcal{H}}(\epsilon_t) - \gamma$, for all $\gamma > 0$. In the rest of the paper, we omit the factor $\gamma$, since it can be chosen arbitrarily small, thus being a negligible factor in the regret bound. We then define the following strategy mixture $\boldsymbol{\xi}^\diamond$ as follows:

$$\boldsymbol{\xi}^\diamond := \begin{cases} \boldsymbol{x}^\varnothing & \text{w.p. } 1/T^{1/4} \\ \boldsymbol{\xi}^* & \text{w.p. } 1 - 1/T^{1/4} \end{cases} .$$

Thus, we first show that $\boldsymbol{\xi}^\diamond$ satisfies the per-round expected constraints defined by $\overline{B}_t^{(i)}$. Indeed it holds, for all $i \in [m]$:

$$\overline{B}_t^{(i)} - \mathop{\mathbb{E}}_{\boldsymbol{x}\sim\boldsymbol{\xi}^\diamond}[\bar{c}_t(\boldsymbol{x})] = \overline{B}_t^{(i)} - \left( 1 - \frac{1}{T^{1/4}} \right) \mathop{\mathbb{E}}_{\boldsymbol{x}\sim\boldsymbol{\xi}^*}[\bar{c}_t(\boldsymbol{x})] - \frac{1}{T^{1/4}} \mathop{\mathbb{E}}_{\boldsymbol{x}\sim\boldsymbol{\xi}^\varnothing}[\bar{c}_t(\boldsymbol{x})]$$
$$= \left( 1 - \frac{1}{T^{1/4}} \right) B_t^{(i)} - \left( 1 - \frac{1}{T^{1/4}} \right) \mathop{\mathbb{E}}_{\boldsymbol{x}\sim\boldsymbol{\xi}^*}[\bar{c}_t(\boldsymbol{x})] - \frac{1}{T^{1/4}} \mathop{\mathbb{E}}_{\boldsymbol{x}\sim\boldsymbol{\xi}^\varnothing}[\bar{c}_t(\boldsymbol{x})]$$
$$= \left( 1 - \frac{1}{T^{1/4}} \right) B_t^{(i)} - \left( 1 - \frac{1}{T^{1/4}} \right) \mathop{\mathbb{E}}_{\boldsymbol{x}\sim\boldsymbol{\xi}^*}[\bar{c}_t(\boldsymbol{x})]$$
$$\geq - \left( 1 - \frac{1}{T^{1/4}} \right) \epsilon_t^{(i)}$$
$$\geq -\epsilon_t^{(i)},$$

where we employed the definition of $\boldsymbol{\xi}^*$ and $\boldsymbol{\xi}^\varnothing$.

Thus, returning to the quantity of interest, it holds:

$$\sup_{\boldsymbol{\xi}\in\Xi} \sum_{t=1}^{\tau} \left[ \mathop{\mathbb{E}}_{\boldsymbol{x}\sim\boldsymbol{\xi}} [f_t(\boldsymbol{x})] + \sum_{i\in[m]} \boldsymbol{\lambda}_t[i] \cdot \left( \overline{B}_t^{(i)} - \mathop{\mathbb{E}}_{\boldsymbol{x}\sim\boldsymbol{\xi}} [c_t(\boldsymbol{x})[i]] \right) \right]$$
$$\geq \sum_{t=1}^{\tau} \left[ \mathop{\mathbb{E}}_{\boldsymbol{x}\sim\boldsymbol{\xi}^\diamond} [f_t(\boldsymbol{x})] + \sum_{i\in[m]} \boldsymbol{\lambda}_t[i] \cdot \left( \overline{B}_t^{(i)} - \mathop{\mathbb{E}}_{\boldsymbol{x}\sim\boldsymbol{\xi}^\diamond} [c_t(\boldsymbol{x})[i]] \right) \right]$$
$$\geq \sum_{t=1}^{\tau} \left[ \mathop{\mathbb{E}}_{\boldsymbol{x}\sim\boldsymbol{\xi}^\diamond} [\bar{f}_t(\boldsymbol{x})] + \sum_{i\in[m]} \boldsymbol{\lambda}_t[i] \cdot \left( \overline{B}_t^{(i)} - \mathop{\mathbb{E}}_{\boldsymbol{x}\sim\boldsymbol{\xi}^\diamond} [\bar{c}_t(\boldsymbol{x})[i]] \right) \right] - \left( 4 + \frac{4T^{1/4}}{\rho} \right) \sqrt{2\tau \ln \frac{T}{\delta}}$$
$$\geq \sum_{t=1}^{\tau} \left[ \mathop{\mathbb{E}}_{\boldsymbol{x}\sim\boldsymbol{\xi}^\diamond} [\bar{f}_t(\boldsymbol{x})] \right] - \frac{T^{1/4}}{\rho} \sum_{t=1}^{\tau} \sum_{i\in[m]} \epsilon_t^{(i)} - \left( 4 + \frac{4T^{1/4}}{\rho} \right) \sqrt{2\tau \ln \frac{T}{\delta}}$$
$$= \sum_{t=1}^{\tau} \left[ \left( 1 - \frac{1}{T^{1/4}} \right) \mathop{\mathbb{E}}_{\boldsymbol{x}\sim\boldsymbol{\xi}^*} [\bar{f}_t(\boldsymbol{x})] + \frac{1}{T^{1/4}} \mathop{\mathbb{E}}_{\boldsymbol{x}\sim\boldsymbol{\xi}^\varnothing} [\bar{f}_t(\boldsymbol{x})] \right] - \frac{T^{1/4}}{\rho} \sum_{t=1}^{\tau} \sum_{i\in[m]} \epsilon_t^{(i)}$$

$$- \left( 4 + \frac{4T^{1/4}}{\rho} \right) \sqrt{2\tau \ln \frac{T}{\delta}}$$

$$= \sum_{t=1}^{\tau} \left[ \left( 1 - \frac{1}{T^{1/4}} \right) \mathop{\mathbb{E}}_{\boldsymbol{x} \sim \boldsymbol{\xi}^*} \left[ \bar{f}_t(\boldsymbol{x}) \right] \right] - \frac{T^{1/4}}{\rho} \sum_{t=1}^{\tau} \sum_{i \in [m]} \epsilon_t^{(i)} - \left( 4 + \frac{4T^{1/4}}{\rho} \right) \sqrt{2\tau \ln \frac{T}{\delta}}$$

$$\geq \sum_{t=1}^{\tau} \mathop{\mathbb{E}}_{\boldsymbol{x} \sim \boldsymbol{\xi}^*} \left[ \bar{f}_t(\boldsymbol{x}) \right] - T^{3/4} - \frac{T^{1/4}}{\rho} \sum_{t=1}^{\tau} \sum_{i \in [m]} \epsilon_t^{(i)} - \left( 4 + \frac{4T^{1/4}}{\rho} \right) \sqrt{2\tau \ln \frac{T}{\delta}}$$

$$\geq \mathrm{OPT}_{\mathcal{H}}(\epsilon_t) - T^{3/4} - (T - \tau) - \frac{T^{1/4}}{\rho} \sum_{t=1}^{\tau} \sum_{i \in [m]} \epsilon_t^{(i)} - \left( 4 + \frac{4T^{1/4}}{\rho} \right) \sqrt{2\tau \ln \frac{T}{\delta}},$$

where the second inequality holds, with probability at least $1 - \delta$, by Lemma E.2 and upper bounding the Lagrangian multiplier with $T^{\frac{1}{4}}/\rho$. Following the same analysis of Theorem C.6 and noticing that:

$$\sum_{t=1}^{\tau} \sum_{i \in [m]} \epsilon_t^{(i)} \leq \sum_{t=1}^{\tau} \sum_{i \in [m]} \epsilon_t^{(i)},$$

concludes the proof. $\qquad\square$

# D Omitted Proofs for OLRC with Bandit Feedback

In this section, we provide the results and the omitted proofs for OLRC with *bandit feedback*.

## D.1 Theoretical Guarantees of Algorithm 2

Similarly to the *full feedback* case, we start by providing a lower bound to the expected rewards attained by Algorithm 2.

**Lemma D.1.** *Algorithm 2, when instantiated in the bandit feedback setting with a primal regret minimizer (with bandit feedback) which attains a regret upper bound $R_T^P$, with probability at least $1 - \delta_P$, and a dual regret minimizer which attains a regret upper bound $R_T^D$, guarantees the following bound:*

$$\sum_{t=1}^{\tau} f_t(\boldsymbol{x}_t) \geq \sup_{\boldsymbol{x} \in \mathcal{X}} \sum_{t=1}^{\tau} \left[ f_t(\boldsymbol{x}) + \sum_{i \in [m]} \boldsymbol{\lambda}_t[i] \cdot \left( B_t^{(i)} - c_t(\boldsymbol{x})[i] \right) \right]$$

$$- \sum_{t=1}^{\tau} \sum_{i \in [m]} \boldsymbol{\lambda}[i] \cdot \left( B_t^{(i)} - c_t(\boldsymbol{x}_t)[i] \right) - \frac{2}{\rho_{\min}} R_\tau^D - \left( 1 + \frac{2}{\rho_{\min}} \right) R_\tau^P,$$

*which holds with probability at least $1 - \delta_P$, where $\boldsymbol{\lambda} \in \mathcal{L}$ is an arbitrary Lagrange multiplier and $\tau$ is stopping time of the algorithm.*

*Proof.* Similarly to the proof of Lemma C.1, we will refer to the stopping time of Algorithm 2 as $\tau$. We employ the no-regret property of the primal regret minimizer $\mathcal{R}^P$, which works with bandit feedback. Given that, it holds, with probability at least $1 - \delta_P$:

$$\sup_{\boldsymbol{x} \in \mathcal{X}} \sum_{t=1}^{\tau} \left[ f_t(\boldsymbol{x}) + \sum_{i \in [m]} \boldsymbol{\lambda}_t[i] \cdot \left( B_t^{(i)} - c_t(\boldsymbol{x})[i] \right) \right]$$

$$- \sum_{t=1}^{\tau} \left[ f_t(\boldsymbol{x}_t) - \sum_{i \in [m]} \boldsymbol{\lambda}_t[i] \cdot c_t(\boldsymbol{x}_t)[i] \right] \leq \left( 1 + \frac{2}{\rho_{\min}} \right) R_\tau^P, \quad (13)$$

where the $(1 + 2/\rho_{\min})$ factor is the dependence on the payoffs range given as feedback to the primal regret minimizer.

Thus we can rearrange Equation (13) to obtain the following bound:

$$\sum_{t=1}^{\tau} f_t(\boldsymbol{x}_t) \geq \sup_{\boldsymbol{x} \in \mathcal{X}} \sum_{t=1}^{\tau} \left[ f_t(\boldsymbol{x}) + \sum_{i \in [m]} \boldsymbol{\lambda}_t[i] \cdot \left( B_t^{(i)} - c_t(\boldsymbol{x})[i] \right) \right]$$

$$+ \sum_{t=1}^{\tau} \sum_{i \in [m]} \boldsymbol{\lambda}_t[i] \cdot c_t(\boldsymbol{x}_t)[i] - \left( 1 + \frac{2}{\rho_{\min}} \right) R_\tau^P, \qquad (14)$$

which holds with probability at least $1 - \delta_P$. Given the dual regret minimizer, it holds, for any Lagrange multiplier $\boldsymbol{\lambda} \in \mathcal{L}$:

$$\sum_{t=1}^{\tau} \sum_{i \in [m]} \boldsymbol{\lambda}_t[i] \cdot \left( B_t^{(i)} - c_t(\boldsymbol{x}_t)[i] \right) - \sum_{t=1}^{\tau} \sum_{i \in [m]} \boldsymbol{\lambda}[i] \cdot \left( B_t^{(i)} - c_t(\boldsymbol{x}_t)[i] \right) \leq \frac{2}{\rho_{\min}} R_\tau^D,$$

which in turn implies:

$$\sum_{t=1}^{\tau} \sum_{i \in [m]} \boldsymbol{\lambda}_t[i] \cdot c_t(\boldsymbol{x}_t)[i] \geq \sum_{t=1}^{\tau} \sum_{i \in [m]} \boldsymbol{\lambda}_t[i] \cdot B_t^{(i)} - \sum_{t=1}^{\tau} \sum_{i \in [m]} \boldsymbol{\lambda}[i] \cdot \left( B_t^{(i)} - c_t(\boldsymbol{x}_t)[i] \right) - \frac{2}{\rho_{\min}} R_\tau^D,$$
(15)

where the $2/\rho_{\min}$ factor follows from the dual payoffs range. We substitute Equation (15) in Equation (14) to obtain, with probability at least $1 - \delta_P$:

$$\sum_{t=1}^{\tau} f_t(\boldsymbol{x}_t) \geq \sup_{\boldsymbol{x} \in \mathcal{X}} \sum_{t=1}^{\tau} \left[ f_t(\boldsymbol{x}) + \sum_{i \in [m]} \boldsymbol{\lambda}_t[i] \cdot \left( B_t^{(i)} - c_t(\boldsymbol{x})[i] \right) \right]$$

$$- \sum_{t=1}^{\tau} \sum_{i \in [m]} \boldsymbol{\lambda}[i] \cdot \left( B_t^{(i)} - c_t(\boldsymbol{x}_t)[i] \right) - \frac{2}{\rho_{\min}} R_\tau^D - \left( 1 + \frac{2}{\rho_{\min}} \right) R_\tau^P.$$

This concludes the proof. $\qquad \square$

We refine the previous lower bound. This is done by means of the following lemma.

**Lemma D.2.** *For any $\delta \in (0, 1)$, Algorithm 2, when instantiated in the bandit feedback setting with a primal regret minimizer (with bandit feedback) which attains a regret upper bound $R_T^P$, with probability at least $1 - \delta_P$, and a dual regret minimizer which attains a regret upper bound $R_T^D$, guarantees the following bound:*

$$\sum_{t=1}^{T} f_t(\boldsymbol{x}_t) \geq \text{OPT}_{\mathcal{H}} - (T - \tau) - \frac{1}{\rho_{\min}} \sum_{t=1}^{\tau} \sum_{i \in [m]} |\epsilon_t^{(i)}| - \left( 4 + 4 \max_{\boldsymbol{\lambda} \in \mathcal{L}} \|\boldsymbol{\lambda}\|_1 \right) \sqrt{2\tau \ln \frac{T}{\delta}}$$

$$- \sum_{t=1}^{\tau} \sum_{i \in [m]} \boldsymbol{\lambda}[i] \cdot \left( B_t^{(i)} - c_t(\boldsymbol{x}_t)[i] \right) - \frac{2}{\rho_{\min}} R_\tau^D - \left( 1 + \frac{2}{\rho_{\min}} \right) R_\tau^P,$$

*which holds with probability at least $1 - (\delta + \delta_P)$, where $\boldsymbol{\lambda} \in \mathcal{L}$ is an arbitrary Lagrange multiplier and $\tau$ is stopping time of the algorithm.*

*Proof.* We employ Lemma D.1 to obtain:

$$\sum_{t=1}^{\tau} f_t(\boldsymbol{x}_t) \geq \sup_{\boldsymbol{x} \in \mathcal{X}} \sum_{t=1}^{\tau} \left[ f_t(\boldsymbol{x}) + \sum_{i \in [m]} \boldsymbol{\lambda}_t[i] \cdot \left( B_t^{(i)} - c_t(\boldsymbol{x})[i] \right) \right]$$

$$- \sum_{t=1}^{\tau} \sum_{i \in [m]} \boldsymbol{\lambda}[i] \cdot \left( B_t^{(i)} - c_t(\boldsymbol{x}_t)[i] \right) - \frac{2}{\rho_{\min}} R_\tau^D - \left( 1 + \frac{2}{\rho_{\min}} \right) R_\tau^P.$$

To conclude the proof, we focus on bounding the term:

$$\sup_{\boldsymbol{x} \in \mathcal{X}} \sum_{t=1}^{\tau} \left[ f_t(\boldsymbol{x}) + \sum_{i \in [m]} \boldsymbol{\lambda}_t[i] \cdot \left( B_t^{(i)} - c_t(\boldsymbol{x})[i] \right) \right].$$

Specifically, notice that by definition of the probability mixture $\Xi$, it holds:

$$\sup_{\boldsymbol{x} \in \mathcal{X}} \sum_{t=1}^{\tau} \left[ f_t(\boldsymbol{x}) + \sum_{i \in [m]} \boldsymbol{\lambda}_t[i] \cdot \left( B_t^{(i)} - c_t(\boldsymbol{x})[i] \right) \right]$$

$$= \sup_{\boldsymbol{\xi} \in \Xi} \sum_{t=1}^{\tau} \left[ \mathbb{E}_{\boldsymbol{x} \sim \boldsymbol{\xi}} [f_t(\boldsymbol{x})] + \sum_{i \in [m]} \boldsymbol{\lambda}_t[i] \cdot \left( B_t^{(i)} - \mathbb{E}_{\boldsymbol{x} \sim \boldsymbol{\xi}} [c_t(\boldsymbol{x})[i]] \right) \right].$$

Thus, employing the same analysis of Lemma 4.1 concludes the proof. □

We are ready to prove the final result on the regret bound of Algorithm 2 for bandit feedback.

**Theorem D.3.** *For any $\delta \in (0,1)$, Algorithm 2, when instantiated in the bandit feedback setting with a primal regret minimizer (with bandit feedback) which attains, with probability at least $1 - \delta_P$, a regret upper bound $R_T^P$ and a dual regret minimizer which attains a regret upper bound $R_T^D$, guarantees, with probability at least $1 - (\delta + \delta_P)$, the following regret bound:*

$$R_T \leq 1 + \frac{1}{\rho_{\min}} + \frac{2}{\rho_{\min}} R_T^D + \left( 1 + \frac{2}{\rho_{\min}} \right) R_T^P + \left( 4 + \frac{4}{\rho_{\min}} \right) \sqrt{2T \ln \frac{T}{\delta}}.$$

*Proof.* The analysis is equivalent to the one of Theorem 4.2, once applied Lemma D.2 and after noticing the employment of Lemma E.1 is not necessary. □

### D.1.1 Robustness to Baselines Deviating from the Spending Plan

We provide the regret of Algorithm 2 with respect to a baseline which deviates from the spending plan.

**Theorem D.4.** *For any $\delta \in (0,1)$, Algorithm 2, when instantiated in the bandit feedback setting with a primal regret minimizer (with bandit feedback) which attains, with probability at least $1 - \delta_P$, a regret upper bound $R_T^P$ and a dual regret minimizer which attains a regret upper bound $R_T^D$, guarantees, with probability at least $1 - (\delta + \delta_P)$, the following regret bound:*

$$R_T \leq 1 + \frac{1}{\rho_{\min}} + \frac{2}{\rho_{\min}} R_T^D + \left( 1 + \frac{2}{\rho_{\min}} \right) R_T^P + \left( 4 + \frac{4}{\rho_{\min}} \right) \sqrt{2T \ln \frac{T}{\delta}} + \frac{1}{\rho_{\min}} \sum_{t=1}^{T} \sum_{i \in [m]} \epsilon_t^{(i)}.$$

*Proof.* We employ Lemma D.1 to obtain:

$$\sum_{t=1}^{\tau} f_t(\boldsymbol{x}_t) \geq \sup_{\boldsymbol{x} \in \mathcal{X}} \sum_{t=1}^{\tau} \left[ f_t(\boldsymbol{x}) + \sum_{i \in [m]} \boldsymbol{\lambda}_t[i] \cdot \left( B_t^{(i)} - c_t(\boldsymbol{x})[i] \right) \right]$$

$$- \sum_{t=1}^{\tau} \sum_{i \in [m]} \boldsymbol{\lambda}[i] \cdot \left( B_t^{(i)} - c_t(\boldsymbol{x}_t)[i] \right) - \frac{2}{\rho_{\min}} R_\tau^D - \left( 1 + \frac{2}{\rho_{\min}} \right) R_\tau^P.$$

The proof follows from noticing that:

$$\sup_{\boldsymbol{x} \in \mathcal{X}} \sum_{t=1}^{\tau} \left[ f_t(\boldsymbol{x}) + \sum_{i \in [m]} \boldsymbol{\lambda}_t[i] \cdot \left( B_t^{(i)} - c_t(\boldsymbol{x})[i] \right) \right]$$

$$= \sup_{\boldsymbol{\xi} \in \Xi} \sum_{t=1}^{\tau} \left[ \mathbb{E}_{\boldsymbol{x} \sim \boldsymbol{\xi}} [f_t(\boldsymbol{x})] + \sum_{i \in [m]} \boldsymbol{\lambda}_t[i] \cdot \left( B_t^{(i)} - \mathbb{E}_{\boldsymbol{x} \sim \boldsymbol{\xi}} [c_t(\boldsymbol{x})[i]] \right) \right],$$

and employing the same analysis of Theorem C.2. □

## D.2 Theoretical Guarantees of Algorithm 5

In this section, we present the results attained by the meta procedure provided in Algorithm 5.

### D.2.1 Algorithm

We first provide the algorithm for the bandit feedback case.

---

**Algorithm 5** Meta-algorithm for arbitrarily small $\rho_{\min}$ and *bandit feedback*

---

**Require:** Horizon $T$, budget $B$, spending plans $\mathcal{B}_T^{(i)}$ for all $i \in [m]$, primal regret minimizer $\mathcal{R}^P$ (*bandit feedback*), dual regret minimizer $\mathcal{R}^D$ (*full feedback*)
  1: Define $\hat{\rho} := \rho/T^{1/4}$
  2: Define $\overline{B}_t^{(i)} := B_t^{(i)}\left(1 - T^{-1/4}\right) \quad \forall t \in [T], i \in [m]$
  3: Run Algorithm 2 for *bandit feedback* with $\rho_{\min} \leftarrow \hat{\rho}, B_t^{(i)} \leftarrow \overline{B}_t^{(i)}$

---

### D.2.2 Analysis

In this section, we provide the analysis of Algorithm 5. We start by lower bounding the expected rewards attained by the meta procedure.

**Lemma D.5.** *Algorithm 5, when instantiated with a primal regret minimizer (with bandit feedback) which attains a regret upper bound $R_T^P$, with probability at least $1 - \delta_P$, and a dual regret minimizer which attains a regret upper bound $R_T^D$, guarantees the following bound:*

$$\sum_{t=1}^{\tau} f_t(\boldsymbol{x}_t) \geq \sup_{\boldsymbol{x} \in \mathcal{X}} \sum_{t=1}^{\tau} \left[ f_t(\boldsymbol{x}) + \sum_{i \in [m]} \boldsymbol{\lambda}_t[i] \cdot \left( \overline{B}_t^{(i)} - c_t(\boldsymbol{x})[i] \right) \right]$$
$$- \sum_{t=1}^{\tau} \sum_{i \in [m]} \boldsymbol{\lambda}[i] \cdot \left( \overline{B}_t^{(i)} - c_t(\boldsymbol{x}_t)[i] \right) - \frac{2}{\hat{\rho}} R_\tau^D - \left( 1 + \frac{2}{\hat{\rho}} \right) R_\tau^P,$$

*which holds with probability at least $1 - \delta_P$, where $\boldsymbol{\lambda} \in \mathcal{L}$ is an arbitrary Lagrange multiplier and $\tau$ is stopping time of the algorithm.*

*Proof.* The proof is analogous to the one of Lemma D.1 after substituting $\rho_{\min}$ with $\hat{\rho}$ and $B_t^{(i)}$ with $\overline{B}_t^{(i)}$, for all $i \in [m], t \in [T]$. $\qquad\qquad\square$

We refine the previous lower bound as follows.

**Lemma D.6.** *For any $\delta \in (0, 1)$, Algorithm 5, when instantiated with a primal regret minimizer (with bandit feedback) which attains a regret upper bound $R_T^P$, with probability at least $1 - \delta_P$ and a dual regret minimizer which attains a regret upper bound $R_T^D$, guarantees the following bound, with probability at least $1 - (\delta + \delta_P)$:*

$$\sum_{t=1}^{\tau} f_t(\boldsymbol{x}_t) \geq \mathrm{OPT}_\mathcal{H} - (T - \tau) - T^{\frac{3}{4}} - \left( 4 + \frac{4T^{\frac{1}{4}}}{\rho} \right) \sqrt{2\tau \ln \frac{T}{\delta}} - \frac{2T^{\frac{1}{4}}}{\rho} R_\tau^D - \left( 1 + \frac{2T^{\frac{1}{4}}}{\rho} \right) R_\tau^P,$$

*where $\tau$ is the stopping time of the algorithm.*

*Proof.* We first employ Lemma D.5 and the definition of $\hat{\rho}$ to obtain, with probability at least $1 - \delta_P$:

$$\sum_{t=1}^{\tau} f_t(\boldsymbol{x}_t) \geq \sup_{\boldsymbol{x} \in \mathcal{X}} \sum_{t=1}^{\tau} \left[ f_t(\boldsymbol{x}) + \sum_{i \in [m]} \boldsymbol{\lambda}_t[i] \cdot \left( \overline{B}_t^{(i)} - c_t(\boldsymbol{x})[i] \right) \right]$$
$$- \sum_{t=1}^{\tau} \sum_{i \in [m]} \boldsymbol{\lambda}[i] \cdot \left( \overline{B}_t^{(i)} - c_t(\boldsymbol{x}_t)[i] \right) - \frac{2T^{\frac{1}{4}}}{\rho} R_\tau^D - \left( 1 + \frac{2T^{\frac{1}{4}}}{\rho} \right) R_\tau^P.$$

Thus we select the Lagrange variable as $\boldsymbol{\lambda} = \mathbf{0}$ vector to establish the following bound:

$$\sum_{t=1}^{\tau} f_t(\boldsymbol{x}_t) \geq \sup_{\boldsymbol{x} \in \mathcal{X}} \sum_{t=1}^{\tau} \left[ f_t(\boldsymbol{x}) + \sum_{i \in [m]} \boldsymbol{\lambda}_t[i] \cdot \left( \overline{B}_t^{(i)} - c_t(\boldsymbol{x})[i] \right) \right] - \frac{2T^{\frac{1}{4}}}{\rho} R_\tau^D - \left( 1 + \frac{2T^{\frac{1}{4}}}{\rho} \right) R_\tau^P,$$

which holds with probability at least $1 - \delta_P$. We now focus on bounding the term:

$$\sup_{\boldsymbol{x} \in \mathcal{X}} \sum_{t=1}^{\tau} \left[ f_t(\boldsymbol{x}) + \sum_{i \in [m]} \boldsymbol{\lambda}_t[i] \cdot \left( \overline{B}_t^{(i)} - c_t(\boldsymbol{x})[i] \right) \right].$$

This is done equivalently to Lemma C.4 once noticed that, by definition of the strategy mixture space, it holds:

$$\sup_{\boldsymbol{x} \in \mathcal{X}} \sum_{t=1}^{\tau} \left[ f_t(\boldsymbol{x}) + \sum_{i \in [m]} \boldsymbol{\lambda}_t[i] \cdot \left( \overline{B}_t^{(i)} - c_t(\boldsymbol{x})[i] \right) \right]$$

$$= \sup_{\boldsymbol{\xi} \in \Xi} \sum_{t=1}^{\tau} \left[ \mathop{\mathbb{E}}_{\boldsymbol{x} \sim \boldsymbol{\xi}} [f_t(\boldsymbol{x})] + \sum_{i \in [m]} \boldsymbol{\lambda}_t[i] \cdot \left( \overline{B}_t^{(i)} - \mathop{\mathbb{E}}_{\boldsymbol{x} \sim \boldsymbol{\xi}} [c_t(\boldsymbol{x})[i]] \right) \right].$$

This concludes the proof. $\qquad\square$

Hence, we proceed upper bounding the difference between the horizon $T$ and the stopping time $\tau$, when only *bandit feedback* is available. This is done by means of the following lemma.

**Lemma D.7.** *For any $\delta \in (0, 1)$, Algorithm 5, when instantiated with a primal regret minimizer (with bandit feedback) which attains a regret upper bound $R_T^P$, with probability at least $1 - \delta_P$ and a dual regret minimizer which attains a regret upper bound $R_T^D$, guarantees the following bound:*

$$T - \tau \leq \frac{14}{\rho} \cdot \frac{R_T^P + R_T^D}{\sqrt{T}} T^{\frac{3}{4}},$$

*which holds with probability at least $1 - \delta_P$.*

*Proof.* We proceed similarly to Lemma C.5 and we suppose by contradiction that $T - \tau > CT^{3/4}$, thus, it holds $\tau < T - CT^{3/4}$.

We proceed upper and lower bounding the value of the Lagrangian given as feedback to the primal regret minimizer. Thus, we employ the no-regret property of the primal regret minimizer $\mathcal{R}^P$. Given that, it holds, with probability at least $1 - \delta_P$:

$$\sum_{t=1}^{\tau} \left[ f_t(\boldsymbol{x}_t) - \sum_{i \in [m]} \boldsymbol{\lambda}_t[i] \cdot c_t(\boldsymbol{x}_t)[i] \right]$$

$$\geq \sum_{t=1}^{\tau} \left[ f_t(\boldsymbol{x}^{\varnothing}) - \sum_{i \in [m]} \boldsymbol{\lambda}_t[i] \cdot c_t(\boldsymbol{x}^{\varnothing})[i] \right] - \left( 1 + \frac{2T^{1/4}}{\rho} \right) R_\tau^P$$

$$= - \left( 1 + \frac{2T^{1/4}}{\rho} \right) R_\tau^P,$$

where we already substituted the value of $\hat{\rho}$.

Similarly, to upper bound the same quantity, we proceed as follows:

$$\sum_{t=1}^{\tau} \left[ f_t(\boldsymbol{x}_t) - \sum_{i \in [m]} \boldsymbol{\lambda}_t[i] \cdot c_t(\boldsymbol{x}_t)[i] \right]$$

$$\leq \tau - \sum_{t=1}^{\tau} \sum_{i \in [m]} \boldsymbol{\lambda}_t[i] \cdot c_t(\boldsymbol{x}_t)[i]$$

$$\leq \tau + \sum_{t=1}^{\tau} \sum_{i \in [m]} \boldsymbol{\lambda}[i] \cdot \left( \overline{B}_t^{(i)} - c_t(\boldsymbol{x}_t)[i] \right) + \frac{2T^{\frac{1}{4}}}{\rho} R_\tau^D - \sum_{t=1}^{\tau} \sum_{i \in [m]} \boldsymbol{\lambda}_t[i] \cdot \overline{B}_t^{(i)} \qquad (16a)$$

$$\leq \tau + \sum_{t=1}^{\tau} \sum_{i \in [m]} \boldsymbol{\lambda}[i] \cdot \left( \overline{B}_t^{(i)} - c_t(\boldsymbol{x}_t)[i] \right) + \frac{2T^{\frac{1}{4}}}{\rho} R_\tau^D$$

$$= \tau + \sum_{t=1}^{\tau} \sum_{i \in [m]} \boldsymbol{\lambda}[i] \cdot \left( \left( 1 - \frac{1}{T^{1/4}} \right) B_t^{(i)} - c_t(\boldsymbol{x}_t)[i] \right) + \frac{2T^{\frac{1}{4}}}{\rho} R_\tau^D$$

$$= \tau + \frac{T^{1/4}}{\rho} \cdot \sum_{t=1}^{\tau} \left( \left( 1 - \frac{1}{T^{1/4}} \right) B_t^{(i^*)} - c_t(\boldsymbol{x}_t)[i^*] \right) + \frac{2T^{\frac{1}{4}}}{\rho} R_\tau^D \quad \text{(16b)}$$

$$\leq \tau + \frac{T^{1/4}}{\rho} \cdot \left( \left( 1 - \frac{1}{T^{1/4}} \right) T\rho - T\rho + 1 \right) + \frac{2T^{\frac{1}{4}}}{\rho} R_\tau^D \quad \text{(16c)}$$

$$= \tau + \frac{T^{1/4}}{\rho} - T + \frac{2T^{\frac{1}{4}}}{\rho} R_\tau^D$$

$$< T - CT^{3/4} + \frac{T^{1/4}}{\rho} - T + \frac{2T^{\frac{1}{4}}}{\rho} R_\tau^D \quad \text{(16d)}$$

$$= -CT^{3/4} + \frac{T^{1/4}}{\rho} + \frac{2T^{\frac{1}{4}}}{\rho} R_\tau^D,$$

where Inequality (16a) holds by the no-regret property of the dual regret minimizer $\mathcal{R}^D$, Equation (16b) holds selecting $\boldsymbol{\lambda}$ s.t. $\boldsymbol{\lambda}[i^*] = {}^{T^{1/4}}\!/\rho$ and $\boldsymbol{\lambda}[i] = 0$ for all other $i \in [m]$, where $i^*$ is the depleted resource – notice that, there must exists a depleted resource since $T - \tau > 0$ –, Inequality (16c) holds since the following holds for the resource $i^* \in [m]$:

$$\sum_{t=1}^{\tau} c_t(\boldsymbol{x}_t)[i^*] + 1 \geq B = T\rho,$$

and finally Inequality (16d) holds since $T - \tau > CT^{3/4}$.

Setting $C \geq \frac{14}{\rho} \cdot \frac{R_T^P + R_T^D}{\sqrt{T}}$ we reach the contradiction. This concludes the proof. $\qquad \square$

We are now ready to prove the final regret bound of Algorithm 5.

**Theorem D.8.** *For any $\delta \in (0, 1)$, Algorithm 5, when instantiated with a primal regret minimizer (with bandit feedback) which attains a regret upper bound $R_T^P$, with probability at least $1 - \delta_P$, and a dual regret minimizer which attains a regret upper bound $R_T^D$, guarantees, with probability at least $1 - (\delta + \delta_P)$, the following regret bound:*

$$R_T \leq \frac{14}{\rho} \cdot \frac{R_T^P + R_T^D}{\sqrt{T}} T^{\frac{3}{4}} + T^{\frac{3}{4}} + \left( 4 + \frac{4T^{\frac{1}{4}}}{\rho} \right) \sqrt{2T \ln \frac{T}{\delta}} + \frac{2T^{\frac{1}{4}}}{\rho} R_T^D + \left( 1 + \frac{2T^{\frac{1}{4}}}{\rho} \right) R_T^P.$$

*Proof.* The analysis is equivalent to the one of Theorem C.6, once applied Lemma D.6 and Lemma D.7 and after noticing the employment of the Azuma-Höeffding inequality is not necessary. $\qquad \square$

### D.2.3 Robustness to Baselines Deviating from the Spending Plan

We provide the regret of Algorithm 5 with respect to a baseline which deviates from the spending plan.

**Theorem D.9.** *For any $\delta \in (0, 1)$, Algorithm 5, when instantiated with a primal regret minimizer (with bandit feedback) which attains a regret upper bound $R_T^P$, with probability at least $1 - \delta_P$, and a dual regret minimizer which attains a regret upper bound $R_T^D$, guarantees, with probability at least $1 - (\delta + \delta_P)$, the following regret bound:*

$$R_T(\epsilon_t) \leq \frac{14}{\rho} \cdot \frac{R_T^P + R_T^D}{\sqrt{T}} T^{\frac{3}{4}} + T^{\frac{3}{4}} + \left( 4 + \frac{4T^{\frac{1}{4}}}{\rho} \right) \sqrt{2T \ln \frac{T}{\delta}}$$

$$+ \frac{T^{\frac{1}{4}}}{\rho} \sum_{t=1}^{T} \sum_{i \in [m]} \epsilon_t^{(i)} + \frac{2T^{\frac{1}{4}}}{\rho} R_T^D + \left( 1 + \frac{2T^{\frac{1}{4}}}{\rho} \right) R_T^P.$$

*Proof.* We follow the proof of Lemma D.6 to obtain:

$$\sum_{t=1}^{\tau} f_t(\boldsymbol{x}_t) \geq \sup_{\boldsymbol{x}\in\mathcal{X}} \sum_{t=1}^{\tau}\left[f_t(\boldsymbol{x}) + \sum_{i\in[m]}\boldsymbol{\lambda}_t[i]\cdot\left(\overline{B}_t^{(i)} - c_t(\boldsymbol{x})[i]\right)\right] - \frac{2T^{\frac{1}{4}}}{\rho}R_\tau^D - \left(1 + \frac{2T^{\frac{1}{4}}}{\rho}\right)R_\tau^P,$$

which holds with probability at least $1 - \delta_P$. To get the final bound we proceed as in Theorem C.7 once noticed that, by definition of the strategy mixture space, it holds:

$$\sup_{\boldsymbol{x}\in\mathcal{X}} \sum_{t=1}^{\tau}\left[f_t(\boldsymbol{x}) + \sum_{i\in[m]}\boldsymbol{\lambda}_t[i]\cdot\left(\overline{B}_t^{(i)} - c_t(\boldsymbol{x})[i]\right)\right]$$

$$= \sup_{\boldsymbol{\xi}\in\Xi} \sum_{t=1}^{\tau}\left[\mathbb{E}_{\boldsymbol{x}\sim\boldsymbol{\xi}}\left[f_t(\boldsymbol{x})\right] + \sum_{i\in[m]}\boldsymbol{\lambda}_t[i]\cdot\left(\overline{B}_t^{(i)} - \mathbb{E}_{\boldsymbol{x}\sim\boldsymbol{\xi}}\left[c_t(\boldsymbol{x})[i]\right]\right)\right],$$

and employing Lemma D.7 after noticing that the employment of the Azuma-Höeffding inequality is not necessary. This concludes the proof. $\qquad\square$

# E  Technical Lemmas

In this section, we present some concentration results which are necessary to prove the regret bounds of the algorithms we propose.

**Lemma E.1.** *For any $\delta \in (0,1)$ and far all $t \in [T]$, with probability at least $1 - \delta$, it holds:*

$$\left|\sum_{\tau=1}^{t}\left[f_\tau(\boldsymbol{x}_\tau) - \sum_{i\in[m]}\boldsymbol{\lambda}[i]c_\tau(\boldsymbol{x}_\tau)[i]\right] - \sum_{\tau=1}^{t}\left[\mathbb{E}_{\boldsymbol{x}\sim\boldsymbol{\xi}_\tau}\left[f_\tau(\boldsymbol{x})\right] - \sum_{i\in[m]}\boldsymbol{\lambda}[i]\mathbb{E}_{\boldsymbol{x}\sim\boldsymbol{\xi}_\tau}\left[c_\tau(\boldsymbol{x})[i]\right]\right]\right|$$

$$\leq (4 + 4\|\boldsymbol{\lambda}\|_1)\sqrt{2t\ln\frac{T}{\delta}}.$$

*Proof.* The proof follows from the fact that the quantity of interest is a Martingale difference sequence where the per-step difference is bounded by $(2 + 2\|\boldsymbol{\lambda}\|_1)$. Thus, we employ the Azuma-Höeffding inequality with a Union Bound on the rounds $T$ to conclude the proof. $\qquad\square$

**Lemma E.2.** *For any $\delta \in (0,1)$, far all $t \in [T]$, for any sequence of strategy mixtures $\{\boldsymbol{\xi}_\tau\}_{\tau=1}^{t}$ and for any sequence of Lagrange multipliers $\{\boldsymbol{\lambda}_\tau\}_{\tau=1}^{t}$ it holds, with probability at least $1 - \delta$:*

$$\left|\sum_{\tau=1}^{t}\left[\mathbb{E}_{\boldsymbol{x}\sim\boldsymbol{\xi}_\tau}\left[\bar{f}_\tau(\boldsymbol{x})\right] - \sum_{i\in[m]}\boldsymbol{\lambda}_\tau[i]\mathbb{E}_{\boldsymbol{x}\sim\boldsymbol{\xi}_\tau}\left[\bar{c}_\tau(\boldsymbol{x})[i]\right]\right]\right.$$

$$\left. - \sum_{\tau=1}^{t}\left[\mathbb{E}_{\boldsymbol{x}\sim\boldsymbol{\xi}_\tau}\left[f_\tau(\boldsymbol{x})\right] - \sum_{i\in[m]}\boldsymbol{\lambda}_\tau[i]\mathbb{E}_{\boldsymbol{x}\sim\boldsymbol{\xi}_\tau}\left[c_\tau(\boldsymbol{x})[i]\right]\right]\right| \leq \left(4 + 4\max_{\boldsymbol{\lambda}\in\mathcal{L}}\|\boldsymbol{\lambda}\|_1\right)\sqrt{2t\ln\frac{T}{\delta}}.$$

*Proof.* The proof follows from the fact that the quantity of interest is a Martingale difference sequence where the per-step difference is bounded by $\max_{\boldsymbol{\lambda}\in\mathcal{L}}(2 + 2\|\boldsymbol{\lambda}\|_1)$. Thus, we employ the Azuma-Höeffding inequality with a Union Bound on the rounds $T$ to conclude the proof. $\qquad\square$

