# OpenReview forum: "No-Regret Learning Under Adversarial Resource Constraints: A Spending Plan Is All You Need!"
_NeurIPS.cc/2025/Conference — NeurIPS 2025 poster_

### Official Review · Reviewer_AGGq · 2025-07-02

**Clarity:** 3
**Significance:** 3
**Originality:** 3
**Rating:** 5
**Confidence:** 3

**Summary:**

This work considers online resource allocation (ORA) and online learning with resource constraints (OLRC). To achieve sublinear regret in adversarial scenarios, the authors assume that a spending plan is provided to the learner at the outset. They then establish sublinear regret guarantees with respect to baselines that follow the spending plan in expectation.

**Questions:**

In lines 70–81, the authors compare their approach with [13,33]. Specifically, this work assumes that the spending plan is given at the beginning, whereas [13,33] construct the spending plan during the learning process. Which setting is more challenging? Intuitively, constructing the spending plan on the fly might be more natural.

Why do the baselines only satisfy the spending plan in expectation? Is this an inherent limitation of the proposed method?

Although Remark 4.3 and Theorem D.3 illustrate some results for the bandit case, a corresponding formal theorem is not presented in the main paper. Could the authors clarify or include a formal theorem for the bandit setting?

**Ethical Concerns:**

["NO or VERY MINOR ethics concerns only"]

**Final Justification:**

After carefully reviewing the rebuttal, I have decided to increase my score by 1.

**Limitations:**

yes

**Quality:**

3

**Strengths And Weaknesses:**

Strengths:

Using the spending plan to construct baselines is the main novel contribution of this work and may have wide-ranging applications.

The paper addresses both the ORA and OLRC settings. For OLRC, the analysis covers both the full-information and bandit feedback scenarios. The results appear to be solid and rigorously derived.

Weaknesses:

The work does not provide lower bounds, so it remains unclear whether the obtained results are optimal.

The readability of the paper could be enhanced; for example, adding proof sketches and technical challenges in the main text would help improve clarity.

Additionally, numerical experiments could further increase the work’s appeal by demonstrating its practical performance.

---

> ### Author Rebuttal · Authors · 2025-07-30
>
> We thank the Reviewer for the positive evaluation of our work.
>
> > The work does not provide lower bounds, so it remains unclear whether the obtained results are optimal.
>
> We thank the Reviewer for the interesting question. Assuming $1/\rho_{\min}$ is constant, we can easily state that our bounds for OLRC are provably tight in $T$. Indeed, even in the simpler multi-armed **unconstrained** case (where the decision space is a set of finite arms), a lower bound of the form $R_T=\Omega(\sqrt{T})$ holds (see, [1]). For ORA, the standard $\Omega(\sqrt{T})$ lower bound on the dynamic regret in the stochastic setting, which is presented in [2], still holds when a spending plan is given in input.
>
> As concerns the $1/\rho_{\min}$ dependence in the regret bound, we believe that it is necessary when primal-dual methods are employed. Indeed, a dependency on the inverse of the Slater's parameter is standard for primal-dual algorithms for constrained online learning setting (see [3],[4], and many others). We leave as an interesting open problem the development of algorithms for both ORA and OLRC, which are not primal-dual.
>
> We will include this discussion in the final version of the paper.
>
> > The readability of the paper could be enhanced; for example, adding proof sketches and technical challenges in the main text would help improve clarity.
>
> We thank the Reviewer for the comment. We did our best to include as many insights as possible on the techniques and results within the main text. However, due to space constraints, some technical details had to be deferred to the appendix. We commit to incorporating additional details in the main body of the paper in the final version, using the extra page available.
>
> > In lines 70–81, the authors compare their approach with [13,33]. Specifically, this work assumes that the spending plan is given at the beginning, whereas [13,33] construct the spending plan during the learning process. Which setting is more challenging? Intuitively, constructing the spending plan on the fly might be more natural.
>
> We thank the Reviewer for the interesting question.
>
> First, we would like to highlight that our paper is motivated by the fact that on several internet platforms, the spending plan is given to the budget-management algorithm (coming from some ML model). In that sense, it is useful to develop a set of algorithms that work directly with an arbitrary given spending plan.
>
> That said, building the spending plan is more challenging than assuming to have one as input. But in order to handle that setting, [13] and [33] make assumptions that simplify their analysis, for example allowing them not to deal with the $1/\rho_{\min}$ dependence. Moreover, they deal with the ORA case only, while we study the OLRC and  bandit setting, too.
>
> To conclude, we believe that providing theoretical guarantees (for both ORA and OLRC) while assuming a given spending plan is a fundamental step towards the complete understanding of the learning with spending plan framework, and also arguably the more realistic setup, given that in practice these spending plans will be derived from heuristic ML approaches.  Of course, we also believe that future work can build on our results in order to design meaningful algorithms for constructing spending plans that are optimized to work well with our primal-dual setup.
>
> > Why do the baselines only satisfy the spending plan in expectation? Is this an inherent limitation of the proposed method?
>
> We believe that it is a point in favor of our work. The baseline that satisfies the spending plan in expectation is much stronger than the one that satisfies the plan deterministically. Intuitively, constraints in expectation are weaker than constraints that must hold deterministically. Similar results for simpler settings are provided for instance, in [19].
>
> As further evidence, when comparing with a baseline that satisfies the constraints deterministically, the ORA problem becomes trivial, since it is possible to directly solve a LP problem at each round, finding the optimal per round solution. Instead, since only a sample of the rewards and costs function is observed and the baseline satisfies the spending plan in expectation, the algorithm has to dynamically adapt to the changing environment in order to attain sublinear regret, while satisfying the budget constraints.
>
> Please let us know if further discussion on this point is necessary.
>
> > Although Remark 4.3 and Theorem D.3 illustrate some results for the bandit case, a corresponding formal theorem is not presented in the main paper. Could the authors clarify or include a formal theorem for the bandit setting?
>
> Due to space constraints, we have deferred the formal theorem in the Appendix (see Theorem D.3. and Theorem D.8.). We commit to inserting it in the main text in the final version of the paper, where we will be allowed to use an additional page.
>
> [1] "Bandit algorithms", Lattimore and Szepesvári (2020)
>
> [2] "The best of many worlds: Dual mirror descent for online allocation problems", Balseiro et al. (2023)
>
> [3] "Exploration-Exploitation in Constrained MDPs", Efroni et al. (2020)
>
> [4] "Truly No-Regret Learning in Constrained MDPs", Müller et al. (2024)

---

> > ### Comment · Reviewer_AGGq · 2025-08-04
> >
> > Thank you for your detailed response. After carefully reviewing the rebuttal, I have decided to increase my score by 1.

---

### Official Review · Reviewer_5sDf · 2025-07-02

**Clarity:** 4
**Significance:** 3
**Originality:** 3
**Rating:** 4
**Confidence:** 3

**Summary:**

This paper studies the online decision-making problem under adversarial resource constraints. In this problem, a decision maker has a fixed budget for multiple resources. Requests arrive sequentially, and each request generates a reward and consumes resources based on a given allocation. Before the decision-making process, a spending plan, which specifies the spending budget for each step (or request), is provided to the decision maker. The goal of the algorithm is to minimize the regret with respect to a benchmark that follows the spending plan.

The paper considers two settings based on the timing of decisions:

- Online resource allocation, where the algorithm observes the reward and resource consumption functions before making decisions.

- Online learning with resource constraints, where the algorithm makes decisions first and then observes the reward and consumption functions (under full feedback or bandit feedback).

The authors develop online algorithms for both settings. In the online resource allocation setting, the algorithm achieves a dynamic regret of $\tilde{O}(\sqrt{T}/\rho_{\min})$. In the online learning with resource constraints setting, the algorithm achieves a static regret of $\tilde{O}(\sqrt{T}/\rho_{\min})$, where $\rho_{\min}$ is the minimum spending budget across all steps and resources.

To address scenarios where $\rho_{\min}$ is small, the paper proposes modifications by redefining $\rho_{\min}$ and the spending plan. These modifications yield regret bounds independent of $\rho_{\min}$, which is desirable when $\rho_{\min}$ is small. Additionally, the paper demonstrates robustness to sub-linear errors in the spending plan.

**Questions:**

- In the adversarial setting, a worst-case instance could involve that all resources are consumed in a single step, with no resource consumption in other steps. This corresponds to a spending plan where one step receives the entire budget and all others receive zero. Can the extended version of the algorithm handle this case? What happens if there are errors in the spending plan in such scenarios?

- The regret bounds depend on $\rho_{\min}$, which is defined as the minimum budget across all steps and resources. Would it be possible to redefine $\rho_{\min}$ to exclude zero elements, using the minimum among all non-zero elements instead?

- In practice, the full spending plan may not be available at the beginning of the decision process. Can the algorithm handle an online revelation of the spending plan? Moreover, the horizon $T$ is assumed to be known in advance, which is not always realistic. Can the algorithm be adapted for an unknown time horizon?

- In the theorems in Section 5, the parameter $\rho$ appears in many expressions, but I believe it should be $\rho_{\min}$. Could you clarify this?

**Ethical Concerns:**

["NO or VERY MINOR ethics concerns only"]

**Final Justification:**

The authors have answered my questions; however, my original concerns about the algorithmic novelty and lower bound are not fully resolved. So I will keep my positive rating, albeit with some reservations.

**Quality:**

3

**Strengths And Weaknesses:**

Strength
- The paper addresses an important online decision-making problem with broad applications. The notion of a spending plan is a practical and reasonable addition, helping to overcome worst-case impossibility results.

- The systematic algorithm design and analysis for both settings, online resource allocation and online learning with constraints, are nice.

- The analysis of robustness to errors in the spending plan is a valuable and practical addition to the main results.


Weakness


- The algorithmic ideas are not entirely new. The online resource allocation algorithm closely resembles that of Santiago et al. (ICML 2020), with the main change being the use of a predefined spending plan instead of a uniform budget $B/T$. The online learning with constraints approach shares similarities with the bandits with knapsacks literature. The paper lacks a thorough comparison between its proposed algorithm and those existing approaches.

- There are no lower bounds provided, making it unclear how tight the regret bounds are.

---

> ### Author Rebuttal · Authors · 2025-07-30
>
> We thank the Reviewer for the positive evaluation of our work.
>
> > The algorithmic ideas are not entirely new. The online resource allocation algorithm closely resembles that of Santiago et al. (ICML 2020), with the main change being the use of a predefined spending plan instead of a uniform budget $B/T$. The online learning with constraints approach shares similarities with the bandits with knapsacks literature. The paper lacks a thorough comparison between its proposed algorithm and those existing approaches.
>
> We agree with the Reviewer that the novelty of our work does not primarily lie in the primal-dual template. Nonetheless, we believe that this is a common feature of many works on both ORA and OLRC (see [1], [2], [3]), where the main contribution lies in the analysis, while the algorithms are subject to small---but fundamental---changes. Furthermore, notice that Algorithm 3 is indeed novel in the literature; thus, we believe that our work is valuable from an algorithmic perspective, too.
>
> We commit to inserting a more comprehensive discussion in the final version of the paper.
>
> > There are no lower bounds provided, making it unclear how tight the regret bounds are.
>
> We thank the Reviewer for the insightful comment. Assuming $1/\rho_{\min}$ is constant, we can easily state that our bounds for OLRC are provably tight in $T$. Indeed, even in the simpler multi-armed **unconstrained** case (where the decision space is a set of finite arms), a lower bound of the form $R_T=\Omega(\sqrt{T})$ holds (see, [4]). For ORA, the standard $\Omega(\sqrt{T})$ lower bound on the dynamic regret in the stochastic setting, which is presented in [1], still holds when a spending plan is given in input.
>
> As concerns the $1/\rho_{\min}$ dependence in the regret bound, we believe that it is necessary when primal-dual methods are employed. Indeed, a dependency on the inverse of the Slater's parameter is standard for primal-dual algorithms for constrained online learning setting (see [5],[6], and many others). We leave as an interesting open problem to develop algorithms for both ORA and OLRC, which are not primal-dual.
>
> We will surely include this discussion in the final version of the paper.
>
> > In the adversarial setting, a worst-case instance could involve that all resources are consumed in a single step, with no resource consumption in other steps. This corresponds to a spending plan where one step receives the entire budget and all others receive zero. Can the extended version of the algorithm handle this case? What happens if there are errors in the spending plan in such scenarios?
>
> We thank the Reviewer for pointing out this aspect. As is standard in the literature, the costs (and the per-round spending plan) are assumed to be in $[0,1]$ and we are in the regime $B=\Omega(T)$; thus, it is not possible that the entire budget is allocated in a single round. Technically, it is well known that it is impossible to learn in settings in which the budget can be depleted in a single round. Intuitively, this is related to learning problems with a constant number of rounds, in which regret algorithms are ineffective.
>
> Please let us know if further discussion is necessary.
>
> > The regret bounds depend on $\rho_{\min}$, which is defined as the minimum budget across all steps and resources. Would it be possible to redefine $\rho_{\min}$ to exclude zero elements, using the minimum among all non-zero elements instead?
>
> We thank the Reviewer for the interesting question. The answer is yes and we will explain this in the paper.
>
> In general, some preprocessing steps can improve the empirical results. Removing zero elements is a good starting point that does not change the results. More generally, it is possible to remove all the elements below a given threshold. Specifically, when the number of rounds where $B_t^{(i)}$ is small is of order $\mathcal{O}(\sqrt{T})$, those rounds can be erased (playing the void action) and it is possible not to count the associated $B_t^{(i)}$ in the computation of $\rho_{\min}$, without changing the theoretical guarantees of the algorithm.
>
> Please let us know if further discussion is necessary.
>
> > In practice, the full spending plan may not be available at the beginning of the decision process. Can the algorithm handle an online revelation of the spending plan? Moreover, the horizon $T$ is assumed to be known in advance, which is not always realistic. Can the algorithm be adapted for an unknown time horizon?
>
> We thank the Reviewer for the interesting question. We believe that handling an online revelation of the spending plan and an unknown time horizon is indeed possible. For the unknown time horizon, it is sufficient to employ a primal and a dual regret minimizer that works with an adaptive time horizon (e.g., online gradient descent with adaptive learning rate for the dual). For the online revelation of the time horizon, we believe that a doubling trick approach could work. Specifically, the algorithm is instantiated with an estimate of $\rho_{\min}$, which we call $\tilde{\rho}$ (and is initialized to $1$). Every time the online spending plan reveals a new $B^{(i)}_t\leq \tilde \rho/2$, we reset the algorithm with the new $\tilde \rho = B^{(i)}_t$ and the results still hold up to logarithmic terms.
>
> We will include this discussion in the final version of the paper.
>
> > In the theorems in Section 5, the parameter $\rho$ appears in many expressions, but I believe it should be $\rho_{\min}$. Could you clarify this?
>
> In Section 5.1, we obtain regret bounds which are independent of $\rho_{\min}$. The quantity the Reviewer noticed is $\rho=B/T$, which is constant by definition. Since this is one of the main contributions of our work, please let us know if further discussion on this point is necessary.
>
> [1] "Dual Mirror Descent for Online Allocation Problems", Balseiro et al. (2020)
>
> [2] "The best of many worlds: Dual mirror descent for online allocation problems", Balseiro et al. (2023)
>
> [3] "Adversarial Bandits with Knapsacks", Immorlica et al. (2023)
>
> [4] "Bandit algorithms", Lattimore and Szepesvári (2020)
>
> [5] "Exploration-Exploitation in Constrained MDPs", Efroni et al. (2020)
>
> [6] "Truly No-Regret Learning in Constrained MDPs", Müller et al. (2024)

---

> > ### Comment · Reviewer_5sDf · 2025-08-04
> >
> > Thanks for your detailed response to my comments and questions. I will retain my positive rating.

---

### Official Review · Reviewer_VN8t · 2025-07-03

**Clarity:** 3
**Significance:** 3
**Originality:** 3
**Rating:** 4
**Confidence:** 3

**Summary:**

The paper proposes a framework for online decision-making with resource constraints, focusing on adversarial environments where standard regret benchmarks are unachievable. The key contribution is the introduction of a spending plan benchmark: a user-specified resource consumption profile that serves as a feasible benchmark for regret analysis. The authors design primal-dual algorithms, applicable to both full and bandit feedback settings, that achieve sublinear regret relative to this spending plan. The analysis includes a meta-algorithm to handle small minimum budget values and provides robustness guarantees for small deviations from the spending plan. The work aims to unify online resource allocation and online learning with resource constraints under a common regret framework.

**Questions:**

1) In the ORA setting, since the reward and cost functions are fully known before action selection, the "regret" essentially measures the loss of the proposed dynamic optimization heuristic relative to a feasible baseline spending plan. There is no learning of unknown quantities involved. Could the authors clarify this point and distinguish their notion of regret from standard regret in learning settings?

2) The authors claim in Remark 2.2 that sublinear regret is impossible in all settings without a spending plan. However, in the stochastic setting, standard benchmarks are achievable with sublinear regret. The authors may wish to clarify this distinction.

**Ethical Concerns:**

["NO or VERY MINOR ethics concerns only"]

**Final Justification:**

Authors clarified key points, I increased my score to borderline accept.

**Limitations:**

The paper lacks a discussion on limitations and potential negative societal impacts.

**Paper Formatting Concerns:**

No formatting concerns.

**Quality:**

3

**Strengths And Weaknesses:**

Strengths:

1) The paper introduces a general framework for online decision-making under resource constraints that applies to both adversarial and stochastic settings, including bandit and full-information feedback.

2) The spending plan concept provides a structured and feasible baseline in adversarial environments, allowing the authors to derive regret guarantees that are meaningful when competing with a pre-specified resource usage profile.

3) The algorithmic design is modular, leveraging black-box regret minimizers in the primal updates and providing adaptability across different feedback structures.

Weaknesses:

1) The paper’s novelty lies primarily in the design of the spending plan benchmark and its associated regret analysis, rather than in new algorithmic techniques.

2) The choice of spending plan is external and assumed given, yet the regret guarantee depends heavily on this plan’s quality. The paper does not discuss how to design or learn a good spending plan or analyze sensitivity to suboptimal plans.

3) The regret notion in the ORA setting does not reflect classical learning. Since the algorithm operates with full feedback before action selection, there is no learning of unknown reward or cost functions. The regret measure essentially tracks the loss of a dynamic optimization heuristic relative to the spending plan, rather than regret in the traditional online learning sense.

---

> ### Author Rebuttal · Authors · 2025-07-30
>
> We thank the Reviewer for the effort in evaluating our work.
>
> > The paper’s novelty lies primarily in the design of the spending plan benchmark and its associated regret analysis, rather than in new algorithmic techniques.
>
> We agree with the Reviewer that the novelty of our work does not primarily lie in the primal-dual template. However, we highlight that this is a common feature of many works on both ORA and OLRC (e.g., [1], [2], [3]). The main contribution in this literature, as well as in our work, lies in the analysis and in the small--but fundamental--changes that are necessary in order to achieve various properties of the general primal-dual template.
>
> Furthermore, while Algorithm 3 acts as a wrapper for the primal-dual template, it is, to the best of our knowledge, a novel technique in the literature. We believe this contribution is algorithmically valuable as well.
>
> > The choice of spending plan is external and assumed given, yet the regret guarantee depends heavily on this plan’s quality. The paper does not discuss how to design or learn a good spending plan or analyze sensitivity to suboptimal plans.
>
> We believe that providing theoretical guarantees (for both ORA and OLRC) while assuming a good spending plan is given as input is a fundamental and valuable step towards the complete understanding of the learning with spending plan framework. We strongly believe that future works can build on our results in order to design meaningful spending plans. Indeed, our baseline follows the spending plan in expectation only. This is important because it means that our setup can be combined with spending-plan estimation algorithms that then feed those estimates to our algorithms.
>
>
> Finally, on the statement that we do not analyze sensitivity to suboptimal spending plans, this is not correct. In Section 5.2, we show how our algorithms are robust to baselines that do not follow the spending plan exactly. Notice that by relaxing the baseline in this way, we are actually showing results exactly for the case where the spending plan is not accurate: if you follow the predicted spending plan, but it is not accurate, then there will be some sequence of relaxation parameters $\epsilon_t$ such that the optimal spending rates are captured by that relaxed plan. Thus, our results in Section 5.2 actually show that our approach degrades gracefully with the amount of inaccuracy in the spending plan: you incur regret linear in the amount of error. Looking at the paper, we did not add enough explanation of the equivalence between these two things. In the final version of the paper we will make this much clearer.
>
> > The regret notion in the ORA setting does not reflect classical learning. Since the algorithm operates with full feedback before action selection, there is no learning of unknown reward or cost functions. The regret measure essentially tracks the loss of a dynamic optimization heuristic relative to the spending plan, rather than regret in the traditional online learning sense. In the ORA setting, since the reward and cost functions are fully known before action selection, the "regret" essentially measures the loss of the proposed dynamic optimization heuristic relative to a feasible baseline spending plan. There is no learning of unknown quantities involved. Could the authors clarify this point and distinguish their notion of regret from standard regret in learning settings?
>
> We respectfully disagree with the Reviewer on this point. The Reviewer may be conflating "classical learning" and the online resource allocation literature. Our (dynamic) regret notion in the ORA setting is standard for the ORA literature (see e.g., [1,2,4,5,6]). Notice that, even though rewards and costs are fully revealed before action selection, there is still learning involved. What is being learned is the appropriate way to bid in order to expend the budget at the correct rate. The reason this must be learned is that the requests (i.e., the reward and cost functions) are sampled from some underlying distributions (in the stochastic setting), or generated arbitrarily (in the adversarial setting). In either case, the online aspect arises because the algorithm must learn how to act in a way such that the budget expenditure is smoothed out in a reasonable way. All of this occurs because the resource constraint means that these decisions are connected across rounds, and thus affect each other.
>
> We will surely include this discussion in the final version of the paper.
>
> > The authors claim in Remark 2.2 that sublinear regret is impossible in all settings without a spending plan. However, in the stochastic setting, standard benchmarks are achievable with sublinear regret. The authors may wish to clarify this distinction.
>
> We would like to clarify the quote from Remark 2.2, which says:
>
> "It is well known that when no spending plan is available, it is impossible to achieve sublinear regret bounds in all the settings presented in this work ..."
>
> This quote is correct. The Reviewer's comment only applies to the classical stochastic setting where the **same** distribution is used at every time step, which is not a setting presented in our paper. In the final version of the paper, we will more clearly specify "unlike the setting with a fixed distribution that does not vary with time".
>
> In more detail, in our work, the rewards and costs are sampled at each episode from adversarially changing distributions. Thus, our setting is more general than the standard adversarial one---where the rewards and costs change over time but the true function is observed---and the lower bounds from the adversarial setting still hold in our case. In ORA with adversarial rewards and costs, sublinear dynamic regret is provably impossible (see [4]). A similar result extends to OLRC (with adversarial rewards and costs) for the static regret.
>
> We hope to have addressed the Reviewer's concerns. Please let us know if further discussion is necessary.
>
> [1] "Dual Mirror Descent for Online Allocation Problems", Balseiro et al. (2020)
>
> [2] "The best of many worlds: Dual mirror descent for online allocation problems", Balseiro et al. (2023)
>
> [3] "Adversarial Bandits with Knapsacks", Immorlica et al. (2023)
>
> [4] "Learning in repeated auctions with budgets: Regret minimization and equilibrium", Balseiro et al. (2019)
>
> [5] "Online Resource Allocation under Horizon Uncertainty", Balseiro et al. (2023)
>
> [6] "Robust budget pacing with a single sample.", Balseiro et al. (2023)

---

> > ### Comment · Reviewer_VN8t · 2025-08-05
> >
> > Thanks for the clarifications.

---

### Official Review · Reviewer_yQdM · 2025-07-07

**Clarity:** 3
**Significance:** 3
**Originality:** 2
**Rating:** 5
**Confidence:** 3

**Summary:**

The paper studies the problems of online resource allocation (where rewards and costs are observed before action selection) and the online learning with resource constraints (where rewards and costs they are observed after action selection, under full feedback or bandit feedback). While it is known that a sublinear regret isn't achievable by any spending plan in these setting, the current paper considers the easier, but still realistic, setting where a budget plan is given to the algorithm. The paper provides algorithms with sublinear regret bounds in this case.

**Questions:**

Please elaborate on how to reduce the bandit setting to the full information setting.

**Ethical Concerns:**

["NO or VERY MINOR ethics concerns only"]

**Limitations:**

The authors did not explicitly discuss the limitations of their work in the main body.

**Paper Formatting Concerns:**

No concerns.

**Quality:**

3

**Strengths And Weaknesses:**

Strengths:
- The paper tackles a practically relevant problem in ML.
- The algorithms presented are intuitive and simple.
- The paper is very well-written and easy to follow.

Weaknesses:
- Having access to a good budget plan is not always possible, and so the methods proposed may have a narrow scope.
- The paper does not provide any experimental validation.
- There are a couple of places where the paper would have benefited from more detailed explanations (see my comments below).

Comments:
- After Theorem 4.2, it would have been helpful to elaborate a little more on how the bandit setting can be reduced to the full-information setting.
- Same comment for Section 5.1 (lines 352-358); more explanation on how to deal with small rho would have been very helpful.
- $\rho$ in Algorithm 3 (and in Section 5.1 in general) is not defined/scoped.

---

> ### Author Rebuttal · Authors · 2025-07-30
>
> We thank the Reviewer for the positive evaluation of our work.
>
> > Having access to a good budget plan is not always possible, and so the methods proposed may have a narrow scope.
>
> We agree with the Reviewer that having access to a good spending plan is not always possible. Nonetheless, spending plans are used in practice, and thus it is of interest to develop a theoretical understanding of how they should be used. We believe that our work of providing theoretical guarantees (for both ORA and OLRC) while assuming a good spending plan is a fundamental and necessary step towards an understanding of the real-world problem of how to learn with a spending plan. Second, our baseline follows the spending plan in expectation only. This is important because it means that our setup can be combined with spending-plan estimation algorithms that then feed those estimates to our algorithms. Finally, in Section 5.2, we show how our algorithms are robust to baselines that do not follow the spending plan, thus improving their applicability in practice.
>
> > After Theorem 4.2, it would have been helpful to elaborate a little more on how the bandit setting can be reduced to the full-information setting.
>
> We thank the Reviewer for the comment. We did our best to give as many insights as possible about our algorithms, but due to space constraints, we had to prioritize some settings over others.
>
> We better highlight the relation between full and bandit feedback in the following. Notice that our primal-dual algorithm exploits the no-regret guarantees of the primal and the dual regret minimizer. Thus, if a bandit regret minimizer yields guarantees similar to those for OLRC with full feedback, our analysis for the bandit feedback follows immediately from the full feedback one.
> The main difference lies in the fact that, in general, the guarantees provided by primal regret minimizers under bandit feedback differ from those obtained with full feedback. This is because the Lagrangian observed as feedback is different, which can, in principle, lead to different optima. Nonetheless, we can employ the fact that, by definition of the strategy mixture space $\Xi$, it holds that $\sup_ {\boldsymbol{\xi}\in\Xi} \sum_{t=1}^\tau r_t^P = \sup_ {\boldsymbol{x}\in\mathcal{X}} \sum_{t=1}^\tau r_t^P$, where $r_t^P$ is the (Lagrangian) loss built for the full feedback setting. Specifically, the optimal solution defined over the "full-feedback" Lagrangian is a pure strategy; thus, the optimal solution for the "full-feedback" Lagrangian and the "bandit" one are equivalent (see lines 321-322). Consequently, the regret guarantees of the primal bandit feedback regret minimizer are comparable to the full feedback one, up to the fact that, in the bandit feedback, the regret bound holds with high probability.
>
> Please let us know if further discussion is necessary.
>
> We commit to enlarging the explanation on this part in the final version of the paper.
>
> > Same comment for Section 5.1 (lines 352-358); more explanation on how to deal with small rho would have been very helpful.
>
> Similarly to the previous answer, we commit to enlarging the explanation on this part in the final version of the paper.
>
> From a technical perspective, the key problem for handling small $\rho_{\min}$ is that the Lagrangian space must be instantiated so that the Lagrangian may reach a value of $1/\rho_{\min}$ to compensate budget violation. Thus, the primal regret bound (a similar reasoning holds for the dual, too), which scales linearly in the payoff range, would be of order $1/\rho_{\min}\sqrt{T}$, which can be linear for small $\rho_{\min}$. In order to get a sublinear result, we cap the Lagrangian to $T^{1/4}/\rho$, attaining a regret of order $T^{3/4}$, since $\rho$ is constant by definition. Nonetheless, capping the Lagrangian to $T^{1/4}/\rho$ is not sufficient to "satisfy" the budget constraints, that is, to allow the budget violation to compensate large rewards in the Lagrangian function. Thus, we rescale the spending plan with the same quantity, in order to be slightly pessimistic in the budget constraints part of the Lagrangian and, subsequently, to allow a Lagrangian of order $T^{1/4}/\rho$ to be large enough to satisfy the new pessimistic constraint.
>
> Please let us know if further discussion is necessary.
>
> > $\rho$ in Algorithm 3 (and in Section 5.1 in general) is not defined/scoped.
>
> We thank the Reviewer for pointing out this aspect. The parameter $\rho$ is defined in the preliminaries at line 101. Specifically, $\rho$ is the ratio between the overall budget $B$ and the horizon $T$, which is constant by definition since $B=\Omega(T)$. Nonetheless, we agree with the Reviewer that, for the sake of clarity, the definition of $\rho$ can be remarked in the final section. We commit to doing it in the final version of the paper.

---

> > ### Comment · Reviewer_yQdM · 2025-08-08
> > **Reply**
> >
> > Thank you for you response. I will maintain my positive assessment.

---

### Decision · Program_Chairs · 2025-09-17

**Decision:**

Accept (poster)

**Comment:**

Overall, reviewers are positive about the paper. They think the paper studies a well-motivated and practically relevant problem, and the paper offers a nice framework to tackle the problem.

The following two papers might be worth citing in "Related Work" section "Online Learning with Resource Constraints" paragraph (and  they might be solving special cases of the submission):

Contextual Bandits with Packing and Covering Constraints: A Modular Lagrangian Approach via Regression.  Aleksandrs Slivkins, Xingyu Zhou, Karthik Abinav Sankararaman, Dylan J. Foster. COLT 2023, JMLR 2024.
Budget Pacing in Repeated Auctions: Regret and Efficiency without Convergence. Jason Gaitonde, Yingkai Li, Bar Light, Brendan Lucier, and Aleksandrs Slivkins. ITCS 2023.

In particular, the open problem (emphasized in conclusions) of the COLT 2023 paper might be quite related to this paper.